# PRIVACY-AWARE VIDEO ANOMALY DETECTION THROUGH ORTHOGONAL SUBSPACE PROJECTION

## ABSTRACT

Video anomaly detection (VAD) is central to modern surveillance, yet most existing methods optimize for accuracy while overlooking critical ethical concerns such as privacy and transparency. For deployment in real-world settings, VAD should not only detect anomalies reliably but also respect fundamental privacy principles. We propose the Orthogonal Projection Layer (OPL), a lightweight architectural module that suppresses task-irrelevant variations, including background clutter and noise, to produce representations focused on anomaly-relevant cues. Faces, unlike other cues such as gait or body pose, are highly sensitive biometric identifiers: they uniquely reveal identity, are tightly regulated by data protection laws, and pose immediate risks of misuse. To address the privacy risks inherent in human-centered anomalies, we extend this idea to the Guided OPL (G-OPL). Using only weak supervision from face-presence indicators, G-OPL selectively removes facial attributes while retaining non-identifying human features needed for anomaly detection. A cosine alignment loss ensures that facial information is systematically captured and neutralized, without requiring identity labels or adversarial training. We further introduce a privacy-aware evaluation framework that jointly assesses anomaly detection accuracy, privacy preservation, and interpretability. Our analysis uncovers how projection layers filter sensitive information, why this improves transparency, and under what conditions ethical design also enhances robustness. Extensive experiments confirm that embedding ethical constraints directly into model design strengthens privacy protection while maintaining, and in some cases improving, anomaly detection performance. These results position projection-based architectures as a principled path toward trustworthy and deployable VAD systems.

## 1 INTRODUCTION

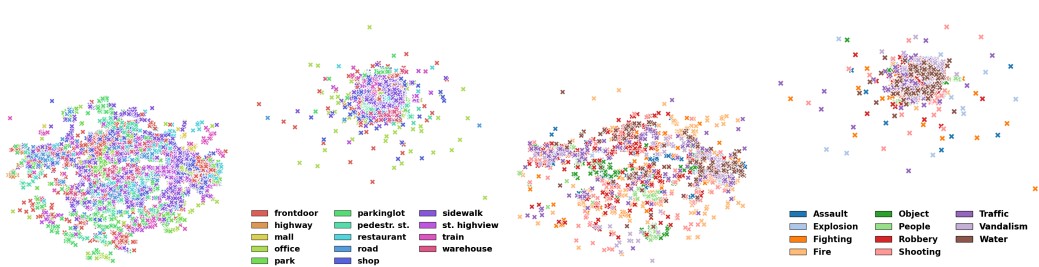

(a) OPL: By scenario  (b) G-OPL: By scenario  (c) OPL: By anomaly type  (d) G-OPL: By anomaly type

Figure 1: UMAP of *removed features* on MSAD. OPL removes broad nuisance factors like backgrounds, producing loosely clustered features that reflect scenario variations. G-OPL, guided by face presence, isolates sensitive biometric cues, resulting in a more compact, overlapping cluster that does not align with anomaly types. This contrast shows how OPL and G-OPL complement each other by disentangling nuisance and privacy-sensitive information.

Video Anomaly Detection (VAD) is a longstanding challenge in computer vision, with wide-ranging applications in public safety, infrastructure monitoring, and surveillance systems (Wang et al., 2019; Ding & Wang, 2024; 2025; Zhu et al., 2024). Recent progress in VAD has largely focused on pushing the boundaries of performance through increasingly complex architectures and learning objectives.

Figure 2: Pipeline overview. The Orthogonal Projection Layer (OPL, light blue) and its guided variant (G-OPL, dark blue and red) are lightweight, fully differentiable, and easily integrate into standard anomaly detection architectures (black). OPL suppresses nuisance factors through orthogonal projections, while G-OPL explicitly removes sensitive attributes via semantic suppression ($\lambda_{\text{face}}\mathcal{L}_{\text{cos}}$) guided by attribute signals (*e.g.*, faces, red arrows). An orthogonality regularizer ($\lambda_{\text{orth}}\mathcal{L}_{\text{orth}}$) ensures stable optimization. OPL / G-OPL can be flexibly inserted after a feature extractor (or encoder), individual layer, or block. This design embeds privacy directly into the model, enabling precise control over sensitive information at the representation level with minimal impact on detection performance. See Appendix for *practical insights on layer placement* and *privacy protection*.

However, much of this work overlooks critical ethical dimensions, particularly those concerning privacy and interpretability. As AI systems transition into high-stakes, real-world environments, addressing these dimensions is not optional but essential for responsible deployment.

A key limitation of existing VAD models is their tendency to retain and exploit information that is either irrelevant to the task or ethically sensitive. Attributes such as faces, clothing styles, or background context are rarely necessary for identifying anomalies and can even introduce bias, overfitting, or privacy violations. Without explicit mechanisms to suppress these factors, current models risk learning representations that undermine both fairness and user trust. To address this, we propose a principled architectural solution for embedding ethical constraints directly into the learning process. We introduce the Orthogonal Projection Layer (OPL), a lightweight, differentiable module that learns to project intermediate features away from subspaces containing task-irrelevant or undesirable information. Initially, OPL focuses on removing general nuisance factors such as lighting variations, background clutter, and camera motion, guided by the anomaly detection objective itself.

We further extend this idea with the Guided Orthogonal Projection Layer (G-OPL), which targets the removal of explicitly sensitive attributes such as faces. G-OPL uses weak supervision via lightweight face detectors (*e.g.*, RetinaFace) to extract face embeddings on the fly. These embeddings serve as geometric guides, and a cosine similarity loss encourages the projection basis to align with face-related components in the latent space, so that they can be effectively suppressed. Importantly, our method avoids adversarial training or gradient reversal, leading to more stable optimization and interpretable subspace disentanglement. Fig. 1 illustrates the interpretability benefits of our approach: OPL isolates nuisance factors related to scene dynamics, while G-OPL compacts sensitive biometric information, revealing their complementary roles in shaping ethical feature spaces. We perform an extensive analysis of where and how to integrate these projection layers, what types of information they remove, and how they impact both model utility and ethical alignment. We also introduce novel privacy-aware metrics tailored to anomaly detection settings, enabling a more holistic evaluation of ethical trade-offs in VAD. Our **contributions** are summarized as follows:

i. We propose OPL, a differentiable projection mechanism that filters task-irrelevant features by learning orthogonal subspaces, and extend it to Guided OPL (G-OPL) for explicitly suppressing sensitive attributes such as faces.

ii. We introduce privacy-aware evaluation metrics for VAD, enabling principled assessment of the trade-offs between performance, interpretability, and privacy preservation.

iii. We conduct thorough experiments analyzing the placement and frequency of projection layers, the nature of removed information, and their impact on both anomaly detection and ethical alignment, offering actionable insights for ethical model design.

## 2 RELATED WORK

**Privacy-preserving representation learning.** Learning representations that do not encode sensitive attributes is a core objective in privacy-aware machine learning. A common strategy is adversarial training with gradient reversal (Ganin et al., 2016), where an auxiliary classifier attempts to predict

sensitive attributes, and the encoder is trained to inhibit this through gradient inversion. Other approaches perform subspace manipulation, such as nullspace projection (Kim et al., 2019; Ravfogel et al., 2020), which analytically removes directions associated with sensitive information. These methods have primarily been developed for classification tasks in fairness and demographic privacy contexts. In contrast, we adapt these principles to the more complex VAD setting, which presents unique challenges due to the spatio-temporal nature of video and the lack of explicit labels for nuisance or sensitive factors. Our method avoids adversarial objectives, using geometric supervision via cosine similarity and weak face signals for more stable and interpretable training. Here, geometric refers to latent feature alignment, not physical 3D geometry.

**Orthogonal projection and feature disentanglement.** Orthogonal projection has been used in disentangled representation learning to separate task-relevant and task-irrelevant information (Ranasinghe et al., 2021; Moyer et al., 2018; Sarhan et al., 2020). These methods often rely on known sensitive labels or contrastive setups in static image tasks. Our OPL generalizes this idea for dynamic, unlabeled video data by learning to remove nuisance subspaces directly through anomaly detection loss. We further introduce G-OPL, which incorporates weak supervision from faces to target privacy-relevant information. Unlike classical projection approaches, G-OPL is fully differentiable, requires no explicit sensitive labels, and aligns privacy removal with interpretable geometric signals.

**Ethics and interpretability in VAD.** While ethical AI has received increasing attention, especially around fairness and transparency, most work in VAD treats these aspects as afterthoughts. Techniques such as dataset balancing or saliency visualization are often used post hoc to audit ethical behavior. Explainable AI methods like Grad-CAM (Selvaraju et al., 2020) and causal analysis in VAD (Du et al., 2024) offer interpretability but do not directly influence representation learning. Our approach goes further by embedding ethical constraints at the model level through projection layers that structurally suppress sensitive content in the latent space. This leads to improved privacy and interpretability, all without sacrificing scalability or compatibility with existing architectures.

**Comparison with prior projection-based methods.** While orthogonal projection has been explored in fairness and disentangled representation learning, like INLP (Ravfogel et al., 2020), OPL-2021 (Ranasinghe et al., 2021), DAMS (An et al., 2025) and CAE-LSP (Yu et al., 2023). These methods differ from ours in terms of objectives, assumptions, and design. Most prior work either assumes access to ground-truth sensitive labels (*e.g.*, INLP), operates in static or low-dimensional settings (*e.g.*, CAE-LSP), or applies generic decorrelation losses (*e.g.*, OPL-2021) without explicit suppression. In contrast, our G-OPL learns task-specific sensitive subspaces using only weak signals (*e.g.*, face presence or embeddings), and uniquely integrates privacy-aware evaluation metrics tailored for VAD. To the best of our knowledge, this is the first projection-based method designed for privacy-aware video anomaly detection under realistic constraints.

## 3 METHOD

**Overview.** We propose the Orthogonal Projection Layer (OPL) that filters nuisance directions from features via learned subspace projections. We further extend this to Guided OPL (G-OPL), which enables targeted suppression of sensitive features (*e.g.*, faces) using weak supervision (*e.g.*, face presence). Both modules are lightweight, fully differentiable, and compatible with standard VAD architectures. They can be flexibly inserted after a feature extractor (or encoder), individual layer, or block. Fig. 2 provides an overview of the model architecture with OPL and G-OPL layers inserted. To evaluate the utility-privacy trade-off, we introduce three novel metrics tailored to anomaly detection. The appendix provides theoretical motivation, clarifies how our model-level protection differs from dataset- and learning-level approaches, and situates our work within the broader literature.

**Notations.** Throughout this paper, scalars are denoted by italic letters (*e.g.*, $x$); vectors by lowercase boldface (*e.g.*, $\boldsymbol{x}$); and matrices by uppercase boldface (*e.g.*, $\boldsymbol{X}$).

### 3.1 ORTHOGONAL PROJECTION LAYER

The *Orthogonal Projection Layer* (OPL) is designed to explicitly remove task-irrelevant information from intermediate feature representations. Unlike classical methods such as PCA or fixed subspace removal, our OPL learns the nuisance subspace jointly with the main task and is fully differentiable, allowing adaptive identification and removal of nuisance directions during training.

Formally, let $\boldsymbol{f} \in \mathbb{R}^d$ be an intermediate feature extracted from either a backbone network (*e.g.*, feature extractor) or a layer. We define a learnable weight matrix $\boldsymbol{W} \in \mathbb{R}^{k \times d}$ whose rows parameterize the nuisance subspace to be removed ($1 < k < d$). This matrix can be learned from scratch alongside the anomaly detection objective. To obtain a numerically stable and orthonormal basis for the nuisance subspace, we perform QR decomposition on the transpose of $\boldsymbol{W}$:

$$\boldsymbol{W}^\top = \boldsymbol{Q}\boldsymbol{R}, \tag{1}$$

where $\boldsymbol{Q} \in \mathbb{R}^{d \times k}$ contains $k$ orthonormal basis vectors spanning the nuisance directions, and $\boldsymbol{R}$ is an upper-triangular matrix. Using $\boldsymbol{Q}$, we construct the orthogonal complement projection matrix:

$$\boldsymbol{P} = \boldsymbol{I}_d - \boldsymbol{Q}\boldsymbol{Q}^\top, \tag{2}$$

with $\boldsymbol{I}_d$ being the $d \times d$ identity matrix. Projecting $\boldsymbol{f}$ onto the orthogonal complement yields:

$$\boldsymbol{f}_{\text{proj}} = \boldsymbol{P}\boldsymbol{f} = \boldsymbol{f} - \boldsymbol{Q}\boldsymbol{Q}^\top\boldsymbol{f}, \tag{3}$$

removing nuisance components while retaining task-relevant information.

> Unlike fixed subspace removal techniques (*e.g.*, PCA), our OPL learns the nuisance directions *jointly* with the anomaly detection objective. This dynamic learning enables the network to identify and suppress task-irrelevant factors that are most detrimental to detection performance. Our OPL stands out by offering a plug-and-play, learnable projection layer that is task-specific, modular, and interpretable. The integration of QR decomposition ensures practical stability and efficiency lacking in prior projection-based approaches. These properties establish OPL as a foundational building block for task-specific and robust video analysis systems.

## 3.2 Guided OPL for Privacy-Aware Face Suppression

While OPL removes nuisance factors like background or lighting, it may leave sensitive attributes such as identity, gender, or age in the features. We propose the *Guided Orthogonal Projection Layer* (G-OPL) to selectively suppress these privacy-sensitive components using weak supervision.

**Facial identity as a sensitive signal.** We focus on facial identity not for recognition, but for removing biometric traits implicitly encoded in facial features. Although typical VAD datasets lack identity annotations, the mere presence of faces introduces privacy risks. Intermediate neural features can retain identity cues that, if leaked or combined with external data, may enable re-identification.

To reduce privacy risks, we use binary face-presence indicators as weak supervision, automatically extracted via RetinaFace (Deng et al., 2020), which handles small, occluded, or low-resolution faces. When multiple faces are detected, we average their embeddings to guide the projection and separate identity-related from task-relevant features. We also create 50 face videos from the Georgia Tech Face Database (of Computing, 2000), using their average embeddings as controlled sensitive signals.

**Face-guided suppression.** Both the original video frames and detected / generated face crops are passed through the same encoder (*e.g.*, I3D). This ensures that the embeddings $\boldsymbol{f}$ (from video) and $\boldsymbol{f}_{\text{face}}$ (from cropped faces) reside in the same latent space, allowing us to compare them meaningfully. Since both embeddings are high-level features, our method operates entirely at the feature level rather than in the pixel space. This avoids the need for explicit face masks or boundary annotations, and makes the projection robust to partial occlusions, variable face sizes, and fuzzy edges commonly seen in surveillance videos. The goal is to identify and remove the facial component from $\boldsymbol{f}$ using projection operator (Equation 3). Instead of adversarial training or an explicit sensitive attribute classifier, G-OPL adopts a direct geometric approach. We guide the projection basis $\boldsymbol{Q}$ to span directions aligned with facial identity by penalizing cosine similarity between the face embedding and the sensitive component of $\boldsymbol{f}$:

$$\mathcal{L}_{\text{task}} = \mathcal{L}_{\text{ori}} + \lambda_{\text{face}} \left( 1 - \cos\left(\boldsymbol{f}_{\text{face}}, \boldsymbol{Q}\boldsymbol{Q}^\top\boldsymbol{f}\right) \right), \tag{4}$$

where $\mathcal{L}_{\text{ori}}$ is the standard anomaly detection loss, and $\lambda_{\text{face}}$ controls the strength of the privacy-driven guidance. This alignment term encourages $\boldsymbol{Q}$ to capture face-related directions, which the projection operator then removes (Equation 2). The loss is activated only when a face is detected, ensuring efficient and targeted updates.

G-OPL combines the interpretability of projection-based methods with the flexibility of weak supervision. Unlike prior approaches that rely on encoder-decoder architectures or adversarial training, our method operates directly on intermediate features. It provides a principled, scalable, and deployment-friendly solution for privacy-aware anomaly detection, while improving robustness by removing irrelevant and potentially harmful signals.

**Orthogonality regularization.** Although QR ensures orthonormality per forward pass, gradient updates can erode it over time. To preserve orthogonality and subspace separation, we include:

$$\mathcal{L}_{\text{orth}} = \left\| \boldsymbol{Q}^\top \boldsymbol{Q} - \boldsymbol{I}_k \right\|_F^2, \tag{5}$$

where $\|\cdot\|_F$ is the Frobenius norm. The overall loss becomes:

$$\mathcal{L}_{\text{total}} = \mathcal{L}_{\text{task}} + \lambda_{\text{orth}} \mathcal{L}_{\text{orth}}, \tag{6}$$

with $\lambda_{\text{orth}}$ controlling regularization strength. This is particularly important when stacking multiple projection layers or operating in high-dimensional spaces.

In the Appendix, we provide the *algorithmic implementation of OPL and G-OPL* and discuss *practical considerations for their placement within the network*.

**Efficiency at inference.** At test time, G-OPL behaves like OPL. The projection matrix $\boldsymbol{Q}$ is fixed, and no face detection or additional input is required. This ensures privacy-aware behavior with negligible extra runtime cost, critical for real-time or resource-constrained applications.

### 3.3 PRIVACY-AWARE EVALUATION METRICS

We propose three privacy-aware metrics to measure how well our projection layers suppress sensitive data while retaining task-relevant signals.

**Sensitive Subspace Capture (SSC)** measures the degree to which the learned projection subspace captures sensitive information (*e.g.*, faces). Given an orthonormal basis $\boldsymbol{Q} \in \mathbb{R}^{d \times k}$ that spans the sensitive subspace learned via G-OPL, and a batch of face embeddings $\boldsymbol{f}_{\text{face}}^{(i)} \in \mathbb{R}^d$, we compute the projection of each embedding onto the subspace: $\hat{\boldsymbol{f}}^{(i)} = \boldsymbol{Q}\boldsymbol{Q}^\top \boldsymbol{f}_{\text{face}}^{(i)}$. The SSC score is defined as the average cosine similarity between the original embedding and its projected component:

$$\text{SSC} = \cos\left(\hat{\boldsymbol{f}}^{(i)}, \boldsymbol{f}_{\text{face}}^{(i)}\right). \tag{7}$$

Intuitively, a higher SSC indicates that a larger portion of the original features lies within the sensitive subspace, implying that $\boldsymbol{Q}$ effectively isolates directions strongly associated with sensitive attributes. This makes SSC a valuable diagnostic tool for assessing whether the learned subspace aligns with privacy-relevant components in the data.

**Anomaly Retention Distance (ARD).** To evaluate how well the projection preserves the performance, we compare the *frame-level* anomaly score distributions before and after applying the projection layer. Let $\boldsymbol{y}_{\text{raw}}$ and $\boldsymbol{y}_{\text{proj}}$ denote the anomaly scores computed from the raw features and the projected features, respectively. We treat $\boldsymbol{y}_{\text{raw}}$ and $\boldsymbol{y}_{\text{proj}}$ as samples from underlying distributions and estimate their densities, denoted by $P_{\text{raw}}(y)$ and $P_{\text{proj}}(y)$, using kernel density estimation or histograms. The ARD is then defined as the Kullback-Leibler (KL) divergence between the two:

$$\text{ARD} = \text{KL}\left(P_{\text{raw}}(y) \,\|\, P_{\text{proj}}(y)\right). \tag{8}$$

A lower ARD value indicates that the projection has minimal impact on anomaly score distribution, *i.e.*, the detection behavior remains consistent, suggesting that task-relevant information has been retained. Conversely, a high ARD implies that the projection has distorted the decision landscape, potentially removing useful cues along with sensitive ones. Thus, ARD serves as a quantitative measure of utility preservation in the presence of privacy-aware projections.

**Privacy Decay (PD) & First-layer PD (FPD).** To measure how effectively sensitive information is suppressed throughout the network, we introduce PD. At each layer / block $l$ containing (or following) a projection module (*e.g.*, G-OPL), we extract intermediate feature $\boldsymbol{f}^{(l)}$ and train a lightweight

| Method | Assault AUC | AP | Explosion AUC | AP | Fighting AUC | AP | Fire AUC | AP | Obj. Fall AUC | AP | People Fall AUC | AP | Robbery AUC | AP | Shooting AUC | AP | Traffic Acc. AUC | AP | Vandalism AUC | AP | Water Inc. AUC | AP | Overall AUC | AP |
|---|---|---|---|---|---|---|---|---|---|---|---|---|---|---|---|---|---|---|---|---|---|---|---|---|
| RTFM (I3D) | 53.9 | 66.4 | 66.0 | 76.6 | 79.8 | 88.6 | 44.9 | 71.1 | 84.6 | 89.3 | 45.7 | 52.6 | 70.2 | 88.0 | 87.5 | 89.2 | 64.1 | 57.7 | 74.9 | 73.0 | 98.1 | 99.6 | 86.6 | 68.4 |
| MGFN (SwinT) | 50.2 | 49.6 | 50.9 | 58.1 | 57.2 | 67.1 | 51.4 | 74.2 | 41.3 | 51.6 | 44.4 | 40.3 | 40.1 | 68.5 | 51.4 | 63.9 | 50.4 | 42.3 | 42.6 | 40.9 | 58.6 | 87.2 | 69.3 | 33.6 |
| MGFN (I3D) | 53.9 | 60.2 | 59.1 | 66.5 | 80.6 | 89.5 | 66.1 | 82.9 | 89.9 | 94.6 | 53.6 | 44.9 | 72.2 | 85.4 | 68.3 | 80.6 | 66.9 | 54.7 | 84.4 | 78.5 | 81.9 | 96.1 | 81.2 | 59.3 |
| UR-DMU | 56.9 | 64.5 | 67.9 | 74.5 | 83.9 | 90.4 | 61.2 | 82.9 | 92.1 | 95.8 | 42.5 | 43.7 | 63.5 | 79.3 | 81.4 | 87.8 | 62.0 | 55.6 | 84.7 | 77.0 | 98.5 | 99.5 | 85.0 | 68.3 |
| EGO | 52.2 | 57.5 | 57.6 | 74.4 | 66.5 | 72.8 | 62.9 | 86.7 | 92.3 | 94.8 | 35.4 | 43.8 | 64.8 | 87.5 | 68.6 | 78.4 | 69.9 | 64.3 | 88.1 | 81.4 | 81.9 | 95.4 | 87.3 | 64.4 |
| IEF-VAD | 66.0 | - | 66.3 | - | 79.8 | - | 49.4 | - | 75.9 | - | 42.5 | - | 66.9 | - | 86.9 | - | 70.1 | - | 75.8 | - | 88.9 | - | 82.1 | - |
| RTFM-OPL (I3D) | 57.0 | 62.4 | 77.7 | 85.7 | 74.1 | 84.8 | 49.6 | 75.5 | 87.7 | 92.1 | 53.3 | 50.4 | 72.4 | 89.0 | 84.1 | 89.5 | 69.5 | 58.7 | 84.8 | 80.9 | 99.2 | 99.8 | 86.5 | 68.2 |
| MGFN-OPL (SwinT) | 59.1 | 56.5 | 52.7 | 57.0 | 44.3 | 55.2 | 63.0 | 76.4 | 58.3 | 59.3 | 40.6 | 36.0 | 49.5 | 70.7 | 55.3 | 62.1 | 49.1 | 39.6 | 60.8 | 53.7 | 44.4 | 78.9 | 78.2 | 47.5 |
| MGFN-OPL (I3D) | 71.3 | 69.4 | 61.8 | 73.0 | 87.8 | 92.8 | 81.0 | 93.0 | 94.3 | 96.5 | 45.9 | 45.0 | 65.1 | 81.1 | 82.7 | 89.1 | 64.2 | 55.2 | 90.8 | 86.4 | 68.7 | 92.0 | 86.2 | 68.3 |
| RTFM-G-OPL/OPL (I3D) | 50.2 | 62.4 | 69.4 | 80.6 | 69.5 | 84.4 | 71.8 | 87.0 | 88.7 | 92.4 | 52.3 | 53.3 | 71.4 | 88.2 | 87.8 | 91.0 | 62.5 | 54.7 | 82.0 | 79.6 | 97.5 | 99.4 | 88.0 | 70.9 |
| MGFN-G-OPL/OPL (I3D) | 52.4 | 59.8 | 66.5 | 76.8 | 88.8 | 92.2 | 77.2 | 89.0 | 90.5 | 95.1 | 45.9 | 42.8 | 65.4 | 80.1 | 71.9 | 81.8 | 53.9 | 46.4 | 83.1 | 75.1 | 81.5 | 96.0 | 84.0 | 65.8 |

Table 1: **Performance by anomaly type on MSAD.** We compare against recent methods (same for Table 2): RTFM (Tian et al., 2021), MGFN (Chen et al., 2023b), UR-DMU (Zhou et al., 2023), EGO (Ding et al., 2025), and IEF-VAD (Jeong et al., 2025). **Bold** marks the best, underlined the second-best. Our **G-OPL/OPL** achieves competitive or superior results across anomaly types, highlighting its robustness, especially when paired with strong base models (*e.g.*, RTFM with I3D).

| Method | Frontdoor AUC | AP | Mall AUC | AP | Office AUC | AP | Parkinglot AUC | AP | Pedestr. st. AUC | AP | Restaurant AUC | AP | Road AUC | AP | Shop AUC | AP | Sidewalk AUC | AP | St. highview AUC | AP | Train AUC | AP | Warehouse AUC | AP | Overall AUC | AP |
|---|---|---|---|---|---|---|---|---|---|---|---|---|---|---|---|---|---|---|---|---|---|---|---|---|---|---|
| RTFM (I3D) | 81.8 | 79.3 | 88.1 | 76.6 | 76.8 | 72.8 | 80.7 | 45.8 | 94.0 | 48.5 | 88.3 | 79.1 | 84.3 | 57.9 | 85.3 | 75.6 | 88.3 | 68.8 | 72.0 | 28.5 | 51.4 | 3.3 | 82.7 | 57.0 | 86.6 | 68.4 |
| MGFN (SwinT) | 59.5 | 51.7 | 18.5 | 20.1 | 64.1 | 52.3 | 67.9 | 19.0 | 75.9 | 9.7 | 67.9 | 44.0 | 70.6 | 26.3 | 62.7 | 43.0 | 69.0 | 25.9 | 75.3 | 23.3 | 65.4 | 5.2 | 70.1 | 30.1 | 69.3 | 33.6 |
| MGFN (I3D) | 82.5 | 80.8 | 73.8 | 71.3 | 71.5 | 58.2 | 68.9 | 14.8 | 94.8 | 36.2 | 95.1 | 76.5 | 91.3 | 35.8 | 85.6 | 78.4 | 78.5 | 57.2 | 77.9 | 29.3 | 40.3 | 2.1 | 58.3 | 24.2 | 81.2 | 59.3 |
| UR-DMU | 84.8 | 82.8 | 91.0 | 83.8 | 77.8 | 67.3 | 91.4 | 53.9 | 81.9 | 11.5 | 93.1 | 87.4 | 83.0 | 64.4 | 81.3 | 64.5 | 86.5 | 64.1 | 85.0 | 37.7 | 59.0 | 3.1 | 81.2 | 59.1 | 85.0 | 68.3 |
| EGO | 85.2 | 81.6 | 82.3 | 73.4 | 80.0 | 71.7 | 96.8 | 75.2 | 97.5 | 52.0 | 94.3 | 73.9 | 89.8 | 64.6 | 83.4 | 72.2 | 87.1 | 45.0 | 28.2 | 10.1 | 80.8 | 7.8 | 84.7 | 46.6 | 87.3 | 64.4 |
| IEF-VAD | - | - | - | - | - | - | - | - | - | - | - | - | - | - | - | - | - | - | - | - | - | - | - | - | 82.1 | - |
| RTFM-OPL (I3D) | 85.6 | 82.3 | 85.6 | 80.2 | 77.2 | 72.0 | 76.9 | 26.4 | 96.6 | 50.5 | 90.2 | 81.3 | 76.9 | 53.3 | 88.6 | 82.8 | 84.9 | 56.5 | 66.8 | 26.7 | 42.4 | 2.3 | 86.1 | 66.8 | 86.5 | 68.2 |
| MGFN-OPL (SwinT) | 68.5 | 57.8 | 89.0 | 61.8 | 68.4 | 53.4 | 79.4 | 39.0 | 74.5 | 5.0 | 51.6 | 36.1 | 67.3 | 28.1 | 77.1 | 60.3 | 81.1 | 41.9 | 87.1 | 45.8 | 83.5 | 11.9 | 83.8 | 52.4 | 78.2 | 47.5 |
| MGFN-OPL (I3D) | 84.4 | 84.1 | 80.2 | 74.7 | 74.7 | 65.0 | 87.0 | 30.9 | 93.5 | 53.1 | 91.2 | 87.6 | 80.0 | 55.7 | 82.1 | 69.4 | 86.8 | 63.8 | 98.1 | 95.1 | 70.8 | 9.1 | 89.9 | 76.1 | 86.2 | 68.3 |
| RTFM-G-OPL/OPL (I3D) | 82.0 | 79.3 | 91.0 | 81.4 | 74.3 | 72.0 | 79.4 | 27.2 | 86.9 | 36.1 | 90.3 | 81.4 | 72.4 | 46.7 | 89.0 | 82.5 | 87.0 | 65.1 | 84.9 | 37.8 | 70.4 | 12.0 | 86.3 | 79.6 | 88.0 | 70.9 |
| MGFN-G-OPL/OPL (I3D) | 84.4 | 83.3 | 90.0 | 84.8 | 75.9 | 62.3 | 70.4 | 16.9 | 90.5 | 25.8 | 95.7 | 90.2 | 71.4 | 43.1 | 79.7 | 64.5 | 83.8 | 63.3 | 87.7 | 41.3 | 44.7 | 2.3 | 64.8 | 41.1 | 84.0 | 65.8 |

Table 2: **Performance by scenario on MSAD.** Results on 12 test scenarios (excluding Highway and Park without anomalies) show our method's strong adaptability and robustness, consistently outperforming or matching top baselines and recent state-of-the-art methods while achieving better balance across scenarios (this table) and anomaly types (Table 1).

classifier (*e.g.*, a linear SVM) to predict sensitive attributes (*e.g.*, face presence). The classification accuracy at layer $l$ is denoted as $\text{Acc}^{(l)}$, and PD is defined as:

$$\text{PD} = \left\{ \left( l, \text{Acc}^{(l)} \right) \right\}_{l=1}^{L}, \tag{9}$$

where $L$ is the number of probed layers. Plotting $\text{Acc}^{(l)}$ over $l$ gives a PD curve, showing how sensitive information decays through network; sharper drops indicate stronger protection by G-OPL.

We further define *First-layer Privacy Decay (FPD)* as $\text{Acc}^{(1)}$, which captures the face presence accuracy at the first G-OPL. FPD quantifies the immediate suppression of sensitive information at the network's entry point. A low FPD reflects strong initial privacy preservation, minimizing the risk of early-stage leakage and offering a clear diagnostic for the projection layer's effectiveness.

> These metrics form a robust framework for evaluating privacy-aware anomaly detection. SSC diagnoses whether sensitive features are being captured and encoded consistently; ARD quantifies utility preservation; and PD / FPD shows how privacy is suppressed across the network.

## 4 EXPERIMENT

### 4.1 EXPERIMENTAL SETUP

**Datasets & protocols.** We evaluate on five VAD datasets: ShanghaiTech (ShT) (Luo et al., 2017), UCF-Crime (UCF) (Sultani et al., 2018b), CUHK Avenue (CUHK) (Lu et al., 2013), UCSD Ped2 (Ped2) (Vijay et al., 2010), and MSAD (Zhu et al., 2024). For MSAD, we use Protocol ii. For all datasets, we follow standard evaluation protocols and report frame-level AUC and Average Precision (AP). Beyond accuracy, we use privacy-aware metrics to quantify how well sensitive attributes are suppressed, providing an ethical perspective on model performance.

To compare our metrics with a standard privacy evaluation, we adopt a classifier-based probe following the cMAP paradigm (Ravfogel et al., 2020). Using a pre-trained ArcFace model, we extract facial embeddings and test whether projected features $\hat{f}^{(i)}$ retain identity cues by learning a linear

| Method | ShanghaiTech | | | | | | UCF-Crime | | | | | | CUHK Avenue | | | | | | UCSD Ped2 | | | | |
|---|---|---|---|---|---|---|---|---|---|---|---|---|---|---|---|---|---|---|---|---|---|---|---|
| | AUC | AP | SSC↑ | ARD↓ | FPD↓ | Arc↓ | AUC | AP | SSC↑ | ARD↓ | FPD↓ | Arc↓ | AUC | AP | SSC↑ | ARD↓ | FPD↓ | Arc↓ | AUC | AP | SSC↑ | ARD↓ | FPD↓ |
| USTN-DSC (Yang et al., 2023) | 73.8 | - | - | - | - | - | - | - | - | - | - | - | 89.9 | - | - | - | - | - | 98.1 | - | - | - | - |
| FPDM (Yan et al., 2023) | 78.6 | - | - | - | - | - | 74.7 | - | - | - | - | - | 90.1 | - | - | - | - | - | - | - | - | - | - |
| HSC (Sun & Gong, 2023) | 83.4 | - | - | - | - | - | - | - | - | - | - | - | 93.7 | - | - | - | - | - | 98.1 | - | - | - | - |
| TEVAD (Chen et al., 2023a) | 98.1 | - | - | - | - | - | 84.9 | - | - | - | - | - | - | - | - | - | - | - | 98.7 | - | - | - | - |
| PEL4VAD (Pu et al., 2024) | 98.1 | 72.6 | - | - | - | - | 86.8 | 34.0 | - | - | - | - | - | - | - | - | - | - | - | - | - | - | - |
| VadCLIP (Wu et al., 2024) | - | - | - | - | - | - | 88.0 | - | - | - | - | - | - | - | - | - | - | - | - | - | - | - | - |
| EGO (Ding et al., 2025) | 97.3 | - | - | - | - | - | 81.7 | - | - | - | - | - | 83.1 | - | - | - | - | - | 93.2 | - | - | - | - |
| TeD-SPAD (Fioresi et al., 2023) | 90.6 | - | - | - | - | - | 74.8 | - | - | - | - | - | - | - | - | - | - | - | - | - | - | - | - |
| SPAct (Dave et al., 2022) | 87.7 | - | - | - | - | - | 73.9 | - | - | - | - | - | - | - | - | - | - | - | - | - | - | - | - |
| MGNAD (Park et al., 2020) | 70.5 | - | - | - | - | - | - | - | - | - | - | - | 88.5 | - | - | - | - | - | 97.0 | - | - | - | - |
| MGFN (Chen et al., 2023b) | 75.3 | 22.7 | - | - | 0.98 | 0.14 | 77.0 | 13.0 | - | - | 0.79 | 0.05 | 67.3 | 37.7 | - | - | 0.94 | 0.44 | 86.8 | 76.7 | - | - | 0.91 |
| MGFN-**G−OPL/OPL** | **83.7** | **42.0** | 0.10 | 0.23 | 0.96 | **0.01** | **83.3** | **15.2** | **0.99** | 0.07 | 0.49 | 0.02 | **70.8** | **40.5** | 0.39 | **18.0** | 0.58 | 0.13 | **93.9** | **93.6** | 0.39 | 0.94 | **0.31** |
| MGFN-**G−OPL/OPL**† | **89.5** | 41.9 | **0.52** | **0.16** | **0.68** | **0.01** | 80.9 | 14.8 | 0.72 | 6.46 | **0.01** | 0.03 | 69.1 | **43.5** | **0.81** | 23.6 | 1.0 | 0.31 | NA | NA | NA | NA | NA |
| RTFM (Tian et al., 2021) | 96.8 | 71.6 | - | - | 0.72 | 0.99 | 74.3 | 20.1 | - | - | 0.68 | 1.0 | 83.3 | 66.3 | - | - | 1.0 | 1.0 | 85.6 | 82.0 | - | - | 0.31 |
| RTFM-**G−OPL/OPL** | **97.3** | **74.7** | **0.97** | 0.29 | 0.19 | **0.01** | **78.3** | **30.9** | **0.98** | **0.03** | 0.23 | 0.03 | **84.9** | 66.2 | **0.89** | 0.26 | **0.0** | **0.31** | **89.6** | 75.0 | **0.98** | 0.06 | **0.29** |
| RTFM-**G−OPL/OPL**† | 97.2 | 74.6 | 0.20 | **0.17** | **0.01** | 0.02 | 74.1 | **30.1** | 0.33 | 0.30 | 0.64 | **0.02** | 83.9 | 65.7 | 0.11 | 3.86 | **0.0** | **0.31** | NA | NA | NA | NA | NA |

Table 3: **Results on ShanghaiTech, UCF-Crime, CUHK Avenue, and UCSD Ped2.** `G−OPL/OPL` are further evaluated using privacy metrics (SSC↑, ARD↓, FPD↓). † uses RetinaFace embeddings; NA denotes undetectable faces on UCSD Ped2 due to low resolution. **Bold** highlights performance surpassing the baseline or the best privacy variant. Privacy metrics may underperform on datasets with low resolution or small/occluded faces. We also include ArcFace-based identity retrieval as a classifier-based privacy probe (Arc↓), providing an external reference point for comparison. MG-NAD and SPAct are adapted to VAD; TeD-SPAD follows its own setup using MGFN+SwinT. This is *the first comprehensive privacy analysis* for VAD across datasets.

mapping back to the embedding space. We report rank-1 retrieval accuracy (lower is better), simulating an attacker attempting to recover identity from latent features. As shown in Table 3, our method reduces identity leakage across both our proposed metrics (SSC/ARD/FPD) and the ArcFace probe, confirming its robustness and generality.

**Models & setups.** We integrate our OPL and G-OPL into two complementary, state-of-the-art weakly supervised VAD models: RTFM (Tian et al., 2021) and MGFN (Chen et al., 2023b). RTFM is a widely used, lightweight baseline that supports systematic ablation and interpretability. In contrast, MGFN is a more so-

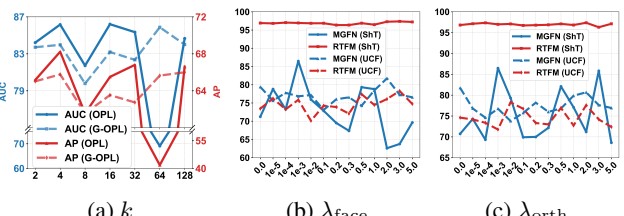

(a) $k$     (b) $\lambda_{\text{face}}$     (c) $\lambda_{\text{orth}}$

Figure 3: Evaluation of key hyperparameters.

phisticated model incorporating attention mechanisms and contrastive learning, making it ideal for evaluating generalization to complex architectures. Both models use pre-trained models, *e.g.*, I3D (Carreira & Zisserman, 2017), Swin Transformer (SwinT) (Liu et al., 2022), making them directly compatible with our plug-and-play projection modules. For OPL/G-OPL layer placement, we follow the *practical guidelines* detailed in the Appendix. We place G-OPL layers immediately after feature extractors (*e.g.*, I3D or SwinT) to suppress early-stage privacy-sensitive cues like faces, guided by weak face embeddings. OPL layers are inserted at deeper stages to filter task-irrelevant nuisances. For MSAD, as all videos have already been face-blurred for privacy protection, we directly use the official pre-extracted I3D and SwinT features provided by the dataset for our experiments.

## 4.2 QUANTITATIVE AND QUALITATIVE EVALUATION

We evaluate our method comprehensively and summarize the key findings and insights below.

**Sensitivity to $k$, $\lambda_{\text{face}}$, and $\lambda_{\text{orth}}$.** As shown in Fig. 3a, performance improves as $k$ increases from 2 to 16, peaking around $k = 4$, before degrading at $k = 64$. This suggests that small to moderate subspaces are sufficient to isolate nuisance factors, while overly large ones begin erasing task-relevant information. For both $\lambda_{\text{face}}$ and $\lambda_{\text{orth}}$ (Fig. 3b and 3c), we observe a stable performance window in the range $[10^{-4}, 10^{-2}]$. Within this range, semantic alignment and subspace decorrelation enhance disentanglement without impeding feature learning. However, large values hurt performance, likely due to over-penalization of useful variations or instability. Notably, ShanghaiTech remains relatively robust across a broad range, while UCF with MGFN, shows higher sensitivity, highlighting the greater importance of guided suppression in unstructured environments.

**Layer placement and frequency.** As shown in Fig. 4a, applying a single early-stage G-OPL (*e.g.*, G1O0) consistently delivers strong and stable performance, achieving top or near-top results on ShT

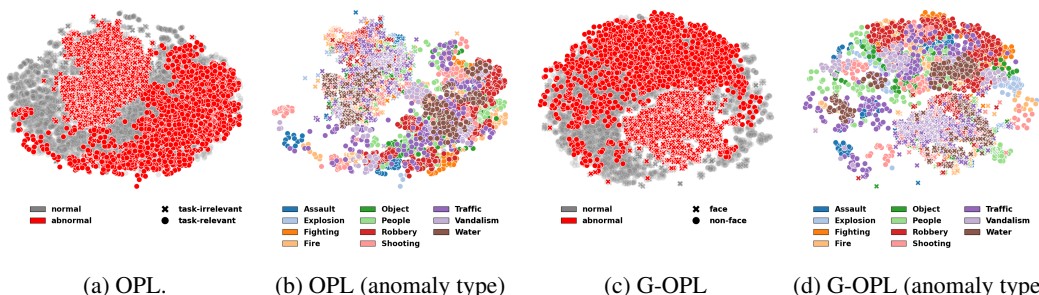

| (a) OPL. | (b) OPL (anomaly type) | (c) G-OPL | (d) G-OPL (anomaly type) |

Figure 5: UMAP plots visualize task-relevant (dots) and removed nuisance/sensitive (crosses) features after the first OPL/G-OPL (using RTFM-I3D on MSAD). Colors indicate *frame-level* labels (normal, abnormal, anomaly types). Removal operates at the feature level, anomalies are detected from the remaining task-relevant features, not from removed components. (a) *vs.* (c): Both OPL and G-OPL successfully disentangle nuisance/sensitive features (crosses) from task-relevant ones (dots), but G-OPL yields more compact clusters of removed features due to its guidance from facial cues, offering clearer separation of irrelevant information. (b) *vs.* (d): For task-relevant features, G-OPL preserves more compact and semantically meaningful clusters for *human-centric* anomalies (*e.g.*, *robbery*, *vandalism*), revealing improved disentanglement of anomaly-relevant factors.

and CUHK. In contrast, progressively adding more OPL layers (G1O1 through G1O5) often leads to diminishing returns or even degraded performance, particularly on UCF. This supports our hypothesis that excessive disentanglement disrupts the retention of subtle anomaly cues, emphasizing the importance of strategic rather than frequent placement of these modules. Introducing G-OPL at deeper layers (G2-G6) maintains competitive results on ShT but shows less stability on UCF and Ped2. Notably, configurations such as G5O0 achieve the highest AUC on Ped2, likely because this dataset contains low-resolution faces that are difficult to detect and thus better handled through late-stage disentanglement. These results validate our design principle: early-stage G-OPL provides the most reliable gains, while additional OPL layers should be used cautiously to avoid over-filtering. Later G-OPL layers may offer dataset-specific benefits, but overcomplicating the placement does not yield universal improvements.

**Effectiveness on MSAD across anomaly types and scenarios.** Our improvements are evident across both anomaly types (*e.g.*, accidents *vs.* crimes) as shown in Table 1, and across diverse scenarios in Table 2, highlighting the robustness and adaptability of our approach. Notably, our method excels in challenging categories such as Fire and Vandalism, where subtle or occluded cues often hinder detection. This underscores the effectiveness of our

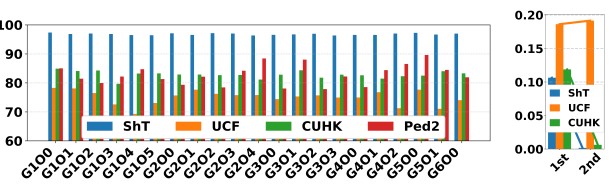

| (a) Evaluation of projection layer placement. | (b) PD. |

Figure 4: (a) Ablation on OPL/G-OPL placement across datasets with RTFM. G$m$O$n$: $m$ G-OPL and $n$ OPL layers. (b) We select model with more G-OPL layers that achieves reasonable VAD performance for privacy decay (PD) curve.

disentanglement mechanism in filtering out task-irrelevant information while preserving critical anomaly-related signals. Furthermore, the performance gains are particularly pronounced when integrating `OPL` with stronger backbones like I3D, confirming the complementary nature of our design to existing architectures. While baselines such as EGO (Ding et al., 2025) exhibit robustness in certain scenarios through fusion strategies, they struggle with anomaly type diversity, where our method maintains balanced and consistently superior performance. Importantly, these findings suggest that our model not only mitigates the influence of noise and distractions but also enhances the model's ability to preserve causal and contextually relevant cues essential for accurate anomaly recognition (see also Fig. 5). This capability is particularly valuable in scenarios characterized by visual occlusions, complex dynamics, or subtle anomalies, where conventional fusion-based methods tend to underperform. This highlights the broader potential of our approach for advancing robust and generalizable video anomaly detection.

**Discussion on $QQ^\top$ visualization.** Fig. 6 visualizes the projection matrices $QQ^\top$ learned by G-OPL under both detected and generated faces. These matrices offer an interpretable window into

how G-OPL identifies and suppresses privacy-sensitive directions in feature space. When using detected faces as guidance, $QQ^\top$ exhibits more diverse and irregular patterns. This reflects the variability inherent in real-world surveillance footage, where faces differ in pose, scale, occlusion, and lighting. Despite this variability, G-OPL successfully learns to capture meaningful face-related directions, as evidenced by consistent localized energy regions in the projection matrices (Fig. 6a & 6c). These patterns confirm that even weak, noisy supervision suffices to guide the suppression of sensitive information. In contrast, when guided by clean, generated faces from a controlled dataset, $QQ^\top$ reveals more structured and concentrated patterns. This indicates that the learned subspace is more focused and coherent (Fig. 6b & 6d), effectively aligning with the dominant variations associated with facial identity in a more stable and controlled manner. This demonstrates G-OPL's adaptability: cleaner supervision leads to more precise subspace removal. Its flexibility allows effective privacy-aware disentanglement using diverse weak supervision, from noisy real-world data to curated facial signals. The resulting $QQ^\top$ matrices provide interpretable evidence, offering transparency into how sensitive information is removed from learned features.

**Privacy-aware metrics.** Across datasets containing human faces (Table 3), we observe reduced ARD and FPD (see PD curve in Fig. 4b) while SSC remains high, showing that sensitive components are removed without affecting anomaly-relevant semantics. This supports G-OPL's core idea: identity-related variations lie in a subspace orthogonal to anomaly-relevant signals and can be projected out without harming task utility. Using generated faces further improves suppression, achieving lower ARD and FPD than detected faces, demon-

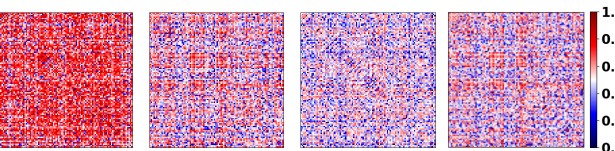

(a) ShT (*det.*)   (b) ShT (*gen.*)   (c) UCF (*det.*)   (d) UCF (*gen.*)

Figure 6: Visualization of $QQ^\top$ (G-OPL of RTFM model) from detected (*det.*) and generated (*gen.*) faces on ShanghaiTech (ShT) and UCF-Crime (UCF). Central $100 \times 100$ regions, where energy concentrates, are shown to better reveal informative subspace patterns. All matrices are log-scaled, globally min-max normalized. Differences reflect dataset- and signal-specific subspace structures, highlighting distinct patterns of sensitive feature disentanglement.

strating robustness under both synthetic and real privacy threats. These results provide the first comprehensive quantitative analysis of privacy leakage in VAD, highlighting G-OPL's practical impact. Importantly, our results show that the proposed metrics (SSC/ARD/FPD) correlate well with established ArcFace-based privacy probes. Notably, cases with low SSC and FPD also demonstrate low identity retrieval accuracy under ArcFace, confirming the consistency of these indicators. However, our metrics offer several advantages over classifier-based probe: they are more interpretable, do not require labeled identities, and are cheaper to compute, especially in the absence of clean face datasets. Thus, we advocate using SSC/ARD/FPD as diagnostic tools for privacy preservation, while ArcFace serves as a strong external attacker baseline.

**Trade-off analysis.** Our results show a nuanced balance between detection performance, privacy, and interpretability. Integrating G-OPL often improves anomaly detection metrics such as AUC and AP, likely because removing irrelevant identity components also eliminates spurious correlations that impair generalization. Minor fluctuations (*e.g.*, with detected faces) remain within acceptable bounds, confirming practical viability. Importantly, the projection matrix $QQ^\top$ provides interpretable evidence of suppressed privacy-sensitive directions, a feature largely absent in prior work. This interpretability bridges performance-driven models and explainable AI, offering both methodological clarity and ethical accountability. Overall, G-OPL provides *a principled framework* that protects privacy, maintains performance, and enhances transparency in anomaly detection.

## 5 CONCLUSION

We propose a novel VAD framework that integrates privacy and interpretability through the Orthogonal Projection Layer (OPL), which suppresses task-irrelevant variations via orthogonal subspace projection. To explicitly remove sensitive attributes such as faces, we extend OPL into the Guided OPL (G-OPL), using cosine alignment without relying on adversarial learning. We further introduce privacy-aware evaluation metrics to quantify privacy leakage in VAD. Our results demonstrate that privacy and interpretability can be effectively embedded into VAD architectures, enabling robust anomaly detection without exploiting sensitive information.

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

# A APPENDIX

## A.1 WHERE PRIVACY CAN BE PROTECTED?

Privacy preservation in machine learning spans multiple levels: from dataset design, to learning paradigms, to model-level architectural innovations. Each level introduces different mechanisms, assumptions, and trade-offs. In this section, we contextualize our work within this broader landscape and highlight how our approach complements and advances model-level privacy-preserving design.

**Dataset-level.** One intuitive approach to privacy preservation involves directly sanitizing the input data. This includes techniques such as face blurring, masking sensitive regions, downsampling, or removing identity-revealing cues. While effective in some cases, these methods can compromise data utility, especially in tasks requiring high-fidelity spatial or temporal information. Moreover, manual or heuristic filters often fail to generalize across domains and may not align well with downstream task requirements.

**Learning-level.** At the learning level, frameworks such as Federated Learning (FL) and Differential Privacy (DP) introduce algorithmic guarantees that limit exposure of raw data. FL decentralizes training by keeping data local to devices, while DP injects carefully calibrated noise into training or inference to bound the information leakage. Although theoretically rigorous, these methods can incur substantial performance or utility loss in practice. They also typically require significant changes to training protocols and infrastructure.

**Model-level: our contribution.** Our proposed framework operates at the model level by embedding privacy-aware mechanisms directly into the network architecture. Specifically, we introduce the OPL and its guided variant *G-OPL*, which learn to remove sensitive components from intermediate feature representations via low-rank, orthonormal projections. These modules act as semantic filters, discarding features that are predictive of protected attributes, such as facial identity or age, while retaining task-relevant signals.

This model-level intervention offers several advantages:

i. **Architecture-integrated ethics.** Privacy preservation is no longer an afterthought or external constraint, but a built-in property of the model's internal structure and training objective.

ii. **Representation-level control.** Unlike dataset filtering or global privacy budgets, OPL/G-OPL operate on structured neural representations where sensitive factors are often more linearly separable, allowing precise, learnable suppression.

iii. **Minimal trade-off with utility.** Our experiments show that anomaly detection performance is largely preserved or even improved, as OPL/G-OPL removes spurious or distracting features that hinder generalization.

iv. **Modularity and scalability.** These projection layers are lightweight, differentiable, and easily pluggable into existing models, requiring no special training pipeline or infrastructure.

**Toward ethical and interpretable AI systems.** By integrating ethical constraints directly into model design, we take a proactive stance in the development of responsible AI. Our approach can be seen as a form of architectural regularization that aligns representation learning with both performance and ethical objectives. Unlike post-hoc privacy filters or external audits, this internalized design philosophy enables continuous, scalable deployment of AI systems that are privacy-aware by construction.

## A.2 TERMINOLOGY AND CONCEPTUAL CLARIFICATIONS

To ensure clarity and accessibility, we define several core concepts used throughout this work.

**Task-relevant features.** These are features that directly contribute to solving the primary task, here, anomaly detection. For example, in surveillance video, task-relevant cues may include motion patterns, temporal consistency, or object trajectories that help distinguish normal from abnormal events. Retaining such features is essential for maintaining detection performance.

**Subspace projection.** We use orthogonal projection to separate and remove specific components (*e.g.*, sensitive information) from learned representations. The assumption is that such information resides in a linearly separable subspace, especially in intermediate neural representations, which are more structured and disentangled than raw inputs.

**Task-relevant residual subspace.** We define the subspace remaining after projection as the task-relevant (or task-aligned) residual subspace. This space is expected to retain features that are informative for downstream anomaly detection task. The proposed G-OPL/OPL learns a subspace that captures sensitive or task-irrelevant variations (*e.g.*, identity), and removes them by projecting features onto its orthogonal complement. This filtering process enhances both robustness and interpretability by preserving only information aligned with the core detection objective.

**Task-aware suppression.** This refers to the process of removing sensitive features while preserving information critical to the main task. Our G-OPL module achieves this by guiding the projection basis to align with sensitive attribute directions, such as facial identity, using a cosine similarity loss. This allows the model to explicitly capture and subsequently suppress privacy-relevant components from representation, without harming anomaly detection performance.

**Sensitive information.** Sensitive information refers to attributes that are irrelevant to the primary task but may raise ethical, legal, or social concerns if retained or exposed. In visual anomaly detection, this can include biometric cues such as facial identity, age, or gender. While these attributes are not necessary for detecting anomalies, they may be unintentionally encoded in intermediate features, potentially leading to privacy violations or biased outcomes.

**Utility-privacy trade-off.** This captures the balance between preserving performance on the primary task and minimizing leakage of sensitive information. A key contribution of this work is the introduction of metrics and architectural tools (OPL/G-OPL) that help navigate and optimize this trade-off.

**Privacy-aware anomaly detection.** Rather than relying on traditional privacy-preserving mechanisms (*e.g.*, encryption or differential privacy), our work focuses on suppressing sensitive content

in feature space. This approach aligns with responsible AI practices by reducing representational harms while maintaining utility.

### A.3 CONNECTIONS TO RELATED WORK

Our approach builds upon and unifies insights from several research threads, including subspace projection, adversarial representation learning, disentanglement, fairness, and invariant learning. Below, we contextualize our OPL and G-OPL by drawing connections to these bodies of work and clarifying the novel contributions our method offers.

**Video anomaly detection.** VAD has been extensively studied, evolving from early approaches based on handcrafted features and statistical models to deep learning-based methods that capture complex spatio-temporal patterns Luo et al. (2021); Liu et al. (2018); Sultani et al. (2018a). Recent models use 3D convolutions (*e.g.*, I3D Carreira & Zisserman (2017)) and memory-augmented architectures Gong et al. (2019) to model temporal dynamics and long-range dependencies in videos. While these methods achieve strong performance, they primarily optimize for detection accuracy, without considering the nature of the information being encoded, such as whether some features are irrelevant to the anomaly task or pose ethical concerns. Our work departs from this trend by explicitly shaping the learned representations to filter out both nuisance and privacy, sensitive features using differentiable projection modules.

**Subspace projection and Nullspace filtering.** Classical subspace projection techniques have long been used in signal processing and machine learning to isolate informative or invariant components of data. Recent approaches such as Nullspace Projection Ravfogel et al. (2020); Wang et al. (2021); Kim et al. (2019) and Iterative Nullspace Projection Ravfogel et al. (2020); Esmaeili et al. (2016) apply similar principles for fairness and debiasing, identifying directions that correlate with protected attributes and projecting them out.

Our OPL shares the goal of isolating and filtering out task-irrelevant or sensitive directions but differs in three key ways: (i) it learns these subspaces in a fully differentiable manner, integrated directly into the network; (ii) it uses orthogonal projection rather than hard nulling, allowing smooth removal of signals with controlled rank; and (iii) G-OPL extends this by incorporating guidance through a geometric alignment loss that directs suppression toward specific sensitive features (*e.g.*, facial identity).

**Adversarial disentanglement and privacy filtering.** Our G-OPL layer is related to adversarial representation learning methods used for fairness Madras et al. (2018); Elazar & Goldberg (2018) and privacy Li et al. (2021), which remove sensitive information by backpropagating gradients from attribute classifiers. However, unlike these methods, G-OPL does not rely on an explicit gradient reversal layer. Instead, it learns a low-rank subspace and projects features orthogonally to sensitive directions. This projection-based suppression provides a more interpretable and geometrically grounded approach compared to generic encoder-decoder or adversarial frameworks.

**Disentangled and compressed representations.** Disentangled representation learning methods, such as $\beta$-VAE Higgins et al. (2017) and FactorVAE Kim & Mnih (2018), aim to decompose features into statistically independent factors of variation.

OPL shares this spirit of structural separation, but with key differences: rather than disentangling latent variables through generative priors, we isolate linearly separable subspaces aligned with nuisance or sensitive attributes. Moreover, OPL is task-aware and explicitly supervised by downstream objectives, unlike unsupervised disentanglement which may not align with task relevance. OPL can also be viewed as a learnable bottleneck, comparable to information bottleneck layers Tishby et al. (2000); Alemi et al. (2016); Wu et al. (2020), but with a geometric focus on orthogonality and projection.

**Spurious correlation and invariant learning.** Efforts to suppress spurious correlations, such as domain adversarial training Ganin et al. (2016), IRM Arjovsky et al. (2019), and feature orthogonality Wang et al. (2022), seek to improve generalization by learning invariant representations.

G-OPL supports this goal by enforcing invariance to known nuisance attributes via guided projection. However, unlike IRM-based methods that require environment annotations, or methods that enforce global invariance, our approach performs localized, layerwise suppression that is lightweight

and adaptable. This makes it more scalable and compatible with standard vision architectures and datasets.

> Our proposed framework integrates principles from projection-based filtering, adversarial privacy learning, and disentangled representation modeling, while introducing key architectural and theoretical advances: (i) differentiable, learnable subspace projection; (ii) orthogonal filtering of sensitive information; and (iii) staged integration across layers. This yields a practical, interpretable, and privacy-aware anomaly detection system with improved utility-privacy trade-offs.

## A.4 THEORETICAL MOTIVATION: SENSITIVE ATTRIBUTES IN LINEARLY SEPARABLE SUBSPACES

In this section, we provide a theoretical basis for the common assumption that sensitive attributes, such as facial identity or gender, are approximately linearly separable in intermediate neural representations. While this is not universally guaranteed, it is supported by prevailing patterns in representation learning.

**Assumption 1** (Structured representation learning). *Let $x \in \mathbb{R}^n$ be an input* (e.g., *an image*), *and let $f : \mathbb{R}^n \to \mathbb{R}^d$ denote the neural network feature extractor. We assume that the intermediate representation $f = f(x)$ is* structured, *meaning that different semantic factors, such as task-relevant and sensitive attributes, are at least partially disentangled in feature space.*

*This assumption is supported by empirical findings that deep networks tend to organize information along semantically meaningful directions in their latent spaces Bengio et al. (2013).*

**Definition 1** (Linear separability in feature space). *A sensitive attribute $s \in \mathcal{S}$ is said to be* linearly separable *in the feature space $\mathbb{R}^d$ if there exists a weight matrix $W_s \in \mathbb{R}^{d \times k}$ (or a vector $w_s$ for binary classification) such that:*

$$s = \arg\max_i (W_s^\top f)_i, \tag{10}$$

*where $(\cdot)_i$ denotes the $i$th component. Equivalently, a linear classifier on $f$ can reliably predict $s$. This implies the existence of a sensitive subspace $\mathcal{S}_s = \mathrm{span}(W_s)$ capturing the most predictive directions for $s$.*

**Proposition 1** (Existence of sensitive subspace). *Let $\mathcal{D} = \{(x^{(i)}, s^{(i)})\}_{i=1}^N$ be a dataset with sensitive labels $s^{(i)}$. Suppose:*

  *i. The feature extractor $f$ is trained for a primary task unrelated to $s$.*

  *ii. The sensitive attribute $s$ is statistically correlated with the input $x$.*

  *iii. The mapping $f = f(x)$ is continuous and piecewise linear* (e.g., *due to ReLU activations*).

*Then, if a linear classifier $w_s$ achieves low prediction error on $s$, the projection of $f$ onto the subspace $\mathcal{S}_s = \mathrm{span}(w_s)$ preserves sensitive information with high fidelity.*

*Proof.* A linear classifier defines a hyperplane in feature space. Its associated projection:

$$f_s = \frac{w_s w_s^\top}{\|w_s\|^2} f \tag{11}$$

isolates the component of $f$ aligned with $s$. In the multiclass or multi-attribute case, a projection onto $\mathrm{span}(W_s)$ retains the informative directions.

By the Eckart-Young theorem Eckart & Young (1936), such projections provide optimal low-rank approximations for preserving predictive information under Frobenius norm constraints.

Thus, sensitive attributes can be geometrically modeled as linearly encoded subspaces within $\mathbb{R}^d$.

$\square$

**Empirical and theoretical justification.** Deep networks tend to disentangle factors of variation through hierarchical abstraction Bengio et al. (2013); Belghazi et al. (2018). Nonlinear entanglements in pixel space often become more linearly separable in intermediate features. This is further

---

**Algorithm 1** Orthogonal Projection Layer (OPL)

---

**Require:** Feature matrix $\boldsymbol{F} \in \mathbb{R}^{B \times d}$, learnable weight matrix $\boldsymbol{W} \in \mathbb{R}^{k \times d}$
**Ensure:** Projected features $\boldsymbol{F}_{\text{proj}}$, residual features $\boldsymbol{F}_{\text{res}}$
    *1. Compute orthonormal basis for nuisance subspace*
  1: $\boldsymbol{W}^\top = \boldsymbol{Q}\boldsymbol{R}$                            ▷ QR decomposition: $\boldsymbol{Q} \in \mathbb{R}^{d \times k}$
    *2. Construct projection matrix*
  2: $\boldsymbol{P} \leftarrow \boldsymbol{I}_d - \boldsymbol{Q}\boldsymbol{Q}^\top$                ▷ Project onto orthogonal complement
    *3. Project and compute residual*
  3: $\boldsymbol{F}_{\text{proj}} \leftarrow \boldsymbol{F} \cdot \boldsymbol{P}$
  4: $\boldsymbol{F}_{\text{res}} \leftarrow \boldsymbol{F} - \boldsymbol{F}_{\text{proj}}$
  5: **return** $\boldsymbol{F}_{\text{proj}}, \boldsymbol{F}_{\text{res}}$

---

**Algorithm 2** Training with Guided OPL (G-OPL)

---

**Require:** Feature batch $\boldsymbol{F} \in \mathbb{R}^{B \times d}$, face embedding batch $\boldsymbol{F}_{\text{face}} \in \mathbb{R}^{B \times d}$, face presence mask $\boldsymbol{m} \in \{0, 1\}^B$
**Require:** Learnable projection matrix basis $\boldsymbol{Q} \in \mathbb{R}^{d \times k}$
**Require:** Loss $\mathcal{L}_{\text{ori}}$, weights $\lambda_{\text{face}}, \lambda_{\text{orth}}$
**Ensure:** Total loss $\mathcal{L}_{\text{total}}$
    *1. Construct projection matrix*
  1: $\boldsymbol{P} \leftarrow \boldsymbol{I}_d - \boldsymbol{Q}\boldsymbol{Q}^\top$
    *2. Project features*
  2: $\boldsymbol{F}_{\text{proj}} \leftarrow \boldsymbol{F} \cdot \boldsymbol{P}$
  3: $\boldsymbol{F}_{\text{sens}} \leftarrow \boldsymbol{F} - \boldsymbol{F}_{\text{proj}}$
    *3. Compute cosine similarity loss (only if face present)*
  4: **for** $i = 1$ to $B$ **do**
  5:     **if** $\boldsymbol{m}[i] = 1$ **then**
  6:         $\mathcal{L}_{\text{face}}^{(i)} \leftarrow 1 - \cos(\boldsymbol{F}_{\text{face}}[i], \boldsymbol{F}_{\text{sens}}[i])$
  7:     **else**
  8:         $\mathcal{L}_{\text{face}}^{(i)} \leftarrow 0$
  9: $\mathcal{L}_{\text{face}} \leftarrow \frac{1}{\sum \boldsymbol{m}} \sum_{i=1}^{B} \mathcal{L}_{\text{face}}^{(i)}$          ▷ Mean over face-present samples
    *4. Orthogonality regularization*
  10: $\mathcal{L}_{\text{orth}} \leftarrow \left\| \boldsymbol{Q}^\top \boldsymbol{Q} - \boldsymbol{I}_k \right\|_F^2$
    *5. Final loss*
  11: $\mathcal{L}_{\text{task}} \leftarrow \mathcal{L}_{\text{ori}} + \lambda_{\text{face}} \cdot \mathcal{L}_{\text{face}}$
  12: $\mathcal{L}_{\text{total}} \leftarrow \mathcal{L}_{\text{task}} + \lambda_{\text{orth}} \cdot \mathcal{L}_{\text{orth}}$
  13: **return** $\mathcal{L}_{\text{total}}$

---

supported by linear probing studies, which show that simple classifiers can extract semantic information from latent representations Alain & Bengio (2016).

> Although exact linear separability is not guaranteed, it is theoretically and empirically reasonable to model sensitive attributes as occupying identifiable subspaces in neural feature spaces. This motivates our use of orthogonal projection to remove sensitive components while preserving task-relevant information, enabling interpretable, efficient, and ethically aligned representations.

### A.5 ALGORITHMIC IMPLEMENTATION OF OPL AND G-OPL

In this section, we present our Orthogonal Projection Layer (OPL) and its extension, the Guided Orthogonal Projection Layer (G-OPL), both designed to enhance video anomaly detection by removing nuisance and sensitive information from intermediate features. OPL dynamically learns to identify and project out task-irrelevant subspaces, thereby improving robustness and interpretability. Building on this, G-OPL uses weak supervision signals, such as face presence, to explicitly suppress privacy-sensitive components like facial identity through a guided projection mechanism. Both algorithms are detailed in Algorithm 1 and 2. Together, these modules form a modular, differentiable,

and effective framework that balances detection performance with privacy preservation in real-world surveillance applications.

## A.6 PRACTICAL CONSIDERATIONS FOR LAYER PLACEMENT

OPL and G-OPL are designed to be lightweight and modular, enabling flexible integration at various depths within standard VAD architectures. However, their effectiveness depends critically on where they are placed, necessitating careful consideration to balance detection accuracy, privacy preservation, and interpretability.

**Efficient implementation.** OPL and G-OPL are implemented using a simple, bias-free fully connected layer followed by QR decomposition to compute an orthonormal projection basis. This avoids matrix inversion or eigendecomposition, ensuring stable and efficient training. The absence of bias ensures the learned subspace is linear and passes through the origin, essential for valid orthogonal projection. The design is fully differentiable, GPU-efficient, and batch-parallel, introducing minimal overhead even in deep networks. Moreover, the low-rank nature of the projection reduces memory and computation, making these modules well-suited for real-time and resource-constrained applications such as robotics and surveillance.

**Feature-level projection.** Applying orthogonal projection directly to raw video inputs is impractical due to their high dimensionality and nonlinear entanglement of factors like pose, lighting, or occlusion. Such factors rarely lie in linear subspaces in pixel space. In contrast, intermediate features learned by neural networks tend to be semantically structured and more linearly separable. Sensitive attributes such as identity, age, or background context are more explicitly represented in these feature spaces. Projecting at this level allows meaningful and interpretable suppression of unwanted information. Thus, we apply OPL and G-OPL at intermediate feature levels, where subspace learning is more stable and effective.

**Why QR decomposition?** Traditional projection operators often take the form $W(W^\top W)^{-1}W^\top$, where $W$ is a basis matrix of the nuisance subspace. However, computing the inverse $(W^\top W)^{-1}$ is computationally expensive, numerically unstable, and unsuitable for gradient-based optimization when $W$ is ill-conditioned or nearly rank-deficient. In contrast, QR decomposition provides a robust and efficient way to obtain an orthonormal basis $Q$ without matrix inversion. This approach improves numerical stability, yields well-behaved gradients for end-to-end training, and scales gracefully to high-dimensional feature spaces typical of deep convolutional networks.

**Why not adversarial learning?** Traditional privacy-preserving methods often use adversarial learning with a gradient reversal layer (GRL) Ganin et al. (2016), requiring a sensitive classifier to predict protected attributes. However, these approaches are unstable and introduce additional complexity. In contrast, G-OPL is simpler and more interpretable. Because both $f$ and $f_{\text{face}}$ are extracted from the same backbone, they lie in a shared feature space. This enables a direct comparison without needing a classifier or adversarial setup. The cosine loss provides a stable and intuitive training signal, reducing sensitive leakage while preserving task-relevant features.

**Optimal layer placement.** Early network layers capture low-level visual cues (*e.g.*, texture, color, edges) that often correlate with sensitive attributes like facial identity or clothing style. Applying G-OPL at these stages enables targeted suppression of private information before it propagates deeper. The cosine similarity loss directly aligns the projection basis with sensitive features, allowing precise removal without adversarial gradient reversal. In contrast, deeper layers encode high-level semantic information such as object behavior, temporal dynamics, and scene context. At this stage, the standard OPL effectively removes residual nuisance factors like lighting variations, background clutter, or camera motion, improving anomaly localization and robustness by filtering out spurious correlations.

To systematically evaluate these effects, we experiment with multiple configurations involving single or stacked OPL and G-OPL layers at different depths. We analyze their impact on anomaly detection performance, privacy leakage, and interpretability. Our findings reveal key trade-offs and offer practical guidelines for ethically aligned and effective VAD deployment.

A.7 BEYOND A LINEAR LAYER: HOW OPL/G-OPL DIFFER FROM STANDARD FC LAYERS

Although our proposed OPL and its guided variant (G-OPL) use a learnable linear transformation internally, they fundamentally diverge from conventional fully connected (FC) layers in terms of objective, structure, and interpretability. Below, we clarify these differences to highlight the unique role played by OPL/G-OPL in enforcing task-aligned and privacy-aware representation learning.

**Objective: subspace projection *vs.* feature transformation.** A standard FC layer learns arbitrary affine transformations to map features from one space to another, typically followed by a nonlinearity. Its goal is to expand model capacity and enable feature recombination to support the end task (*e.g.*, classification). In contrast, OPL and G-OPL are explicitly designed to enforce geometric structure in the representation space by learning a low-dimensional, orthonormal subspace that captures specific semantic information, either task-irrelevant (OPL) or sensitive (G-OPL). Instead of merely transforming features, these layers act as *filters*, removing unwanted components via orthogonal projection.

**Structure: orthonormality and projection geometry.** At the heart of OPL is a bias-free linear layer whose weights are interpreted not as a transformation matrix, but as a subspace basis. Specifically, given weights $W \in \mathbb{R}^{d \times k}$, we compute its orthonormal basis $Q \in \mathbb{R}^{d \times k}$ via QR decomposition. This $Q$ defines a $k$-dimensional subspace within the feature space $\mathbb{R}^d$.

The projection matrix (Equation 2) is then applied to the input features $f$, yielding equation 3, which removes the component of $f$ lying in the learned subspace. This geometric operation is fundamentally different from the arbitrary affine transformation performed by an FC layer. The orthonormality constraint and residual-based design confer interpretability, mathematical soundness, and stability.

**Learning signal: supervised subspace removal.** A key difference between the two layers lies in their training signals. The OPL is trained implicitly through the downstream anomaly detection loss, learning to filter out nuisance directions irrelevant to the task. In contrast, G-OPL receives explicit supervision from a sensitive attribute detector by maximizing the cosine similarity between sensitive attribute embeddings and the projected feature components. This directs the projection subspace to capture sensitive information, enabling its suppression in the residual features. This targeted and interpretable learning dynamic contrasts with standard fully connected layers, which lack such a disentangling bias.

**Interpretability and stability.** The outputs of OPL/G-OPL decompose input into two parts: projection $f_{\text{proj}}$ (preserved, task-relevant) and residual $f - f_{\text{proj}}$ (removed, task-irrelevant or sensitive). This clear separation offers transparency into what model retains and discards at each stage, making internal representation more interpretable. Additionally, the orthogonality constraint stabilizes optimization by preventing degenerate subspaces or ill-conditioned weight matrices, an issue common in deep FC layers.

> Although OPL and G-OPL are implemented using bias-free linear layers, they are not conventional FC layers in disguise. Their subspace-based design, orthogonality constraints, and supervision-driven learning make them principled tools for structured, privacy-aware feature filtering. This distinction is essential for understanding their role in promoting robust, interpretable, and ethically aligned visual anomaly detection.

A.8 WHY STACK MULTIPLE OPL/G-OPL LAYERS?

Stacking multiple OPL/G-OPL offers a principled and effective strategy for progressively filtering out different forms of undesirable information, ranging from private attributes to task-irrelevant nuisances, across network depth. This design choice is not merely architectural convenience but reflects a structured approach to disentangling and regulating feature representations at multiple levels of abstraction.

**Progressive removal of structured information.** Neural networks encode features hierarchically: early layers capture fine-grained, low-level details (*e.g.*, texture, color), while deeper layers encode semantic, task-aligned abstractions (*e.g.*, motion patterns, object interactions). Sensitive or irrelevant information may appear at different depths and in different forms. A single projection may be

| Dataset | RTFM | | | | | | | | MGFN | | | | | | | |
|---|---|---|---|---|---|---|---|---|---|---|---|---|---|---|---|---|
| | G-OPL | OPL | $\lambda_{\text{orth}}$ | $\lambda_{\text{face}}$ | batch-size | max-epoch | AUC | AP | G-OPL | OPL | $\lambda_{\text{orth}}$ | $\lambda_{\text{face}}$ | batch-size | max-epoch | AUC | AP |
| MSAD-**G−OPL/OPL** | 1 | 0 | 1e−4 | 1.0 | 16 | 100 | 88.0 | 70.9 | 1 | 0 | 1e−2 | 1e−3 | 16 | 100 | 84.0 | 65.8 |
| ShT-**G−OPL/OPL** | 1 | 0 | 2.0 | 3.0 | 16 | 100 | 97.3 | 74.7 | 1 | 0 | 1e−3 | 0.3 | 16 | 100 | 83.7 | 42.0 |
| ShT-**G−OPL/OPL**† | 1 | 0 | 1 | 0 | 16 | 100 | 97.2 | 74.6 | 1 | 0 | 3.0 | 0.3 | 16 | 100 | 89.5 | 41.9 |
| UCF-**G−OPL/OPL** | 1 | 0 | 1e−2 | 3.0 | 16 | 100 | 78.3 | 30.9 | 1 | 0 | 0.1 | 0.1 | 16 | 100 | 83.3 | 15.2 |
| UCF-**G−OPL/OPL**† | 1 | 0 | 1e−2 | 3.0 | 16 | 100 | 74.1 | 30.1 | 1 | 0 | 0.1 | 0.1 | 16 | 100 | 80.9 | 14.8 |
| CUHK-**G−OPL/OPL** | 1 | 0 | 0.1 | 2.0 | 4 | 100 | 84.9 | 66.2 | 3 | 0 | 1e−5 | 1.0 | 3 | 100 | 70.8 | 40.5 |
| CUHK-**G−OPL/OPL**† | 1 | 0 | 1e−5 | 0.0 | 4 | 100 | 83.9 | 65.7 | 1 | 0 | 2.0 | 0.5 | 3 | 100 | 69.1 | 43.5 |
| Ped2-**G−OPL/OPL** | 5 | 0 | 0.1 | 0.5 | 2 | 200 | 89.6 | 75.0 | 1 | 0 | 5.0 | 0.5 | 2 | 200 | 93.9 | 93.6 |

Table 4: Optimal hyperparameters for RTFM and MGFN across all datasets. † indicates configurations using detected faces. G-OPL and OPL denote the presence of the respective projection layers; $\lambda_{\text{orth}}$ and $\lambda_{\text{face}}$ are the weights for orthogonality and face alignment losses, respectively. Batch size, training epochs, and resulting AUC and AP scores are also reported.

insufficient to fully remove them, especially if they re-emerge or evolve in deeper layers. By stacking OPL/G-OPL modules, we allow the network to iteratively refine and purify representations, removing unwanted components as they reappear or become linearly separable at different depths.

**Interpretation as multi-stage subspace filtering.** Each projection layer defines a learned orthogonal subspace corresponding to attributes that should be removed. Stacking multiple such layers can be interpreted as a multi-stage subspace filtering process:

$$\boldsymbol{f}^{(l+1)} = (\boldsymbol{I}_d - \boldsymbol{Q}^{(l)}\boldsymbol{Q}^{(l)\top})\boldsymbol{f}^{(l)}, \tag{12}$$

where each $\boldsymbol{Q}^{(l)}$ represents a learned sensitive or nuisance subspace at layer $l$. This recursive filtering ensures that different projections can specialize in removing distinct aspects of the signal, *e.g.*, identity in early layers, background clutter or temporal bias in later ones.

**Enhanced expressiveness and modularity.** From a learning perspective, stacking increases the expressiveness and flexibility of the projection mechanism. Rather than relying on a single, global subspace to capture all unwanted variation, the model learns a series of localized, lower-rank subspaces adapted to the representation space of each layer. This reduces optimization difficulty, improves stability, and allows each OPL/G-OPL to focus on a narrower, more interpretable source of variation.

**Improved trade-offs between utility and privacy.** Our experiments show that strategically stacking G-OPL/OPL at selected depths leads to improved privacy-utility trade-offs. G-OPLs placed early suppress high-sensitivity attributes (*e.g.*, face identity), while OPLs later in the network remove nuisance features that could degrade anomaly detection performance (*e.g.*, lighting shifts or background dynamics). This staged design helps retain task-relevant signals while minimizing privacy leakage and interpretability loss.

> Stacking G-OPL/OPL is a principled architectural strategy grounded in the hierarchical nature of learned features. It enables progressive and interpretable removal of harmful signals, improves privacy-utility balance, and supports modular integration in varied deployment settings.

## A.9 DETAILED SETUPS AND CONFIGURATIONS

To evaluate the impact of layer placement, we conduct extensive experiments using two representative VAD models: RTFM and MGFN. Our study spans five widely adopted datasets, MSAD, ShanghaiTech (ShT), UCF-Crime (UCF), CUHK Avenue (CUHK), and UCSD Ped2 (Ped2), to ensure broad applicability and robustness.

We explore two types of face representations to guide the G-OPL module: *(i) Detected faces*: For datasets with discernible facial content (ShT, UCF, CUHK), we use RetinaFace to extract faces and generate corresponding face videos. These are processed alongside the original videos using a Kinetics-pretrained I3D model to obtain face embeddings. *(ii) Generated faces*: To simulate face guidance without relying on in-dataset detection, we create synthetic face videos from the Georgia Tech Face Database, comprising 50 subjects with 15 images each. These are converted into videos and processed in the same way to serve as a generalized face prior.

For the three datasets with detectable faces, we directly compare the effects of using detected versus generated face signals to guide G-OPL.

| Method | Assault | | Explosion | | Fighting | | Fire | | Obj. Fall | | People Fall | | Robbery | | Shooting | | Traffic Acc. | | Vandalism | | Water Inc. | | **Overall** | |
|---|---|---|---|---|---|---|---|---|---|---|---|---|---|---|---|---|---|---|---|---|---|---|---|---|
| | AUC | AP | AUC | AP | AUC | AP | AUC | AP | AUC | AP | AUC | AP | AUC | AP | AUC | AP | AUC | AP | AUC | AP | AUC | AP | AUC | AP |
| MSAD (B1) | 71.3 | 69.4 | 61.8 | 73.0 | 87.8 | 92.8 | 81.0 | 92.9 | 94.3 | 96.5 | 45.9 | 45.0 | 65.1 | 81.1 | 82.7 | 89.1 | 64.2 | 55.2 | 90.7 | 86.4 | 68.7 | 91.9 | 86.2 | 68.3 |
| MSAD (B2) | 61.6 | 67.6 | 45.6 | 55.2 | 66.2 | 75.3 | 67.6 | 82.4 | 85.8 | 84.5 | 44.3 | 39.2 | 65.1 | 78.0 | 57.7 | 69.7 | 56.2 | 48.9 | 76.5 | 71.3 | 95.7 | 97.1 | 85.1 | 59.0 |
| MSAD (B3) | 53.7 | 56.0 | 59.2 | 68.7 | 84.6 | 90.8 | 81.0 | 91.2 | 91.3 | 95.3 | 48.6 | 46.8 | 63.5 | 78.5 | 90.9 | 94.0 | 71.4 | 63.9 | 85.3 | 81.5 | 97.7 | 99.4 | 84.3 | 62.8 |
| MSAD (B1B2B3) | 55.7 | 59.7 | 41.6 | 56.6 | 86.7 | 91.9 | 73.8 | 89.0 | 89.5 | 93.9 | 50.8 | 46.3 | 63.4 | 79.6 | 87.6 | 89.4 | 65.3 | 56.2 | 88.2 | 83.9 | 85.3 | 96.7 | 82.8 | 59.7 |
| MSAD (C1) | 56.8 | 59.9 | 47.0 | 60.7 | 75.6 | 87.0 | 75.7 | 87.1 | 90.5 | 94.7 | 49.2 | 48.4 | 64.8 | 79.6 | 77.9 | 88.1 | 66.7 | 56.7 | 80.5 | 73.7 | 90.1 | 97.8 | 82.9 | 58.6 |
| MSAD (C2) | 66.6 | 71.7 | 57.6 | 65.5 | 94.2 | 96.4 | 78.5 | 91.2 | 90.4 | 94.4 | 52.1 | 50.6 | 66.5 | 82.1 | 88.3 | 90.3 | 67.9 | 58.3 | 70.7 | 69.2 | 82.0 | 96.1 | 85.3 | 65.8 |
| MSAD (C1C2) | 56.4 | 59.2 | 53.9 | 66.5 | 81.4 | 89.4 | 80.3 | 90.1 | 91.6 | 95.5 | 49.7 | 47.0 | 69.6 | 83.7 | 89.1 | 91.7 | 67.7 | 56.0 | 76.6 | 75.0 | 83.0 | 96.4 | 84.1 | 62.5 |
| MSAD (B1C1) | 58.5 | 57.4 | 50.4 | 64.0 | 74.2 | 87.1 | 85.6 | 92.5 | 90.4 | 94.6 | 45.9 | 44.2 | 66.4 | 80.8 | 78.8 | 86.0 | 62.6 | 50.7 | 84.2 | 80.8 | 99.2 | 99.8 | 84.8 | 63.8 |
| MSAD (B2C2) | 46.9 | 52.0 | 58.8 | 69.5 | 77.6 | 88.4 | 83.4 | 91.7 | 90.2 | 94.6 | 53.8 | 47.5 | 68.9 | 83.1 | 74.5 | 83.9 | 58.8 | 48.2 | 83.9 | 74.9 | 84.5 | 96.7 | 83.6 | 61.2 |
| MSAD (ALL) | 59.5 | 57.4 | 53.0 | 57.8 | 79.7 | 86.9 | 66.7 | 84.0 | 87.2 | 85.6 | 53.7 | 48.4 | 68.0 | 81.4 | 77.3 | 81.9 | 63.6 | 50.7 | 84.2 | 80.8 | 99.2 | 99.8 | 84.8 | 63.8 |

Table 5: Anomaly-wise performance of the MGFN model using only OPL configurations on the MSAD dataset.

| Method | Frontdoor | | Mall | | Office | | Parkinglot | | Pedestr. st. | | Restaurant | | Road | | Shop | | Sidewalk | | St. highview | | Train | | Warehouse | | **Overall** | |
|---|---|---|---|---|---|---|---|---|---|---|---|---|---|---|---|---|---|---|---|---|---|---|---|---|---|---|
| | AUC | AP | AUC | AP | AUC | AP | AUC | AP | AUC | AP | AUC | AP | AUC | AP | AUC | AP | AUC | AP | AUC | AP | AUC | AP | AUC | AP | AUC | AP |
| MSAD (B1) | 84.4 | 84.0 | 80.2 | 74.7 | 74.7 | 65.0 | 87.0 | 30.9 | 93.5 | 53.0 | 91.2 | 87.6 | 80.0 | 55.7 | 82.1 | 69.3 | 86.8 | 63.8 | 98.1 | 95.1 | 70.8 | 9.1 | 89.9 | 76.1 | 86.2 | 68.3 |
| MSAD (B2) | 82.0 | 75.8 | 84.3 | 64.3 | 75.2 | 60.5 | 69.5 | 18.9 | 93.1 | 35.4 | 87.2 | 70.8 | 76.1 | 39.6 | 84.1 | 69.7 | 81.8 | 49.4 | 95.1 | 76.6 | 69.2 | 12.7 | 81.2 | 42.6 | 85.1 | 59.0 |
| MSAD (B3) | 87.5 | 85.5 | 79.1 | 76.3 | 74.7 | 64.4 | 82.7 | 25.5 | 86.9 | 15.1 | 92.9 | 88.6 | 74.7 | 43.6 | 80.2 | 62.7 | 84.6 | 63.4 | 90.3 | 49.7 | 51.8 | 5.5 | 71.1 | 35.0 | 84.3 | 62.8 |
| MSAD (B1B2B3) | 83.2 | 82.4 | 58.2 | 57.0 | 69.2 | 57.8 | 81.5 | 24.6 | 82.8 | 12.1 | 88.4 | 84.4 | 81.9 | 45.9 | 81.4 | 67.1 | 88.9 | 65.0 | 93.8 | 54.3 | 52.4 | 4.8 | 67.5 | 24.3 | 82.8 | 59.7 |
| MSAD (C1) | 85.6 | 82.7 | 54.7 | 48.7 | 70.0 | 59.2 | 82.6 | 25.3 | 80.7 | 11.0 | 87.6 | 81.5 | 79.1 | 43.4 | 81.8 | 69.1 | 83.8 | 61.0 | 85.7 | 35.5 | 53.2 | 3.0 | 75.0 | 35.6 | 82.9 | 58.6 |
| MSAD (C2) | 85.3 | 82.6 | 62.7 | 67.8 | 72.0 | 62.7 | 86.7 | 40.3 | 86.7 | 15.3 | 91.7 | 88.1 | 85.7 | 60.7 | 78.0 | 58.7 | 89.7 | 74.1 | 91.9 | 64.4 | 68.0 | 7.3 | 82.5 | 50.0 | 85.3 | 65.8 |
| MSAD (C1C2) | 85.2 | 84.1 | 57.0 | 64.5 | 74.3 | 61.6 | 78.5 | 20.7 | 88.7 | 22.7 | 91.9 | 87.1 | 77.5 | 43.4 | 83.2 | 67.6 | 84.6 | 61.5 | 87.2 | 41.8 | 62.0 | 8.2 | 67.8 | 28.5 | 84.1 | 62.5 |
| MSAD (B1C1) | 83.1 | 82.0 | 86.4 | 81.0 | 73.4 | 57.5 | 75.7 | 19.0 | 86.4 | 14.6 | 96.3 | 91.7 | 74.0 | 43.5 | 85.3 | 75.5 | 84.4 | 64.1 | 84.3 | 39.0 | 49.5 | 4.7 | 69.8 | 28.6 | 84.8 | 63.8 |
| MSAD (B2C2) | 86.4 | 83.5 | 91.0 | 88.7 | 71.1 | 55.3 | 63.9 | 13.3 | 87.8 | 20.6 | 96.7 | 94.5 | 71.3 | 37.7 | 83.4 | 75.2 | 84.3 | 62.3 | 88.3 | 38.5 | 55.6 | 3.5 | 74.4 | 31.0 | 83.6 | 61.3 |
| MSAD (ALL) | 82.0 | 77.1 | 78.2 | 61.1 | 72.5 | 57.9 | 64.0 | 13.7 | 92.4 | 28.2 | 94.2 | 84.1 | 68.1 | 33.5 | 83.4 | 71.7 | 84.7 | 55.6 | 77.5 | 24.1 | 50.9 | 3.1 | 75.0 | 31.8 | 82.5 | 56.3 |

Table 6: Scenario-wise performance of the MGFN model using only OPL configurations on the MSAD dataset.

For placing the OPL and G-OPL layers: (i) for RTFM, we have 6 predefined insertion points

Optimal hyperparameter values for each configuration are summarized in Table 4.

## A.10 ADDITIONAL RESULTS AND EVALUATIONS

**Evaluating the effects of $\lambda_{\text{orth}}$ and $\lambda_{\text{face}}$.** We introduce two critical hyperparameters to modulate the strength of our projection-based objectives: (i) $\lambda_{\text{orth}}$, which controls the orthogonality regularization in OPL/G-OPL to stabilize disentanglement, and (ii) $\lambda_{\text{face}}$, which weights the face-guided suppression loss in G-OPL to enforce privacy preservation.

To understand their influence, we conduct a comprehensive grid search over a wide range of candidate values: 0.0, 1e−5, 1e−4, 1e−3, 1e−2, 0.1, 0.2, 0.3, 0.5, 1.0, 2.0, 3.0, 5.0 for both $\lambda_{\text{orth}}$ and $\lambda_{\text{face}}$, resulting in 169 unique combinations. Each setting is evaluated across five benchmark VAD datasets using both RTFM and MGFN architectures.

The optimal configurations, those that yield the best balance between anomaly detection performance and ethical alignment, are summarized in Table 4. Notably, the ideal weights vary across datasets and models, suggesting that sensitivity to orthogonality and privacy constraints is task- and architecture-dependent. For instance, RTFM often benefits from stronger privacy guidance (*e.g.*,

| Method | Assault | | Explosion | | Fighting | | Fire | | Obj. Fall | | People Fall | | Robbery | | Shooting | | Traffic Acc. | | Vandalism | | Water Inc. | | **Overall** | |
|---|---|---|---|---|---|---|---|---|---|---|---|---|---|---|---|---|---|---|---|---|---|---|---|---|
| | AUC | AP | AUC | AP | AUC | AP | AUC | AP | AUC | AP | AUC | AP | AUC | AP | AUC | AP | AUC | AP | AUC | AP | AUC | AP | AUC | AP |
| MSAD (G1O0) | 52.4 | 59.8 | 66.5 | 76.8 | 88.8 | 92.2 | 77.2 | 89.0 | 90.5 | 95.1 | 45.9 | 42.8 | 65.4 | 80.1 | 71.9 | 81.8 | 53.9 | 46.4 | 83.1 | 75.1 | 81.5 | 96.0 | 84.0 | 65.8 |
| MSAD (G1O1) | 57.0 | 59.5 | 52.2 | 61.3 | 85.9 | 91.1 | 58.8 | 79.9 | 91.9 | 95.7 | 50.1 | 47.3 | 73.1 | 88.0 | 75.3 | 84.5 | 56.4 | 47.8 | 86.5 | 81.7 | 70.2 | 92.8 | 81.8 | 63.4 |
| MSAD (G1O2) | 53.1 | 62.5 | 64.6 | 58.3 | 84.7 | 90.5 | 80.1 | 89.8 | 92.8 | 95.9 | 44.7 | 44.4 | 63.6 | 80.8 | 70.3 | 81.9 | 55.9 | 49.6 | 85.4 | 82.2 | 77.9 | 94.9 | 82.7 | 64.5 |
| MSAD (G1O3) | 57.1 | 62.1 | 62.8 | 76.5 | 71.1 | 86.0 | 38.2 | 68.3 | 90.6 | 94.9 | 52.5 | 50.0 | 69.3 | 85.4 | 71.0 | 83.4 | 64.1 | 58.4 | 77.2 | 73.2 | 96.4 | 99.1 | 82.2 | 65.0 |
| MSAD (G1O4) | 60.6 | 62.8 | 58.8 | 67.5 | 87.0 | 91.7 | 74.1 | 87.3 | 90.5 | 94.8 | 52.5 | 49.0 | 71.9 | 83.8 | 89.4 | 92.6 | 66.9 | 54.7 | 78.6 | 79.3 | 81.0 | 95.8 | 84.3 | 62.8 |
| MSAD (G2O0) | 56.2 | 55.6 | 50.8 | 54.4 | 83.7 | 90.0 | 74.9 | 88.3 | 90.5 | 94.4 | 54.5 | 53.3 | 71.1 | 85.3 | 84.4 | 87.8 | 62.7 | 53.1 | 77.9 | 81.0 | 76.5 | 94.8 | 83.0 | 62.1 |
| MSAD (G2O1) | 51.8 | 56.1 | 46.3 | 56.3 | 85.4 | 91.0 | 71.0 | 86.0 | 91.4 | 95.3 | 50.7 | 47.1 | 82.1 | 86.9 | 59.3 | 51.2 | 86.5 | 81.9 | 83.5 | 96.5 | | | 83.5 | 63.4 |
| MSAD (G2O2) | 62.1 | 68.2 | 47.3 | 56.2 | 83.1 | 87.9 | 79.6 | 90.0 | 89.5 | 94.2 | 37.7 | 39.3 | 65.6 | 83.2 | 75.1 | 84.0 | 56.8 | 53.0 | 83.6 | 74.4 | 71.4 | 93.2 | 82.3 | 64.2 |
| MSAD (G2O3) | 58.2 | 64.0 | 63.6 | 62.0 | 84.7 | 91.2 | 62.3 | 82.9 | 90.5 | 94.7 | 51.6 | 49.3 | 68.6 | 81.7 | 78.8 | 83.5 | 57.6 | 49.3 | 88.4 | 86.2 | 89.5 | 97.6 | 84.4 | 66.0 |
| MSAD (G3O0) | 59.9 | 66.9 | 67.5 | 78.1 | 79.5 | 88.6 | 44.5 | 73.4 | 91.2 | 95.4 | 51.2 | 47.0 | 66.9 | 81.4 | 75.4 | 84.1 | 60.3 | 50.2 | 80.8 | 73.6 | 94.3 | 98.6 | 79.8 | 61.7 |
| MSAD (G3O1) | 65.1 | 70.7 | 52.5 | 58.9 | 79.2 | 86.1 | 61.6 | 79.9 | 89.6 | 94.2 | 38.0 | 39.0 | 68.0 | 84.2 | 65.0 | 71.8 | 53.1 | 48.0 | 79.0 | 68.6 | 76.7 | 94.4 | 82.3 | 61.3 |
| MSAD (G3O2) | 63.1 | 61.5 | 47.3 | 53.1 | 72.3 | 85.5 | 46.9 | 76.6 | 92.4 | 96.0 | 39.5 | 41.3 | 67.3 | 76.4 | 72.3 | 82.6 | 58.5 | 48.5 | 81.1 | 80.0 | 99.1 | 99.8 | 84.8 | 65.1 |
| MSAD (G4O0) | 60.3 | 67.6 | 67.4 | 78.0 | 79.1 | 88.4 | 43.9 | 73.1 | 91.2 | 95.3 | 51.3 | 47.0 | 66.8 | 81.4 | 74.7 | 83.2 | 60.2 | 50.2 | 81.0 | 73.8 | 95.2 | 98.8 | 79.8 | 61.5 |
| MSAD (G4O1) | 59.6 | 65.7 | 66.7 | 77.5 | 77.8 | 87.6 | 39.1 | 70.6 | 90.6 | 94.9 | 50.7 | 45.9 | 66.5 | 81.9 | 75.3 | 84.2 | 58.1 | 48.8 | 81.4 | 73.8 | 96.1 | 99.0 | 80.6 | 62.3 |
| MSAD (G5O0) | 64.7 | 69.7 | 49.7 | 60.5 | 67.8 | 81.5 | 43.0 | 72.7 | 88.3 | 93.6 | 35.4 | 41.2 | 70.8 | 82.3 | 71.3 | 80.1 | 53.8 | 50.1 | 76.1 | 74.0 | 97.2 | 99.3 | 82.0 | 63.2 |

Table 7: MGFN with G-OPL/OPL on MSAD (anomaly-level). Anomaly-wise performance of various GxOy configurations in the MGFN model on the MSAD dataset. The G1O0 configuration consistently achieves top results across different anomaly types.

| Method | Frontdoor | | Mall | | Office | | Parkinglot | | Pedestr. st. | | Restaurant | | Road | | Shop | | Sidewalk | | St. highview | | Train | | Warehouse | | **Overall** | |
|---|---|---|---|---|---|---|---|---|---|---|---|---|---|---|---|---|---|---|---|---|---|---|---|---|---|---|
| | AUC | AP | AUC | AP | AUC | AP | AUC | AP | AUC | AP | AUC | AP | AUC | AP | AUC | AP | AUC | AP | AUC | AP | AUC | AP | AUC | AP | AUC | AP |
| MSAD (G1O0) | 84.4 | 83.8 | 90.0 | 84.8 | 75.9 | 62.3 | 70.4 | 16.6 | 90.5 | 25.8 | 95.7 | 90.2 | 71.4 | 43.1 | 79.7 | 64.5 | 83.8 | 63.3 | 87.7 | 41.3 | 44.7 | 2.3 | 64.8 | 41.1 | 84.0 | 65.8 |
| MSAD (G1O1) | 83.5 | 83.0 | 53.6 | 49.6 | 73.7 | 60.8 | 71.2 | 21.1 | 86.8 | 14.9 | 91.8 | 87.0 | 73.0 | 38.4 | 86.8 | 81.8 | 84.4 | 60.6 | 69.1 | 27.5 | 47.8 | 2.5 | 55.6 | 24.7 | 81.8 | 63.4 |
| MSAD (G1O2) | 82.8 | 82.9 | 63.4 | 60.9 | 74.5 | 62.0 | 75.5 | 20.1 | 84.4 | 13.1 | 93.8 | 88.5 | 72.6 | 39.9 | 84.4 | 75.4 | 84.5 | 62.0 | 97.2 | 86.1 | 44.6 | 2.3 | 66.3 | 37.7 | 82.7 | 64.5 |
| MSAD (G1O3) | 85.5 | 84.5 | 86.7 | 82.6 | 75.0 | 66.1 | 74.5 | 30.7 | 62.3 | 7.4 | 84.2 | 80.4 | 74.9 | 42.3 | 82.9 | 77.1 | 85.3 | 61.7 | 60.5 | 24.4 | 61.4 | 37.1 | 66.4 | 38.3 | 82.2 | 65.0 |
| MSAD (G1O4) | 87.2 | 85.9 | 77.3 | 74.6 | 73.9 | 61.7 | 79.6 | 22.9 | 80.2 | 13.1 | 88.6 | 83.8 | 74.1 | 40.9 | 83.0 | 69.8 | 85.8 | 62.2 | 90.1 | 52.0 | 45.3 | 2.5 | 71.3 | 39.8 | 84.3 | 62.8 |
| MSAD (G2O0) | 83.4 | 82.7 | 54.6 | 45.7 | 72.9 | 60.4 | 68.6 | 14.9 | 86.9 | 14.8 | 86.9 | 81.9 | 73.3 | 44.4 | 82.6 | 74.8 | 86.3 | 61.7 | 92.7 | 50.7 | 37.8 | 4.1 | 66.2 | 24.5 | 83.0 | 62.1 |
| MSAD (G2O1) | 84.4 | 83.6 | 67.8 | 59.2 | 73.9 | 61.4 | 79.9 | 21.6 | 92.6 | 28.6 | 94.9 | 90.3 | 66.3 | 34.8 | 83.4 | 73.6 | 83.5 | 63.8 | 92.5 | 58.3 | 45.7 | 2.4 | 60.5 | 33.9 | 83.5 | 63.4 |
| MSAD (G2O2) | 81.0 | 80.6 | 58.6 | 59.1 | 75.4 | 60.7 | 73.0 | 18.5 | 90.2 | 22.0 | 89.2 | 81.5 | 61.6 | 33.0 | 82.1 | 74.6 | 73.3 | 55.7 | 97.6 | 87.4 | 65.1 | 10.7 | 71.3 | 39.6 | 82.3 | 64.2 |
| MSAD (G2O3) | 82.6 | 82.1 | 89.2 | 83.5 | 77.4 | 62.9 | 73.9 | 20.0 | 95.3 | 44.1 | 94.3 | 88.4 | 73.7 | 45.6 | 83.8 | 73.3 | 86.1 | 64.3 | 66.7 | 25.3 | 50.2 | 2.6 | 84.4 | 58.3 | 84.4 | 66.0 |
| MSAD (G3O0) | 85.9 | 84.6 | 89.3 | 84.4 | 73.7 | 63.8 | 64.2 | 13.4 | 74.3 | 52.2 | 85.2 | 81.4 | 65.3 | 35.5 | 84.1 | 74.9 | 81.9 | 58.8 | 62.6 | 25.3 | 49.0 | 2.5 | 65.5 | 38.4 | 79.8 | 61.7 |
| MSAD (G3O1) | 78.9 | 76.8 | 70.7 | 54.0 | 76.8 | 64.0 | 54.2 | 11.0 | 91.4 | 22.9 | 93.5 | 87.1 | 66.0 | 30.2 | 85.0 | 75.0 | 76.9 | 56.6 | 83.3 | 37.4 | 53.6 | 3.0 | 81.6 | 40.3 | 82.3 | 61.3 |
| MSAD (G3O2) | 84.4 | 84.8 | 94.6 | 91.3 | 74.8 | 59.0 | 71.2 | 19.7 | 88.0 | 17.0 | 93.1 | 85.7 | 69.9 | 34.9 | 84.7 | 70.6 | 83.6 | 61.3 | 77.1 | 31.5 | 57.2 | 5.4 | 65.3 | 24.8 | 84.8 | 65.1 |
| MSAD (G4O0) | 85.9 | 84.6 | 89.8 | 85.7 | 73.7 | 64.0 | 64.7 | 13.6 | 73.8 | 51.5 | 85.2 | 81.3 | 65.3 | 36.2 | 84.3 | 75.2 | 81.7 | 58.3 | 61.6 | 25.1 | 50.8 | 2.6 | 65.8 | 38.4 | 79.8 | 61.5 |
| MSAD (G4O1) | 85.8 | 84.2 | 85.5 | 85.1 | 73.6 | 63.2 | 67.5 | 14.9 | 75.5 | 47.1 | 85.3 | 81.6 | 67.4 | 36.3 | 83.5 | 74.8 | 82.2 | 58.6 | 57.6 | 24.1 | 49.7 | 2.6 | 70.4 | 44.6 | 80.6 | 62.3 |
| MSAD (G5O0) | 78.8 | 79.5 | 68.7 | 67.1 | 74.9 | 62.4 | 67.1 | 18.0 | 88.6 | 16.4 | 89.1 | 79.0 | 67.3 | 41.1 | 83.8 | 69.9 | 83.2 | 58.3 | 52.3 | 22.4 | 61.3 | 4.5 | 73.3 | 35.4 | 82.1 | 63.2 |

Table 8: MGFN with G-OPL/OPL on MSAD (scenario-level). Scenario-wise results for different $GxOy$ configurations using MGFN on the MSAD dataset. The G1O0 setup consistently outperforms most other placements across scenarios.

| Method | Assault | | Explosion | | Fighting | | Fire | | Obj. Fall | | People Fall | | Robbery | | Shooting | | Traffic Acc. | | Vandalism | | Water Inc. | | **Overall** | |
|---|---|---|---|---|---|---|---|---|---|---|---|---|---|---|---|---|---|---|---|---|---|---|---|---|
| | AUC | AP | AUC | AP | AUC | AP | AUC | AP | AUC | AP | AUC | AP | AUC | AP | AUC | AP | AUC | AP | AUC | AP | AUC | AP | AUC | AP |
| MSAD (C1) | 57.0 | 62.4 | 77.7 | 85.7 | 74.1 | 84.8 | 49.6 | 75.5 | 87.7 | 92.1 | 53.3 | 50.4 | 72.4 | 89.0 | 84.1 | 89.5 | 69.5 | 58.7 | 84.8 | 80.9 | 99.2 | 99.8 | 86.5 | 68.2 |
| MSAD (C2) | 51.6 | 66.1 | 73.2 | 82.7 | 76.8 | 85.5 | 61.0 | 82.8 | 83.4 | 89.2 | 47.7 | 54.7 | 68.9 | 87.8 | 75.6 | 82.1 | 84.4 | 62.1 | 86.9 | 83.2 | 98.7 | 99.7 | 86.0 | 68.9 |
| MSAD (C3) | 63.5 | 74.4 | 85.9 | 90.0 | 81.0 | 88.6 | 76.6 | 87.8 | 89.3 | 93.5 | 58.8 | 57.4 | 67.8 | 86.3 | 82.0 | 88.3 | 64.0 | 58.5 | 89.4 | 87.5 | 98.9 | 99.8 | 85.1 | 69.0 |
| MSAD (C4) | 63.1 | 74.8 | 72.8 | 84.0 | 80.0 | 86.9 | 75.5 | 79.6 | 86.4 | 91.7 | 53.3 | 56.2 | 70.0 | 88.7 | 75.6 | 80.8 | 64.3 | 58.3 | 84.3 | 82.2 | 98.3 | 99.6 | 85.0 | 68.7 |
| MSAD (C5) | 66.2 | 70.8 | 79.6 | 86.7 | 77.0 | 86.9 | 51.6 | 76.1 | 90.1 | 93.5 | 44.2 | 48.9 | 73.1 | 88.1 | 87.4 | 90.7 | 59.6 | 57.7 | 74.7 | 71.2 | 99.5 | 99.9 | 86.2 | 68.6 |
| MSAD (C2C3) | 67.3 | 72.0 | 74.7 | 83.2 | 69.4 | 84.2 | 56.2 | 77.5 | 88.3 | 92.4 | 52.6 | 53.8 | 65.9 | 85.2 | 88.6 | 89.3 | 64.5 | 58.4 | 78.8 | 75.4 | 99.9 | 100 | 85.6 | 68.2 |
| MSAD (ALL) | 52.8 | 60.6 | 72.0 | 82.1 | 75.8 | 86.4 | 52.8 | 77.2 | 88.6 | 92.4 | 56.1 | 52.5 | 71.6 | 88.2 | 81.7 | 88.3 | 66.0 | 55.5 | 84.5 | 81.0 | 99.7 | 99.9 | 86.4 | 69.5 |

Table 9: Anomaly-wise performance of the RTFM model using only OPL configurations on the MSAD dataset.

| Method | Frontdoor | | Mall | | Office | | Parkinglot | | Pedestr. st. | | Restaurant | | Road | | Shop | | Sidewalk | | St. highview | | Train | | Warehouse | | **Overall** | |
|---|---|---|---|---|---|---|---|---|---|---|---|---|---|---|---|---|---|---|---|---|---|---|---|---|---|---|
| | AUC | AP | AUC | AP | AUC | AP | AUC | AP | AUC | AP | AUC | AP | AUC | AP | AUC | AP | AUC | AP | AUC | AP | AUC | AP | AUC | AP | AUC | AP |
| MSAD (C1) | 85.7 | 82.4 | 85.7 | 80.3 | 77.2 | 71.8 | 76.8 | 26.3 | 96.6 | 50.4 | 90.4 | 81.5 | 76.8 | 53.3 | 88.6 | 82.9 | 84.9 | 65.5 | 66.5 | 26.7 | 42.9 | 2.3 | 86.1 | 66.8 | 86.5 | 68.2 |
| MSAD (C2) | 84.9 | 83.0 | 91.2 | 87.3 | 76.5 | 73.7 | 77.1 | 23.3 | 76.7 | 11.8 | 85.1 | 74.3 | 66.0 | 39.0 | 85.3 | 79.0 | 86.2 | 58.0 | 63.3 | 26.1 | 71.0 | 20.3 | 88.0 | 72.8 | 86.0 | 68.9 |
| MSAD (C3) | 89.1 | 87.7 | 85.9 | 80.5 | 80.8 | 78.3 | 60.3 | 23.4 | 86.8 | 14.9 | 92.2 | 86.6 | 80.8 | 54.9 | 77.3 | 68.6 | 80.4 | 57.1 | 80.3 | 34.8 | 63.4 | 21.8 | 83.5 | 69.4 | 85.1 | 69.0 |
| MSAD (C4) | 84.6 | 83.8 | 94.5 | 88.8 | 74.9 | 72.0 | 81.1 | 29.6 | 81.3 | 12.0 | 89.6 | 79.4 | 77.4 | 50.5 | 83.9 | 78.2 | 84.9 | 61.3 | 52.2 | 23.1 | 50.4 | 17.5 | 81.0 | 56.3 | 85.0 | 68.7 |
| MSAD (C5) | 82.1 | 80.5 | 97.5 | 95.9 | 84.2 | 79.7 | 87.4 | 64.4 | 78.6 | 11.3 | 88.1 | 79.1 | 86.4 | 58.6 | 82.4 | 68.3 | 90.1 | 71.7 | 48.0 | 22.4 | 91.2 | 28.2 | 80.0 | 54.6 | 86.2 | 68.6 |
| MSAD (C2C3) | 83.4 | 80.8 | 89.3 | 86.3 | 81.7 | 76.9 | 87.2 | 36.3 | 76.9 | 9.2 | 84.7 | 77.9 | 79.6 | 50.9 | 80.9 | 74.3 | 88.7 | 67.7 | 69.3 | 27.6 | 67.8 | 4.8 | 84.8 | 60.9 | 85.6 | 68.2 |
| MSAD (ALL) | 86.0 | 83.3 | 87.9 | 83.6 | 77.6 | 72.4 | 85.2 | 33.4 | 78.8 | 21.9 | 89.6 | 80.8 | 73.4 | 46.9 | 87.4 | 82.2 | 86.6 | 65.5 | 61.8 | 25.1 | 54.2 | 2.8 | 88.1 | 69.6 | 86.4 | 69.5 |

Table 10: Scenario-wise performance of the RTFM model using only OPL configurations on the MSAD dataset.

| Method | Assault | | Explosion | | Fighting | | Fire | | Obj. Fall | | People Fall | | Robbery | | Shooting | | Traffic Acc. | | Vandalism | | Water Inc. | | **Overall** | |
|---|---|---|---|---|---|---|---|---|---|---|---|---|---|---|---|---|---|---|---|---|---|---|---|---|
| | AUC | AP | AUC | AP | AUC | AP | AUC | AP | AUC | AP | AUC | AP | AUC | AP | AUC | AP | AUC | AP | AUC | AP | AUC | AP | AUC | AP |
| MSAD (G1O0) | 50.2 | 62.4 | 69.4 | 80.6 | 69.5 | 84.4 | 71.8 | 87.0 | 88.7 | 92.4 | 52.3 | 53.3 | 71.4 | 88.2 | 87.9 | 91.0 | 62.5 | 54.7 | 82.0 | 79.6 | 97.5 | 99.4 | 88.0 | 70.9 |
| MSAD (G1O1) | 66.0 | 77.5 | 69.6 | 80.8 | 66.5 | 81.2 | 55.2 | 78.9 | 78.0 | 86.7 | 55.1 | 68.3 | 64.7 | 86.6 | 74.5 | 82.5 | 65.2 | 63.5 | 85.7 | 81.8 | 100 | 100 | 86.6 | 70.5 |
| MSAD (G1O2) | 50.6 | 72.4 | 56.2 | 78.3 | 58.4 | 81.1 | 50.9 | 80.7 | 79.9 | 88.0 | 47.2 | 59.4 | 65.0 | 89.0 | 66.4 | 80.2 | 66.4 | 68.7 | 88.6 | 87.8 | 83.8 | 96.5 | 84.8 | 70.3 |
| MSAD (G1O3) | 55.4 | 68.5 | 73.5 | 84.2 | 75.5 | 85.1 | 72.7 | 87.2 | 82.3 | 88.1 | 39.2 | 50.7 | 70.2 | 88.4 | 67.2 | 79.0 | 60.9 | 63.2 | 79.7 | 79.9 | 98.9 | 96.9 | 85.5 | 69.2 |
| MSAD (G1O4) | 46.6 | 67.5 | 68.9 | 82.3 | 71.4 | 83.2 | 79.6 | 90.8 | 84.4 | 90.2 | 48.4 | 60.1 | 67.5 | 88.6 | 69.7 | 81.1 | 67.1 | 65.4 | 86.8 | 87.0 | 95.8 | 99.1 | 85.2 | 69.7 |
| MSAD (G1O5) | 45.5 | 62.9 | 74.9 | 85.0 | 78.3 | 86.8 | 86.8 | 94.4 | 84.5 | 89.4 | 46.5 | 55.1 | 69.1 | 87.7 | 80.0 | 85.8 | 66.2 | 63.2 | 79.7 | 79.9 | 93.5 | 98.6 | 86.3 | 69.7 |
| MSAD (G2O0) | 44.9 | 66.9 | 67.3 | 82.3 | 73.0 | 84.0 | 73.8 | 88.0 | 80.9 | 88.6 | 43.7 | 58.2 | 65.7 | 88.4 | 80.2 | 87.2 | 68.1 | 66.7 | 82.9 | 83.4 | 99.8 | 100 | 86.5 | 70.9 |
| MSAD (G2O1) | 61.4 | 73.4 | 69.5 | 82.4 | 74.1 | 84.2 | 66.7 | 86.1 | 91.3 | | 42.3 | 53.8 | 65.5 | 87.7 | 82.4 | 87.9 | 65.7 | 65.1 | 85.2 | 85.1 | 99.7 | 99.9 | 85.4 | 69.6 |
| MSAD (G2O2) | 50.4 | 74.7 | 60.6 | 79.6 | 69.6 | 84.6 | 54.3 | 83.1 | 71.5 | 84.2 | 51.9 | 70.1 | 65.2 | 88.6 | 81.7 | 87.9 | 64.7 | 66.6 | 82.6 | 83.7 | 99.8 | 100 | 83.5 | 70.5 |
| MSAD (G2O3) | 50.1 | 69.8 | 67.4 | 82.1 | 68.6 | 85.9 | 71.9 | 86.3 | 80.4 | 88.1 | 47.8 | 60.8 | 66.4 | 88.0 | 73.0 | 83.1 | 63.3 | 62.4 | 85.5 | 83.7 | 99.0 | 99.7 | 86.9 | 71.6 |
| MSAD (G2O4) | 59.3 | 74.1 | 65.3 | 80.6 | 69.1 | 83.1 | 76.6 | 89.8 | 86.1 | 90.8 | 43.9 | 52.4 | 64.9 | 87.2 | 70.5 | 82.8 | 63.3 | 67.7 | 91.4 | 89.5 | 97.7 | 99.3 | 86.7 | 70.7 |
| MSAD (G3O0) | 58.0 | 72.0 | 66.7 | 81.1 | 60.4 | 81.7 | 72.7 | 87.1 | 83.8 | 90.0 | 41.6 | 54.4 | 67.3 | 88.3 | 68.8 | 79.8 | 56.5 | 56.8 | 89.1 | 87.4 | 98.9 | 99.7 | 86.1 | 70.1 |
| MSAD (G3O1) | 59.1 | 74.1 | 61.3 | 78.9 | 71.0 | 82.9 | 79.6 | 89.6 | 86.3 | 91.2 | 49.1 | 58.0 | 66.4 | 88.3 | 65.8 | 79.5 | 64.8 | 64.0 | 89.1 | 88.3 | 85.6 | 96.7 | 86.5 | 70.4 |
| MSAD (G3O2) | 66.0 | 79.1 | 64.8 | 80.0 | 67.9 | 81.1 | 44.0 | 76.1 | 72.7 | 84.3 | 45.4 | 61.8 | 66.5 | 88.2 | 81.6 | 86.7 | 61.7 | 57.3 | 81.0 | 78.6 | 99.9 | 100 | 85.2 | 69.3 |
| MSAD (G3O3) | 46.9 | 66.2 | 60.9 | 79.1 | 62.6 | 80.0 | 80.1 | 91.2 | 84.2 | 89.9 | 47.3 | 53.9 | 64.3 | 88.1 | 70.2 | 82.5 | 68.7 | 67.7 | 86.2 | 84.5 | 97.4 | 99.3 | 85.2 | 69.7 |
| MSAD (G4O0) | 50.7 | 68.0 | 70.7 | 83.5 | 76.9 | 86.4 | 79.2 | 89.4 | 83.8 | 89.7 | 41.6 | 54.2 | 66.9 | 88.3 | 74.7 | 83.6 | 66.8 | 64.3 | 89.0 | 87.9 | 99.9 | 100 | 86.8 | 71.4 |
| MSAD (G4O1) | 64.7 | 79.0 | 61.5 | 79.3 | 63.9 | 79.5 | 44.5 | 76.5 | 72.3 | 84.5 | 50.1 | 67.7 | 65.0 | 87.9 | 78.1 | 84.9 | 63.6 | 64.8 | 85.5 | 84.1 | 99.9 | 100 | 85.1 | 70.3 |
| MSAD (G4O2) | 53.8 | 73.2 | 67.2 | 82.2 | 73.7 | 83.9 | 79.1 | 91.1 | 82.9 | 89.5 | 44.7 | 57.7 | 68.8 | 89.7 | 76.7 | 83.3 | 69.9 | 69.9 | 79.4 | 79.8 | 99.8 | 100 | 86.7 | 72.2 |
| MSAD (G5O0) | 60.1 | 77.6 | 57.5 | 78.4 | 67.9 | 83.3 | 43.6 | 76.2 | 72.7 | 84.7 | 50.9 | 68.8 | 63.2 | 87.7 | 70.7 | 81.6 | 62.0 | 64.9 | 81.5 | 82.3 | 99.5 | 99.9 | 83.0 | 68.7 |
| MSAD (G5O1) | 50.5 | 67.1 | 68.1 | 80.8 | 77.2 | 85.8 | 71.9 | 86.6 | 84.2 | 89.8 | 45.3 | 54.7 | 72.0 | 89.5 | 81.9 | 88.8 | 64.8 | 61.1 | 87.1 | 84.5 | 99.9 | 100 | 86.7 | 70.2 |
| MSAD (G6O0) | 58.7 | 73.9 | 66.2 | 80.1 | 59.8 | 79.9 | 76.0 | 88.2 | 84.3 | 90.2 | 41.5 | 56.0 | 66.0 | 87.7 | 83.4 | 86.1 | 61.0 | 59.5 | 86.1 | 85.0 | 95.3 | 99.0 | 87.2 | 70.6 |

Table 11: RTFM with G-OPL/OPL on MSAD (anomaly-level). Anomaly-wise performance of various $GxOy$ configurations in the RTFM model on the MSAD dataset. The G1O0 configuration consistently achieves top results across different anomaly types.

| Method | Frontdoor | | Mall | | Office | | Parkinglot | | Pedestr. st. | | Restaurant | | Road | | Shop | | Sidewalk | | St. highview | | Train | | Warehouse | | Overall | |
|---|---|---|---|---|---|---|---|---|---|---|---|---|---|---|---|---|---|---|---|---|---|---|---|---|---|---|
| | AUC | AP | AUC | AP | AUC | AP | AUC | AP | AUC | AP | AUC | AP | AUC | AP | AUC | AP | AUC | AP | AUC | AP | AUC | AP | AUC | AP | AUC | AP |
| MSAD (G1O0) | 82.0 | 79.3 | 91.0 | 81.4 | 74.3 | 72.0 | 79.4 | 27.2 | 86.9 | 36.1 | 90.3 | 81.4 | 72.4 | 46.7 | 89.0 | 82.5 | 87.0 | 65.1 | 84.9 | 37.8 | 70.4 | 12.0 | 86.3 | 79.6 | 88.0 | 70.9 |
| MSAD (G1O1) | 81.0 | 82.5 | 87.8 | 84.2 | 75.1 | 75.0 | 87.9 | 46.9 | 95.7 | 51.3 | 85.6 | 76.2 | 85.1 | 57.4 | 81.6 | 73.8 | 83.5 | 63.5 | 66.9 | 24.8 | 55.0 | 4.5 | 88.2 | 71.9 | 86.6 | 70.5 |
| MSAD (G1O2) | 80.2 | 82.2 | 88.4 | 80.6 | 75.2 | 75.7 | 80.6 | 47.0 | 86.7 | 25.0 | 90.4 | 81.5 | 83.3 | 64.4 | 82.8 | 79.2 | 83.6 | 62.2 | 90.3 | 65.3 | 51.7 | 2.6 | 76.8 | 63.4 | 84.8 | 70.3 |
| MSAD (G1O3) | 78.3 | 78.3 | 75.9 | 66.8 | 73.8 | 73.7 | 86.9 | 54.8 | 92.5 | 24.9 | 92.9 | 85.3 | 88.3 | 59.7 | 83.8 | 76.8 | 87.3 | 66.1 | 66.0 | 24.3 | 37.5 | 2.1 | 85.9 | 71.2 | 85.5 | 69.3 |
| MSAD (G1O4) | 82.2 | 82.7 | 70.6 | 63.1 | 73.6 | 74.7 | 69.1 | 34.7 | 98.1 | 74.7 | 92.7 | 85.2 | 84.4 | 53.4 | 82.4 | 77.0 | 82.5 | 59.7 | 86.3 | 56.7 | 19.7 | 1.7 | 87.3 | 69.0 | 85.2 | 69.7 |
| MSAD (G1O5) | 80.9 | 81.0 | 72.0 | 62.8 | 74.7 | 72.9 | 70.7 | 27.3 | 89.8 | 36.3 | 94.8 | 88.5 | 79.7 | 51.4 | 84.8 | 77.5 | 80.7 | 56.9 | 96.3 | 82.4 | 50.0 | 3.2 | 86.7 | 66.2 | 86.3 | 69.7 |
| MSAD (G2O0) | 80.7 | 81.4 | 78.7 | 66.1 | 74.2 | 75.6 | 80.2 | 46.7 | 93.9 | 54.2 | 91.8 | 83.7 | 89.3 | 62.8 | 85.4 | 80.2 | 84.5 | 64.3 | 76.3 | 34.3 | 34.9 | 2.0 | 88.3 | 71.7 | 86.5 | 70.9 |
| MSAD (G2O1) | 80.3 | 81.0 | 83.6 | 71.6 | 71.9 | 73.8 | 81.9 | 48.0 | 79.3 | 25.6 | 92.4 | 84.5 | 88.0 | 61.3 | 84.0 | 78.1 | 88.2 | 66.1 | 66.6 | 22.6 | 72.0 | 24.7 | 83.6 | 63.9 | 85.4 | 69.6 |
| MSAD (G2O2) | 74.6 | 80.9 | 81.0 | 71.1 | 75.9 | 79.1 | 88.1 | 57.0 | 70.3 | 23.5 | 80.3 | 72.5 | 88.7 | 67.2 | 78.8 | 75.9 | 79.7 | 61.7 | 81.0 | 56.6 | 74.6 | 35.9 | 84.6 | 69.8 | 83.5 | 70.5 |
| MSAD (G2O3) | 81.3 | 81.9 | 88.5 | 80.9 | 74.1 | 73.6 | 83.4 | 53.8 | 97.6 | 63.0 | 90.8 | 81.9 | 90.2 | 71.5 | 83.8 | 78.6 | 83.6 | 62.0 | 89.9 | 55.9 | 46.6 | 2.4 | 88.4 | 71.1 | 86.9 | 71.6 |
| MSAD (G2O4) | 79.6 | 80.5 | 87.5 | 73.2 | 76.5 | 75.7 | 88.9 | 57.1 | 89.9 | 21.4 | 90.2 | 82.2 | 87.7 | 65.9 | 84.0 | 76.8 | 85.6 | 62.7 | 93.7 | 72.5 | 60.4 | 3.3 | 82.8 | 62.5 | 86.7 | 70.7 |
| MSAD (G3O0) | 80.0 | 80.3 | 89.9 | 82.6 | 74.3 | 74.9 | 80.5 | 46.1 | 95.7 | 39.9 | 90.8 | 81.8 | 83.0 | 61.8 | 84.1 | 78.1 | 81.9 | 58.2 | 91.2 | 59.7 | 61.3 | 3.7 | 85.1 | 67.2 | 86.1 | 70.1 |
| MSAD (G3O1) | 80.7 | 81.4 | 89.0 | 80.3 | 74.8 | 74.7 | 96.0 | 44.7 | 96.0 | 42.3 | 88.4 | 79.5 | 85.1 | 54.6 | 85.0 | 80.0 | 84.4 | 60.3 | 88.9 | 48.2 | 73.2 | 4.7 | 83.6 | 63.9 | 86.5 | 70.4 |
| MSAD (G3O2) | 76.9 | 79.7 | 87.0 | 75.0 | 76.1 | 77.3 | 89.8 | 50.6 | 63.8 | 30.7 | 84.1 | 74.7 | 80.4 | 51.2 | 80.6 | 74.9 | 83.6 | 62.4 | 66.4 | 21.1 | 60.3 | 4.8 | 85.6 | 69.6 | 85.2 | 69.3 |
| MSAD (G3O3) | 80.3 | 80.2 | 68.5 | 64.6 | 72.3 | 74.4 | 81.6 | 47.8 | 85.9 | 14.3 | 91.1 | 83.4 | 85.0 | 55.3 | 86.1 | 81.5 | 82.8 | 60.7 | 94.0 | 73.1 | 56.0 | 4.4 | 79.2 | 59.5 | 85.2 | 69.7 |
| MSAD (G4O0) | 81.9 | 82.2 | 88.3 | 78.3 | 75.9 | 75.9 | 78.2 | 47.3 | 97.9 | 63.9 | 90.9 | 81.9 | 87.9 | 66.5 | 84.5 | 79.2 | 85.1 | 63.5 | 79.3 | 31.8 | 46.9 | 2.6 | 87.7 | 68.4 | 86.8 | 71.4 |
| MSAD (G4O1) | 77.3 | 81.5 | 89.6 | 79.2 | 75.6 | 77.4 | 90.9 | 59.0 | 82.3 | 32.9 | 83.2 | 75.0 | 84.2 | 54.5 | 79.2 | 74.1 | 83.3 | 65.2 | 76.1 | 25.8 | 57.6 | 23.2 | 83.4 | 68.2 | 85.1 | 70.3 |
| MSAD (G4O2) | 78.7 | 81.2 | 87.3 | 80.1 | 76.4 | 76.6 | 73.7 | 35.4 | 94.5 | 52.0 | 92.0 | 84.6 | 81.5 | 46.3 | 86.2 | 84.5 | 84.5 | 63.4 | 91.8 | 74.1 | 42.3 | 22.3 | 83.0 | 69.1 | 86.7 | 72.2 |
| MSAD (G5O0) | 74.8 | 80.8 | 83.3 | 73.3 | 74.1 | 77.0 | 86.8 | 56.7 | 85.9 | 28.9 | 82.7 | 75.8 | 85.0 | 60.6 | 78.7 | 74.5 | 82.1 | 63.3 | 61.2 | 19.1 | 87.0 | 26.8 | 80.8 | 65.4 | 83.0 | 68.7 |
| MSAD (G5O1) | 83.0 | 82.3 | 82.8 | 66.4 | 75.4 | 74.6 | 70.4 | 34.5 | 98.3 | 70.8 | 91.6 | 81.9 | 75.3 | 44.9 | 86.4 | 79.7 | 85.8 | 62.5 | 86.1 | 38.4 | 44.2 | 14.0 | 90.5 | 79.4 | 86.7 | 70.2 |
| MSAD (G6O0) | 79.5 | 80.4 | 72.9 | 62.8 | 75.9 | 76.1 | 78.8 | 42.2 | 97.4 | 65.5 | 92.1 | 83.4 | 84.6 | 57.3 | 85.4 | 78.6 | 84.8 | 61.4 | 86.4 | 41.5 | 76.1 | 6.3 | 90.7 | 80.0 | 87.2 | 70.6 |

Table 12: RTFM with G-OPL/OPL on MSAD (scenario-level). Scenario-wise results for different $GxOy$ configurations using RTFM on the MSAD dataset. The G1O0 setup consistently outperforms most other placements across scenarios.

| Configuration | MSAD | | ShT | | UCF | | CUHK | | Ped2 | |
|---|---|---|---|---|---|---|---|---|---|---|
| | AUC | AP | AUC | AP | AUC | AP | AUC | AP | AUC | AP |
| G1O0 | 88.0 | 70.9 | 97.3 | 74.7 | 78.3 | 30.9 | 84.9 | 66.2 | 85.0 | 71.4 |
| G1O1 | 84.4 | 68.6 | 96.9 | 72.3 | 78.1 | 34.3 | 84.1 | 64.3 | 81.4 | 69.0 |
| G1O2 | 86.9 | 69.8 | 97.0 | 70.5 | 76.5 | 35.7 | 79.7 | 60.3 | 80.0 | 67.9 |
| G1O3 | 86.5 | 69.5 | 96.9 | 72.7 | 72.6 | 30.9 | 84.3 | 66.1 | 82.2 | 71.4 |
| G1O4 | 87.5 | 71.1 | 96.5 | 72.3 | 69.3 | 27.2 | 83.2 | 62.0 | 84.7 | 74.2 |
| G1O5 | 86.2 | 67.6 | 96.4 | 71.7 | 73.0 | 42.7 | 83.3 | 64.8 | 84.2 | 71.8 |
| G2O0 | 84.9 | 69.7 | 97.1 | 73.8 | 75.7 | 29.0 | 82.9 | 64.2 | 79.3 | 67.6 |
| G2O1 | 85.6 | 68.9 | 96.6 | 72.5 | 77.7 | 43.4 | 82.9 | 64.4 | 82.1 | 73.5 |
| G2O2 | 83.7 | 68.3 | 97.1 | 73.5 | 76.2 | 26.3 | 82.7 | 61.8 | 78.4 | 65.7 |
| G2O3 | 87.4 | 71.4 | 97.0 | 73.1 | 72.3 | 30.6 | 82.7 | 62.4 | 84.2 | 70.6 |
| G2O4 | 84.8 | 68.3 | 96.4 | 71.7 | 75.8 | 24.3 | 81.2 | 62.5 | 85.0 | 74.2 |
| G3O0 | 86.2 | 71.1 | 96.6 | 73.5 | 74.4 | 33.2 | 82.8 | 63.2 | 78.1 | 65.9 |
| G3O1 | 82.4 | 68.5 | 96.7 | 72.9 | 75.3 | 23.8 | 84.3 | 65.8 | 88.0 | 73.9 |
| G3O2 | 85.2 | 70.7 | 97.0 | 73.2 | 75.7 | 37.3 | 81.8 | 62.2 | 77.9 | 66.6 |
| G3O3 | 85.5 | 70.4 | 96.3 | 73.0 | 75.0 | 29.0 | 64.4 | 82.8 | 83.1 | 71.9 |
| G4O0 | 86.4 | 69.0 | 96.5 | 72.4 | 76.5 | 26.4 | 63.8 | 82.6 | 78.5 | 67.8 |
| G4O1 | 82.5 | 70.2 | 96.5 | 72.4 | 76.7 | 27.8 | 61.5 | 81.5 | 84.4 | 70.5 |
| G4O2 | 85.9 | 70.4 | 97.0 | 72.1 | 71.3 | 24.5 | 62.0 | 82.3 | 81.3 | 68.5 |
| G5O0 | 84.5 | 71.1 | 97.3 | 73.5 | 77.7 | 35.0 | 63.9 | 82.5 | 89.6 | 75.0 |
| G5O1 | 86.3 | 66.3 | 96.7 | 73.5 | 71.1 | 27.7 | 67.3 | 84.0 | 87.0 | 81.2 |
| G6O0 | 83.6 | 69.2 | 97.0 | 71.5 | 74.0 | 27.0 | 66.5 | 83.3 | 81.9 | 75.3 |

Table 13: Performance of RTFM with G-OPL/OPL across five datasets. This table presents the results of various G-OPL and OPL layer combinations on five benchmark datasets using the RTFM model. The performance trends inform the selection of optimal layer configurations, as summarized in Table 4.

| Configuration | MSAD | | ShT | | UCF | | CUHK | | Ped2 | |
|---|---|---|---|---|---|---|---|---|---|---|
| | AUC | AP | AUC | AP | AUC | AP | AUC | AP | AUC | AP |
| G1O0 | 84.0 | 65.8 | 83.7 | 42.0 | 83.3 | 15.2 | 69.4 | 43.5 | 93.9 | 93.6 |
| G1O1 | 81.8 | 63.4 | 77.0 | 22.4 | 77.1 | 14.0 | 62.9 | 35.0 | 83.9 | 72.2 |
| G1O2 | 82.7 | 64.5 | 66.5 | 17.8 | 76.1 | 14.0 | 62.9 | 36.6 | 70.0 | 61.3 |
| G1O3 | 82.2 | 65.0 | 73.2 | 14.1 | 76.9 | 13.8 | 60.4 | 32.5 | 92.4 | 89.7 |
| G1O4 | 83.3 | 62.8 | 77.5 | 24.3 | 77.3 | 12.3 | 59.3 | 36.9 | 71.9 | 60.4 |
| G2O0 | 83.0 | 62.1 | 55.0 | 11.7 | 72.6 | 13.6 | 59.1 | 28.8 | 74.3 | 63.4 |
| G2O1 | 83.5 | 63.4 | 65.9 | 12.4 | 80.1 | 14.8 | 68.7 | 36.8 | 85.0 | 78.1 |
| G2O2 | 82.3 | 64.2 | 73.5 | 15.4 | 75.7 | 12.8 | 69.8 | 37.7 | 91.7 | 89.7 |
| G2O3 | 83.4 | 66.0 | 75.5 | 17.0 | 73.8 | 9.3 | 68.4 | 36.3 | 77.9 | 56.9 |
| G3O0 | 79.8 | 61.7 | 84.1 | 25.9 | 74.4 | 13.3 | 70.8 | 40.5 | 73.2 | 64.1 |
| G3O1 | 82.3 | 61.3 | 77.5 | 15.4 | 74.1 | 13.2 | 65.5 | 33.4 | 68.1 | 59.2 |
| G3O2 | 84.8 | 65.1 | 71.5 | 14.0 | 76.5 | 9.7 | 63.1 | 36.5 | 79.0 | 72.5 |
| G4O0 | 79.8 | 61.5 | 81.2 | 23.2 | 78.1 | 12.5 | 60.2 | 33.5 | 80.6 | 69.7 |
| G4O1 | 80.6 | 62.3 | 75.3 | 17.7 | 74.8 | 10.0 | 61.4 | 33.4 | 57.2 | 36.7 |
| G5O0 | 82.0 | 63.2 | 83.6 | 23.2 | 74.6 | 24.7 | 69.7 | 45.3 | 78.5 | 61.7 |

Table 14: Performance of MGFN with G-OPL/OPL across five datasets. This table presents the results of various G-OPL and OPL layer combinations on five benchmark datasets using the MGFN model. The performance trends inform the selection of optimal layer configurations, as summarized in Table 4.

$\lambda_{\text{face}}{=}3.0$ on UCF), while MGFN is more responsive to orthogonality regularization (*e.g.*, $\lambda_{\text{orth}}{=}5.0$ on Ped2).

This analysis highlights the importance of tuning these hyperparameters to adaptively control the trade-off between anomaly detection accuracy and the ethical goals of interpretability and privacy.

**Placement of G-OPL/OPL modules** In both RTFM and MGFN architectures, the integration of G-OPL and OPL follows well-defined constraints and placement rules based on the structure of each model.

For RTFM, there are six fixed insertion points for projection modules. The first G-OPL, if present, is always inserted before the aggregate module. Subsequent G-OPL and OPL modules can then be placed within the aggregate block, after each of its five convolutional layers. These positions are denoted as C1 through C5, where C1 refers to the insertion point after the first convolutional layer, C2 after the second, and so on up to C5. The total number of inserted modules must not exceed six, and OPLs can only be placed after at least one G-OPL has been inserted.

For MGFN, there are five predefined insertion points corresponding to the sequence of backbone and convolutional layers. These are labeled as B1, C1, B2, C2, and B3. Here, B1 is the position after the first Transformer block, C1 after the first convolutional layer, B2 after the second Transformer block, C2 after the second convolutional layer, and B3 after the third Transformer block. As in RTFM, the total number of projection layers must not exceed the available slots, and OPL modules can only follow G-OPL modules.

This consistent naming convention (*e.g.*, C1, B1) allows clear identification of module locations and enables the structured evaluation of different G-OPL and OPL configurations across experiments.

### A.11 LAYER PLACEMENT: PRACTICAL GUIDELINES AND EMPIRICAL INSIGHTS

We conduct comprehensive evaluations of OPL and G-OPL layer placements across five benchmark video anomaly detection (VAD) datasets, aiming to offer concrete and practical insights into how these components should be integrated into existing architectures. Our study examines both RTFM and MGFN models, considering performance impact, computational overhead, and sensitivity to different anomaly types and dataset characteristics.

**MGFN + OPL: minimal, effective integration.** For the MGFN model on the MSAD dataset, we find that incorporating a single OPL layer generally leads to the best overall performance. This holds across both anomaly types (Table 5) and evaluation scenarios (Table 6). While adding multiple OPL layers does not drastically reduce performance (typically within a 5% drop in AUC), it introduces additional learnable parameters, which could affect training stability and efficiency. These results suggest that a single, strategically placed OPL layer is often sufficient to suppress nuisance factors effectively without overcomplicating the model.

**RTFM + OPL: sensitive to overuse.** A similar pattern is observed in the RTFM model (Table 9 and 10). Here, too, the best performance is achieved when a single OPL layer is used, for instance, producing the highest AUC for MSAD (C1). However, adding additional OPL layers tends to slightly degrade performance, although still within a reasonable range. This indicates that overuse of OPL may suppress not only irrelevant information but also task-relevant signals, thus harming the anomaly detection capability.

**MGFN + G-OPL + OPL: flexible but layer-dependent.** When both G-OPL and OPL layers are used in the MGFN architecture, performance becomes more sensitive to layer arrangement (Table 7 and 8). The best AUC is achieved using three G-OPL layers followed by two OPL layers, whereas the best average precision (AP) is observed with two G-OPL layers followed by three OPL layers. Nonetheless, we find that using even a single G-OPL layer yields competitive performance while significantly reducing computational cost. This makes it a favorable option for real-world deployments where efficiency is crucial.

**RTFM + G-OPL: less sensitive to face suppression.** In contrast, for the RTFM model, the best performance with G-OPL is achieved when only one layer is added (Table 11 and 12). This suggests that facial information, used as a guiding attribute in G-OPL, is not strongly correlated with anomalous events in the RTFM pipeline, and that extensive suppression may inadvertently obscure useful

contextual cues. Thus, minimal semantic suppression proves to be the most effective approach in this setting.

**Dataset-specific trends and exceptions.** Across the remaining four datasets, ShanghaiTech, UCF-Crime, CUHK Avenue, and UCSD Ped2, we observe similar trends, particularly in the RTFM architecture (Table 13 and 14). A single G-OPL layer typically offers the best balance between privacy control and detection performance. However, Ped2 emerges as an exception. Due to its low resolution and limited facial visibility, facial signals are harder to detect, making it beneficial to stack multiple G-OPL layers to ensure more thorough suppression of residual sensitive information.

Our findings suggest that using a single OPL or G-OPL layer is often sufficient and optimal, offering strong performance with low overhead. Excessive stacking of projection layers can degrade detection performance by suppressing task-relevant signals, especially in architectures like RTFM. The optimal number of projection layers also depends on dataset characteristics such as resolution and the clarity of sensitive attributes. These results offer valuable guidance for effectively and efficiently incorporating privacy-preserving and interpretable layers into VAD architectures.

### A.12 LIMITATIONS

While our proposed method demonstrates strong and consistent improvements across diverse datasets and anomaly types, several limitations remain.

First, the performance gains heavily rely on carefully tuning the placement and frequency of the disentanglement modules (G-OPL and OPL). Although we provide empirical insights into effective configurations, these may not generalize optimally to unseen domains without additional validation or adaptation.

Second, our method exhibits less stability on datasets characterized by high scene diversity or subtle anomaly cues, such as UCF-Crime. This suggests that while our approach effectively filters irrelevant signals, it may still struggle in scenarios where anomaly patterns are highly variable or context-dependent.

Third, the disentanglement process, while beneficial for suppressing task-irrelevant information, introduces additional architectural complexity and computational overhead. This may limit the scalability of our approach for real-time or resource-constrained applications.

Finally, while our focus on privacy-preserving datasets highlights the robustness of our method, its effectiveness on privacy-unconstrained, high-resolution datasets remains less explored. Future work could investigate the interplay between disentanglement, resolution, and anomaly characteristics in broader settings.

### A.13 LLM USAGE DECLARATION

We disclose the use of Large Language Models (LLMs) as general-purpose assistive tools during the preparation of this manuscript. LLMs were used only for minor tasks such as grammar and style improvement, code verification, and formatting suggestions. No scientific ideas, analyses, experimental designs, or conclusions were generated by LLMs. All core research, methodology, experiments, and results were performed and fully verified by the authors.

The authors take full responsibility for all content presented in this paper, including text or code suggestions that were refined with the assistance of LLMs. No content generated by LLMs was treated as original scientific work, and all references and claims have been independently verified. LLMs did not contribute in a manner that would qualify them for authorship.

