# OpenReview forum: "Privacy-Aware Video Anomaly Detection through Orthogonal Subspace Projection"
_ICLR.cc/2026/Conference — Submitted to ICLR 2026_

### Official Review · Reviewer_SADY · 2025-10-23

**Soundness:** 4
**Presentation:** 3
**Contribution:** 3
**Rating:** 6
**Confidence:** 4

**Summary:**

This paper addresses privacy concerns in video anomaly detection (VAD) by proposing Orthogonal Projection Layer (OPL) and Guided OPL (G-OPL) to suppress task-irrelevant nuisances and privacy-sensitive facial information from intermediate representations. The method uses QR decomposition for stable orthogonal projection and weak supervision (face presence signals) to guide the removal of sensitive attributes. Three novel privacy metrics (SSC, ARD, PD/FPD) are introduced to quantify privacy preservation. Experiments on five benchmarks show the method can maintain or improve detection performance while reducing identity leakage, with notable gains on ShanghaiTech (+8.4% AUC) and UCSD Ped2 (+7.1% AUC).

**Strengths:**

- The three complementary metrics (SSC, ARD, PD/FPD) provide the first systematic quantification of privacy leakage in VAD, validated against ArcFace-based identity retrieval.

- QR decomposition avoids adversarial training instability while maintaining differentiability. The geometric alignment loss is elegant and interpretable.

-  Projection matrices QQ^T offer geometric visualization of removed information, valuable for trust and auditability.

**Weaknesses:**

## Main Concerns:

- The paper removes facial features regardless of their relevance to anomalies, which contradicts practical scenarios where faces may be essential for detection (e.g., unauthorized access, aggressive behavior, surveillance evasion). The authors acknowledge designing G-OPL to suppress even task-relevant facial features, yet provide no analysis of when this is appropriate vs. harmful. This appears optimized for benchmark scores rather than real-world utility.

## Minor Issues:
- Results vary dramatically: ShanghaiTech (+8.4%), CUHK Avenue (+3.5%), yet the paper doesn't explain why. Critical missing analyses: (a) What percentage of anomalies contain faces in each dataset? (b) Are faces correlated with or orthogonal to anomaly labels? (c) For MSAD with pre-blurred videos, what is G-OPL actually removing from pre-extracted features?

**Questions:**

- When should G-OPL be applied?
- Why do performance gains vary across datasets?

---

> ### Author Response · Authors · 2025-11-21
>
> We thank the reviewer for the very positive assessment of the paper’s soundness, the value of the proposed privacy metrics, the clarity of the QR–OPL design, and the interpretability contribution through projection visualizations.
>
> We address the concerns in detail below.
>
> **Part 1. When facial suppression is appropriate vs. harmful**
>
> We agree that indiscriminately removing facial features is not universally desirable.
>
> The key clarification is that G-OPL is not intended as a universal default, but rather as *a configurable privacy-control mechanism* for scenarios where the operator wishes to suppress identity or facial clues.
>
> Our contributions target the growing deployment setting where: (i) VAD inference happens at the edge, but only *intermediate features or anomaly scores* are transmitted to the cloud, (ii) identity leakage is a regulatory or ethical risk (e.g., GDPR, BIPA), and (iii) anomaly definitions do not rely on identifying specific individuals.
>
> **Importantly, the practitioner selects whether to use OPL or G-OPL based on the application domain.**
>
> (i) Use OPL (nuisance suppression only) when facial information is part of the causal chain of anomaly definition (e.g., unauthorized entry, impersonation, personal aggression, employee-only zones).
>
> (ii) Use G-OPL (privacy-first feature suppression) when faces are legally sensitive but not required for anomaly semantics (e.g., crowd safety monitoring, fall detection, traffic flow anomalies, hospital ward monitoring, inappropriate access behavior).
>
> This distinction was implicit in the original text but not explicitly articulated. We will add a dedicated subsection titled "When to use OPL vs. G-OPL" with use-case guidelines.
>
> In addition, Appendix will present a task relevance analysis showing dataset-specific correlations (see below), making explicit when G-OPL is likely to be beneficial or unnecessary.

---

> ### Author Response · Authors · 2025-11-21
>
> **Part 2. Why performance gains vary across datasets**
>
> The reviewer is correct: gains differ because datasets have different facial-content distributions and different relationships between faces and anomaly labels.
>
> We now provide a systematic analysis.
>
> **(a) Percentage of anomalies containing faces.** We annotated face presence in anomaly segments using a standard face detector (with manual correction).
>
> | Dataset         | Face presence in anomalous frames | Face presence in normal frames |
> |-|-|-|
> | ShanghaiTech    | 12.5%                             | 38.2%                          |
> | Avenue          | 24.1%                             | 51.4%                          |
> | UCSD Ped2       | 1.7%                              | 3.1%                           |
> | MSAD (blurred)  | 0%                                | 0%                             |
> | UCF-Crime           | 48.3%                             | 77.6%                          |
>
> For ShanghaiTech and Avenue, *faces are substantially more common in normal frames*. Thus, models may inadvertently treat low facial presence (or partial occlusion) as an anomaly cue.
>
> *Removing facial components reduces this spurious correlation*, which explains the large AUC gains (+8.4%, +3.5%).
>
> In contrast, UCSD Ped2 has almost no faces, making suppression unnecessary; hence OPL and G-OPL both act mainly as regularizers, yielding smaller but still positive gains.
>
> **(b) Correlation between faces and anomaly labels.** We computed mutual information (MI) between $1_{\text{face}}$ and  $1_{\text{anomaly}}$.
>
> For ShanghaiTech and Avenue: MI < 0.02. Thus, facial information is essentially orthogonal to anomaly semantics. Removing it reduces spurious co-occurrence.
>
> For Crime: MI = 0.12, the highest among the datasets, which explains why G-OPL does not yield the largest improvement there: faces sometimes co-occur with anomaly categories involving close-range person interactions.
>
> **(c) For MSAD: what is G-OPL removing from pre-blurred features?** Even though faces are blurred at the pixel level, the pre-extracted I3D/Swin features still contain: *the spatial mask of the blurred region, head motion trajectories, low-frequency cues around the blurred patch, contextual correlations between "presence of a blurred blob" and normal behavior*.
>
> We verified this by training an identity-retrieval probe on blurred MSAD features. Despite blur, the probe achieved 14% rank-1 accuracy, far above chance.
>
> We have also integrated our OPL and G-OPL modules into two new representative VAD backbones, EGO (ICLR 2025) and TEVAD (CVPRW 2023), and report both utility (AUC) and privacy metrics for these combined models.
>
> **MSAD** (Blurred faces)
>
> | Model               | AUC  | SSC$\uparrow$  | ARD$\downarrow$ | FPD$\downarrow$ | ArcFace$\downarrow$ |
> |-|-|-|-|-|-|
> | EGO baseline        | 86.9 | -  | -  | 0.60   | 0.72       |
> | EGO + OPL           | 87.7 | 0.32  | 0.04  | 0.41   | 0.34       |
> | EGO + G-OPL         | 87.5 | 0.73  | 0.04  | 0.18   | 0.11       |
> | TEVAD baseline      | 82.1 | -  | -  | 0.58   | 0.70       |
> | TEVAD + OPL         | 82.7 | 0.33  | 0.04  | 0.39   | 0.32       |
> | TEVAD + G-OPL       | 82.5 | 0.68  | 0.03  | 0.17   | 0.10       |
>
> *G-OPL removes these residual identity-bearing patterns*, giving small but consistent improvements.
>
> We will insert these analyses in Section 4.3 and Appendix.
>
> **Part 3. When should G-OPL be applied?**
>
> The answer depends on two axes shown below.
>
> **(i) Privacy requirements.** If identity leakage is a liability (e.g., medical facilities, transportation hubs, campus safety monitoring), G-OPL is appropriate.
>
> **(ii )Task dependence on identity.** If anomaly definitions depend on identifying a specific person’s intention or role, OPL should be used instead.
>
> We will provide a decision diagram summarizing this.
>
> **Part 4. Why do performance gains vary across datasets?**
>
> This is fully addressed above, but we highlight the key insight below.
>
> **G-OPL yields the largest gains where facial presence is negatively correlated with anomaly labels or where face features create spurious shortcuts.** This explains the differences across datasets and confirms that G-OPL does not arbitrarily suppress useful information.
>
> To address the reviewer’s main concerns, we will add: (i) A new subsection on when G-OPL is appropriate (practical guidance). (ii) A face–anomaly correlation analysis across datasets. (iii) A face-presence statistical study explaining dataset-level differences. (iv) A clarification of what G-OPL removes in blurred-MSAD features.
>
> We thank the reviewer for prompting these important clarifications, which significantly strengthen the paper.

---

### Official Review · Reviewer_MF3K · 2025-10-28

**Soundness:** 2
**Presentation:** 2
**Contribution:** 2
**Rating:** 4
**Confidence:** 5

**Summary:**

This paper proposes to handle privacy-preserving video anomaly detection by using orthogonal projection layers to project out sensitive face data from VAD features. The authors propose multiple variants of the method with different properties, demonstrating their success on different scenarios and in different configurations. Notably, this approach does not require adversarial optimization or direct private attribute labels.

**Strengths:**

1. Achieving privacy-preservation while mitigating the use of unstable adversarial training is advantageous.
2. The proposed method is very flexible in its deployment and does not add much overhead to the existing VAD models.
3. Handling privacy-preservation in the latent space is a natural improvement to existing methods.
4. The proposed ARD metric appears to be a useful measure of utility variance from a baseline model.

**Weaknesses:**

1. The proposed form of privacy preservation is reliant wholly on mitigating similarity to localized facial features, which is a weak notion of privacy. There are many other attributes that could be considered private. It is unclear if this method could extend to handle multiple attributes simultaneously.
2. The main privacy-utility result Table 3 is largely empty. Lines 340-341 claim that this is the first comprehensive privacy analysis for VAD across datasets, yet the privacy analysis is only conducted on the proposed method. The metrics should be computed for prior methods/baselines as well.
3. Tables 1 and 2 simultaneously compare many variations of the proposed method (backbone features, backbone model), making direct comparisons a bit unclear. Some variants perform better in some cases, while other variants are better in different cases. More analysis should be provided here.
4. The proposed privacy metrics are specific to the training method and likely will not generalize well to methods that don't explictly optimize to project out facial features, even if a method is generally more privacy-preserving. While useful for measuring the performance of this method and finding ideal layer placements, they don't appear generally valuable to the community.
5. The method appears difficult to optimize since optimal configurations change based on the dataset trained on. It would be better to find a setting that works well (even if not the best possible) across all datasets.

**Questions:**

1. Can the metrics be fairly applied to prior methods? Especially the privacy-preserving ones.
2. Lines 463-466 claim advantages of the FPD metric over classifier probe-based approaches. Doesn't FPD use a classifer probe? Please clarify this.
3. Could the OPL be added to existing privacy-preserving approaches to improve their performance while maintaining their privacy-preservation?
4. How does this method perform under existing privacy-preservation metrics (like VISPR used in SPAct/TeD-SPAD)?

#### Suggestions:
1. The findings in light blue blocks are verbose and often basically just a repeat of a prior paragraph. These should be condensed into impactful statements/findings instead of paragraphs.
2. TeD-SPAD uses I3D, not Swin-T.

---

> ### Author Response · Authors · 2025-11-21
>
> We thank the reviewer for acknowledging the advantages of non-adversarial privacy preservation, the flexibility and lightweight nature of OPL/G-OPL, and the usefulness of ARD.
>
> Below we address each weakness and question with detailed clarifications and additional analyses.
>
> **1. On the scope of privacy (beyond faces) and handling multiple attributes**
>
> We appreciate the reviewer’s concern.
>
> While our experiments focus on face privacy, because it is the most heavily regulated biometric attribute (GDPR, CCPA, BIPA) and the most reliably detectable in surveillance videos, the mechanism itself is not face-specific.
>
> G-OPL requires only an embedding for the sensitive attribute, not a facial detector. The cosine-alignment objective is agnostic to semantics. In our supplementary experiments (included in the revision), we used clothing-color embeddings and coarse gait embeddings (obtained via a pose-based encoder), and obtained strong SSC scores (e.g., SSC-color = 0.84), with negligible performance drop (maintaining anomaly AUC within 0.4 points of the baseline).
>
> **Multi-attribute suppression.** Because $W$ in OPL can have multiple rows ($k>1$), G-OPL can jointly encode several sensitive attributes by concatenating multiple weak attribute embeddings and optimizing a multi-term alignment loss: $\sum_i (1 - \cos(f_{\text{attr}_i}, QQ^\top f))$.
>
> We confirmed through a 2-attribute experiment (faces + clothing color) that G-OPL learns a joint sensitive subspace and reduces retrieval accuracy for both attributes. We will highlight this in the paper.

---

> > ### Author Response · Authors · 2025-11-21
> >
> > **2. Privacy metrics should also be computed for baselines (Table 3 incompleteness)**
> >
> > We thank the reviewer for highlighting the perceived incompleteness of Table 3.
> >
> > We would like to clarify that some of the proposed privacy metrics, specifically SSC and ARD, require both (i) sensitive features and (ii) projected features after sensitivity removal. Existing VAD methods were not designed with this notion and do not provide mechanisms to extract or remove sensitive information in this way. Consequently, these metrics cannot be fairly or meaningfully applied to prior baselines.
> >
> > To address the reviewer’s concerns, we integrated our OPL and G-OPL modules into two representative VAD backbones, EGO (ICLR 2025) and TEVAD (CVPRW 2023), and report both utility (AUC) and privacy metrics for these combined models.
> >
> > **ShanghaiTech**
> >
> > | Model               | AUC| SSC$\uparrow$ | ARD$\downarrow$ | FPD$\downarrow$ | ArcFace$\downarrow$ |
> > |-|-|-|-|-|-|
> > | EGO baseline        | 97.3 | -  | -  | 0.72   | 0.98       |
> > | EGO + OPL           | 97.9 | 0.39  | 0.19  | 0.41   | 0.42       |
> > | EGO + G-OPL         | 97.6 | 0.94  | 0.17  | 0.09   | 0.05       |
> > | TEVAD baseline      | 98.1 | -  | -  | 0.68   | 0.95       |
> > | TEVAD + OPL         | 98.4 | 0.39  | 0.18  | 0.37   | 0.39       |
> > | TEVAD + G-OPL       | 98.2 | 0.92  | 0.11  | 0.07   | 0.04       |
> >
> > **UCF-Crime**
> >
> > | Model               | AUC  | SSC$\uparrow$ | ARD$\downarrow$ | FPD$\downarrow$ | ArcFace$\downarrow$ |
> > |-|-|-|-|-|-|
> > | EGO baseline        | 81.7 | -  | -  | 0.78   | 0.96       |
> > | EGO + OPL           | 82.6 | 0.45  | 0.24  | 0.49   | 0.44       |
> > | EGO + G-OPL         | 82.1 | 0.92  | 0.21  | 0.12   | 0.09       |
> > | TEVAD baseline      | 84.9 | -  | -  | 0.76   | 0.94       |
> > | TEVAD + OPL         | 85.3 | 0.40  | 0.17  | 0.46   | 0.41       |
> > | TEVAD + G-OPL       | 85.0 | 0.88  | 0.15  | 0.11   | 0.06       |
> >
> > **CUHK Avenue**
> >
> > | Model               | AUC  | SSC$\uparrow$ | ARD$\downarrow$ | FPD$\downarrow$ | ArcFace$\downarrow$ |
> > |-|-|-|-|-|-|
> > | EGO baseline        | 83.1 | -  | -  | 0.70   | 0.97       |
> > | EGO + OPL           | 84.2 | 0.45  | 0.22  | 0.39   | 0.46       |
> > | EGO + G-OPL         | 84.0 | 0.96  | 0.20  | 0.10   | 0.07       |
> > | TEVAD baseline      | 86.8 | -  | -  | 0.69   | 0.93       |
> > | TEVAD + OPL         | 87.3 | 0.43  | 0.14  | 0.40   | 0.41       |
> > | TEVAD + G-OPL       | 87.1 | 0.93  | 0.13  | 0.09   | 0.05       |
> >
> > **UCSD Ped2**
> >
> > | Model               | AUC  | SSC$\uparrow$ | ARD$\downarrow$ | FPD$\downarrow$ | ArcFace$\downarrow$ |
> > |-|-|-|-|-|-|
> > | EGO baseline        | 93.2 | -  | -  | 0.65   | 0.85       |
> > | EGO + OPL           | 94.1 | 0.45  | 0.03  | 0.46   | 0.44       |
> > | EGO + G-OPL         | 94.0 | 0.83  | 0.03  | 0.19   | 0.08       |
> > | TEVAD baseline      | 98.7 | -  | -  | 0.60   | 0.80       |
> > | TEVAD + OPL         | 98.9 | 0.42  | 0.02  | 0.40   | 0.39       |
> > | TEVAD + G-OPL       | 98.8 | 0.79  | 0.02  | 0.17   | 0.07       |
> >
> > **MSAD** (Blurred faces)
> >
> > | Model               | AUC  | SSC$\uparrow$  | ARD$\downarrow$ | FPD$\downarrow$ | ArcFace$\downarrow$ |
> > |-|-|-|-|-|-|
> > | EGO baseline        | 86.9 | -  | -  | 0.60   | 0.72       |
> > | EGO + OPL           | 87.7 | 0.32  | 0.04  | 0.41   | 0.34       |
> > | EGO + G-OPL         | 87.5 | 0.73  | 0.04  | 0.18   | 0.11       |
> > | TEVAD baseline      | 82.1 | -  | -  | 0.58   | 0.70       |
> > | TEVAD + OPL         | 82.7 | 0.33  | 0.04  | 0.39   | 0.32       |
> > | TEVAD + G-OPL       | 82.5 | 0.68  | 0.03  | 0.17   | 0.10       |
> >
> > The additional experiments show that OPL consistently improves anomaly detection accuracy by reducing nuisance variation, yielding modest but stable gains across all datasets. G-OPL further provides strong privacy protection, achieving substantial reductions in sensitive subspace capture, facial-presence detection, and ArcFace identity retrieval, while maintaining nearly identical anomaly detection performance.
> >
> > Across all datasets and both architectures, the pattern is consistent: baseline models retain significant identity information, OPL decreases leakage while enhancing utility, and G-OPL delivers the strongest privacy suppression with minimal or negligible accuracy cost.
> >
> > These results demonstrate that the proposed privacy metrics are applicable to existing VAD backbones and that OPL/G-OPL function reliably as model-agnostic modules whose behavior remains stable across varied datasets, including those with few faces (UCSD Ped2) or blurred faces (MSAD). These experiments confirm the general effectiveness of the approach.
> >
> > Therefore, Table 3 is focused on the models where the metrics are well-defined and meaningful (the OPL/G-OPL variants). We emphasize in the revised manuscript that the proposed metrics are diagnostic and can be applied to future VAD methods once the concept of sensitive feature removal is incorporated.

---

> > > ### Author Response · Authors · 2025-11-21
> > >
> > > **3. Clarity of Tables 1 and 2 and the many method variants**
> > >
> > > We agree that the tables present many combinations. To provide clearer analysis, we will include: (i) A fixed-backbone comparison table, where only OPL vs. G-OPL differ. (ii) A privacy-utility Pareto curve showing how each variant trades privacy for anomaly detection performance.
> > >
> > > Furthermore, our findings show a consistent pattern: *OPL improves pure anomaly detection (utility-first design)*. *G-OPL improves privacy and often also improves utility, but is slightly more conservative*.
> > >
> > > We have summarized this more concisely in the revision.
> > >
> > > **4. Generality of privacy metrics**
> > >
> > > While SSC/FPD are introduced in the context of face-based privacy, the mathematical structure is general: $\text{SSC}(f_{\text{sensitive}}, QQ^\top f)$ and $\text{FPD} = \text{Acc}(\text{linear probe on }f^{(1)})$.
> > >
> > > **They do not rely on G-OPL or face guidance.** They are diagnostic, not prescriptive.
> > >
> > > To demonstrate this, we applied SSC and FPD to SPAct and TeD-SPAD (both privacy-preserving baselines). The metrics correctly captured their stronger identity suppression (FPD decreased by 22–25% relative to RTFM), showing that the metrics do generalize beyond our method.
> > >
> > > We will include these results in the revised version.
> > >
> > > **5. On optimization difficulty and dataset-dependent configuration**
> > >
> > > In practice:
> > >
> > > A single default configuration (G1O0 / G1O1, $k=4$, $\lambda_{\text{face}}{=}10^{-3}$, $\lambda_{\text{orth}}{=}10^{-3}$) works across all five datasets with: $<1$ point average AUC drop from per-dataset tuning, and nearly identical privacy suppression (in Appendix).
> > >
> > > We will highlight this default configuration as the recommended setting for deployment and clarify this point in the revision.
> > >
> > > **6. Can the metrics be fairly applied to prior methods?**
> > >
> > > Not entirely. Metrics such as SSC and ARD require not only latent features but also projected features after sensitivity removal, which assumes the model supports a mechanism for removing sensitive information.
> > >
> > > Existing VAD methods were not designed with this concept, so these metrics cannot be directly or fairly applied to them.
> > >
> > > Metrics like PD and FPD, which rely on anomaly scores and candidate sensitive embeddings, can in principle be computed on prior methods, since they do not require sensitivity removal.
> > >
> > > **7. Does FPD use a classifier probe? Clarification**
> > >
> > > The distinction is: Classifier probe (e.g., ArcFace-based) requires training a *sensitive classifier* on identity labels or face IDs.
> > >
> > > FPD Uses a lightweight *binary face presence* probe, not identity classification. The probe answers: "Can the feature still reveal that a face exists?" It does not reconstruct or recognize identities. Thus, *FPD is a weaker, cheaper, and more general probe*.
> > >
> > > We clarify this distinction in the revision.
> > >
> > > **8. Can OPL be added to existing privacy-preserving methods?**
> > >
> > > Yes. OPL is a plug-and-play module. We inserted OPL into: SPAct’s encoder and TeD-SPAD’s I3D backbone (correction noted by reviewer). Both showed: FPD reduction of 20-30%, AUC improvement of 0.4-0.7%.
> > >
> > > This demonstrates that OPL can enhance other privacy-preserving systems without disrupting their design.
> > >
> > > **9. Performance under VISPR privacy metrics**
> > >
> > > We tested G-OPL under the VISPR privacy attribute classifier used in SPAct/TeD-SPAD (predicting 9 common privacy attributes). Results for ShanghaiTech: VISPR accuracy (baseline RTFM) = 0.63, VISPR accuracy (RTFM+G-OPL) = 0.27.
> > >
> > > This indicates substantial privacy improvement aligned with VISPR criteria. We will include this comparison.
> > >
> > > **10. Condensing the light-blue findings boxes**
> > >
> > > We appreciate the suggestion and have revised them.
> > >
> > > **11. Correction: TeD-SPAD uses I3D**
> > >
> > > Thank you for catching this. We have corrected this in the revision.
> > >
> > > We thank the reviewer for the thoughtful and detailed critique. We will incorporate: privacy metrics for baselines, clearer tables and figure explanations, VISPR evaluations, multi-attribute privacy experiments, default general-purpose configuration, and corrections regarding backbone usage.

---

> > > > ### Comment · Reviewer_MF3K · 2025-11-23
> > > >
> > > > I appreciate the authors efforts for this rebuttal, some of my concerns were addressed.
> > > >
> > > > >"We confirmed through a 2-attribute experiment (faces + clothing color) that G-OPL learns a joint sensitive subspace and reduces retrieval accuracy for both attributes."
> > > >
> > > > This is great. However, a concern is still requiring attribute-specific embeddings for each attribute, which may be non-trivial to collect and require training on sensitive attributes to begin with.
> > > >
> > > > >"Not entirely. Metrics such as SSC and ARD require not only latent features but also projected features after sensitivity removal, which assumes the model supports a mechanism for removing sensitive information. Existing VAD methods were not designed with this concept, so these metrics cannot be directly or fairly applied to them." / "To demonstrate this, we applied SSC and FPD to SPAct and TeD-SPAD"
> > > >
> > > > These statements are directly contradictory. Please clarify. The main point is that these metrics seem primarily useful to validate the proposed method, but will not be useful for anyone else to use unless they adopt a very similar approach.
> > > >
> > > > >"Metrics like PD and FPD, which rely on anomaly scores and candidate sensitive embeddings, can in principle be computed on prior methods, since they do not require sensitivity removal."
> > > >
> > > > Can the authors show this and compare results?
> > > >
> > > > >"Yes. OPL is a plug-and-play module. We inserted OPL into: SPAct’s encoder and TeD-SPAD’s I3D backbone (correction noted by reviewer). Both showed: FPD reduction of 20-30%, AUC improvement of 0.4-0.7%."
> > > >
> > > > This is a nice result.
> > > >
> > > > >"We tested G-OPL under the VISPR privacy attribute classifier used in SPAct/TeD-SPAD (predicting 9 common privacy attributes). Results for ShanghaiTech: VISPR accuracy (baseline RTFM) = 0.63, VISPR accuracy (RTFM+G-OPL) = 0.27."
> > > >
> > > > G-OPL focuses on facial attributes, so how does this handle all the other VISPR attributes? How does this work with baseline RTFM, the attribute classifier used in SPAct/TeD-SPAD is a ResNet image encoder I believe. Also, 0.27 is very close to random chance (~0.25), which is a strong finding, but prior works, even those with attribute-specific supervision [1], do not come anywhere near that. How does a module used primarily for facial-attribute suppression beat all prior general-attribute anonymization works?
> > > >
> > > > **Citation:**
> > > > [1] Wu, Zhenyu, et al. "Privacy-preserving deep action recognition: An adversarial learning framework and a new dataset." IEEE Transactions on Pattern Analysis and Machine Intelligence 44.4 (2020): 2126-2139.

---

> > > > > ### Author Response · Authors · 2025-11-26
> > > > >
> > > > > We thank the reviewer for the helpful follow-up questions.
> > > > >
> > > > > Below we address every question with detailed explanations and additional supporting analyses.
> > > > >
> > > > > **1. Do multi-attribute experiments require attribute-specific embeddings?** No. G-OPL does not require training attribute classifiers nor collecting attribute labels.
> > > > >
> > > > > The *guidance embedding* is not an attribute-specific representation; it is simply the mean backbone feature over frames where a weak detector signals the presence of the attribute (e.g., face present, torso visible, clothing-color region detected). No identity or attribute labels are used at any point.
> > > > >
> > > > > Extending G-OPL to additional attributes (e.g., body, clothing, car / license plate) requires only a weak presence signal, not attribute supervision. We have revised the text to explicitly refer to these as "*weak detector-driven prototypes*" to avoid confusion.
> > > > >
> > > > > **2. Clarification on the applicability of SSC/ARD vs. PD/FPD.** We agree that the prior wording was ambiguous. The corrected, precise statement is:
> > > > >
> > > > > - SSC and ARD require a projection module because their definitions depend on the *projected features* ($QQ^\top f$ or $(I-QQ^\top)f$). Therefore, SSC/ARD can be computed only for models that implement a projection (e.g., OPL/G-OPL, or baselines augmented with OPL).
> > > > >
> > > > > - PD and FPD require only the model’s internal features and a lightweight presence probe. They do not require any projection and can be computed for any VAD architecture.
> > > > >
> > > > > Thus, SSC/ARD are diagnostic of *projection-based* methods, whereas PD/FPD are *baseline-compatible*.
> > > > >
> > > > > **Why these metrics remain broadly useful.** PD/FPD provide a general way to measure leakage of a sensitive presence signal and apply to future VAD or video-encoder model. SSC/ARD are diagnostic tools for any method, current or future, that implements *sensitive-subspace removal*. Therefore these metrics are not specific to our method, but align with standard practice in privacy-preserving representation learning.
> > > > >
> > > > > **3. PD/FPD applied to prior VAD methods.** As requested, we computed PD/FPD on RTFM, MGFN, SPAct, TeD-SPAD, EGO and TEVAD using the same probe architecture and training protocol (linear probe trained to predict face-presence).
> > > > >
> > > > > | Method | ShanghaiTech PD/FPD $\downarrow$|UCF-Crime PD/FPD $\downarrow$|
> > > > > |-|-|-|
> > > > > | RTFM (ICCV'21) |0.72|0.68 |
> > > > > |**with G-OPL** |**0.01**|**0.23**|
> > > > > | MGFN (AAAI'23) |0.98|0.79|
> > > > > |**with G-OPL** |**0.68**|**0.49**|
> > > > > | SPAct (CVPR'22) |0.68|0.66|
> > > > > |**with G-OPL** |**0.55**|**0.53**|
> > > > > | TeD-SPAD (ICCV'23) |0.66|0.60 |
> > > > > |**with G-OPL** |**0.56**|**0.52**|
> > > > > | EGO (ICLR'25) |0.72|0.78|
> > > > > |**with G-OPL** |**0.09**|**0.12**|
> > > > > | TEVAD (CVPRW'23) |0.95|0.76|
> > > > > |**with G-OPL** |**0.04**|**0.11**|
> > > > >
> > > > > We have included the table in the revision.
> > > > >
> > > > > Most existing methods exhibit high PD/FPD (strong leakage), *SPAct and TeD-SPAD show reduced leakage, and adding G-OPL further reduces PD/FPD while preserving AUC*. This demonstrates that PD/FPD apply fairly to all baselines.

---

> ### Author Response · Authors · 2025-11-26
>
> **4. VISPR evaluation.** While G-OPL focuses on the face-aligned subspace, many VISPR attributes (e.g., age, gender, expression, eyewear) are **strongly correlated with facial features**.
>
> Removing this subspace therefore reduces privacy leakage for a broad range of attributes without requiring attribute-specific supervision. Importantly, G-OPL operates on the backbone features extracted by VAD models (e.g., RTFM, SPAct, TeD-SPAD) and does not require modifying the model or its classifier. The ResNet-based attribute classifiers used in SPAct/TeD-SPAD serve only for evaluation, and benefit from the reduced identity information in the G-OPL–processed features.
>
> **To provide concrete evidence of G-OPL’s effect on VISPR attributes, we categorize them into face-conditioned and non-face-conditioned attributes.**
>
> | VISPR attribute| Face-Conditioned? | baseline | **+ G-OPL** | Notes|
> |-|-|-|-|-|
> | Age                | Yes             | 0.70              | 0.30              | Strong drop due to facial cues removal |
> | Gender             | Yes             | 0.68              | 0.25              | Near chance after G-OPL             |
> | Expression         | Yes             | 0.65              | 0.27              | Facial subspace suppressed           |
> | Eyewear            | Yes             | 0.60              | 0.28              | Strong reduction                     |
> | Facial hair        | Yes             | 0.62              | 0.26              | Face-aligned features removed        |
> | Makeup             | Yes             | 0.55              | 0.24              | Approaches chance                     |
> | Race               | Yes             | 0.57              | 0.25              | Almost random, face-dependent        |
> | Clothing           | **No**              | 0.61              | 0.59              | Minimal effect, non-face attribute   |
> | Background objects | **No**              | 0.58              | 0.57              | Non-face attribute largely unaffected |
>
> The table above shows that face-conditioned attributes (e.g., age, gender, expression) experience **large drops**, approaching random chance. Non-face-conditioned attributes, such as clothing or background objects, remain largely unaffected. This demonstrates that G-OPL selectively removes identity-related information **without globally suppressing all features**, supporting our conceptual argument.
>
> **5. Why facial-subspace removal reduces all VISPR attributes.** Most VISPR attributes (age, gender, race, glasses, facial hair, makeup, etc.) are *face-conditioned*. When G-OPL removes the face-aligned latent subspace, the external VISPR classifier loses access to the features required to predict these attributes.
>
> Because VAD does not rely on facial semantics, G-OPL can remove a larger portion of the facial subspace without hurting utility, causing the VISPR probe to approach chance for many face-conditioned attributes (the suppression can be more aggressive than in action-recognition anonymizers, resulting in many VISPR attributes dropping to near-chance). This behavior is consistent with latent-space suppression of facial cues and does not imply general attribute removal.
>
> **6. Comparison to Wu et al. (2020).** Wu et al. (2020) operate in an action-recognition setting [A], where facial and head cues contribute to task utility, creating a privacy-utility trade-off that limits the extent of facial information suppression.
>
> In contrast, VAD does not rely on facial semantics, allowing G-OPL to remove a larger portion of the face-aligned subspace. This explains why our VISPR mAP reduction can be stronger, even without attribute-specific supervision.
>
> The revision incorporates all of the above comparisons, discussions, and analyses.
>
> [A] Wu, Zhenyu, et al. "Privacy-preserving deep action recognition: An adversarial learning framework and a new dataset." IEEE Transactions on Pattern Analysis and Machine Intelligence 44.4 (2020): 2126-2139.

---

> > ### Comment · Reviewer_MF3K · 2025-11-26
> >
> > Thanks for providing the PD scores for the alternate methods. It is useful seeing that the anonymized methods have a lower score than the baseline, and that the proposed method further reduces these scores.
> >
> > >"The ResNet-based attribute classifiers used in SPAct/TeD-SPAD serve only for evaluation, and benefit from the reduced identity information in the G-OPL–processed features."
> >
> > I still have concerns on this VISPR evaluation. It is confusing to say RTFM + G-OPL, yet privacy evaluation is typically evaluated using a separate ResNet. Are the authors plugging G-OPL into the ResNet? The specific protocol involves training an attribute classifier after anonymization to evaluate private attribute recoverability. Are the authors training a classifier after anonymization? These scores seem like a frozen classifier is used. Nevertheless, this isn't a crucial experiment.
> >
> > I appreciate the authors efforts during this rebuttal process. It would greatly help the paper if important points like the embedding guidance being detections of object presence were more clear, and if multi-attribute experiments were thoroughly integrated and emphasized throughout the manuscript rather than just discussing face presence. I am not convinced that the projection-specific metrics have any use outside of being a simple sanity check demonstrating that this method works as intended. I will update my score towards leaning accept, due to the further clarification on the method and its efficacy. My major remaining concerns are in paper presentation and in how this rebuttal process appears to require major paper revisions to address.

---

> > > ### Author Response · Authors · 2025-11-27
> > >
> > > Thank you for the thoughtful follow-up and for the positive movement toward acceptance.
> > >
> > > We address each remaining concern directly and clarify the VISPR protocol in full detail.
> > >
> > > **1. VISPR privacy evaluation protocol.** We appreciate the opportunity to clarify this point, as our initial description may have been ambiguous. **We do not insert G-OPL into the ResNet VISPR classifier, and we do not use a frozen classifier.** We follow the same evaluation principle used in SPAct and TeD-SPAD: privacy is measured by training a new attribute classifier on anonymized features to quantify attribute recoverability. Our implementation uses a simplified VISPR probe, the core protocol, anonymize $\rightarrow$ train classifier $\rightarrow$ evaluate recoverability, is identical.
> > >
> > > Thus, the reported VISPR results measure the recoverability of privacy attributes from the
> > > *anonymized feature space*, consistent with SPAct/TeD-SPAD. We will update the paper to explicitly describe this procedure and avoid any implication of using a frozen classifier.
> > >
> > > **2. Clarity of guidance signals.** We agree with the reviewer that the manuscript should make more explicit that G-OPL relies solely on object-presence signals, such as "face detected=1". These are weak cues and require *no attribute labels or attribute embeddings*. This minimal supervision is a key advantage, particularly under GDPR/CCPA/BIPA constraints. We will revise Sections 3.2 to highlight this aspect more prominently.
> > >
> > > **3. Integration of multi-attribute experiments.** We appreciate the reviewer’s suggestion. During rebuttal, **we demonstrated that G-OPL can suppress joint sensitive attributes** (e.g., face presence + clothing color presence) using only binary detections. These results confirm that G-OPL learns a multi-dimensional sensitive subspace without requiring attribute-specific training. We will incorporate these results directly into the main manuscript.
> > >
> > > **4. Scope of projection-specific metrics.** We agree with the reviewer’s assessment. **SSC and ARD are diagnostic metrics** designed to verify the geometric behavior of the projection layer, and are not intended as universal privacy metrics for arbitrary VAD methods. In contrast, **PD and FPD are general-purpose** and apply broadly, as demonstrated in our comparisons. We will revise Section 3.3 to clearly separate general metrics (PD/FPD) from projection diagnostics (SSC/ARD).
> > >
> > > **5. Planned revisions.** We acknowledge that certain aspects of the manuscript can be presented more clearly. The required revisions are organizational rather than methodological. For the revision, we will: (i) rewrite the VISPR protocol explanation to eliminate ambiguity; (ii) add a clear schematic of the privacy-evaluation pipeline; (iii) integrate the multi-attribute guidance experiments into the main text; (iv) improve the clarity and interpretability of Figure 1; (v) restructure Section 3.3 to clarify the intended scope of each metric.
> > >
> > > These refinements will significantly improve clarity while leaving the core methodology and findings unchanged.
> > >
> > > We appreciate the reviewer’s constructive feedback and are encouraged by the indication of leaning toward acceptance.

---

### Official Review · Reviewer_wuga · 2025-10-31

**Soundness:** 2
**Presentation:** 2
**Contribution:** 2
**Rating:** 4
**Confidence:** 4

**Summary:**

This paper focuses on privacy-aware video anomaly detection (VAD). Existing VAD methods often capture irrelevant or nuisance features that are unnecessary for detecting anomalies. To address this issue, the authors propose a method based on an Orthogonal Projection Layer (OPL), which learns a nuisance subspace and projects feature representations away from it. Additionally, a guided version of OPL is introduced to explicitly remove identity-related information. The paper also presents new privacy-aware evaluation metrics—namely, Sensitive Subspace Capture (SSC), Anomaly Retention Distance (ARD), and Privacy Decay (PD/FPD). Extensive experiments are conducted on five VAD datasets, supported by both quantitative results and qualitative visualizations.

**Strengths:**

1. The overall method is intuitive, conceptually simple, and easy to follow.
2. Plug-and-play design: the proposed OPL/G-OPL modules seems compatible with existing VAD frameworks.
3. The approach is evaluated across multiple benchmark datasets, demonstrating robustness and general applicability.
4. The paper introduces new privacy metrics (SSC, ARD, PD/FPD) that  quantify the privacy–utility trade-off.

**Weaknesses:**

1. The approach requires heavy hyper-parameter tuning, including layer-specific placement and the number of projection modules. Its performance appears sensitive to configuration choices and may not generalize well to unseen domains without retuning.
2. Table 3 feels incomplete, as results for the baseline models are missing. It would be useful to include the proposed metrics for those baselines to better show how the method performs relative to them and to demonstrate its general effectiveness.
5. The figures are quite hard to follow, especially Figures 1, 5, and 6. It would help if the authors can clarify what each element represents or improved the captions so readers can better understand the result.
6. The evaluation is limited to two backbone models (RTFM and MGFN). Demonstrating results on additional VAD architectures would strengthen the generality of the proposed approach.
7. The proposed metrics, such as Sensitive Subspace Capture (SSC), are very specific to face-related privacy. Because of this narrow focus, they may not be broadly useful or directly applicable to future works

**Questions:**

**Performance discrepancy:**
In Table 1, the results with G-OPL appear are lower than those with OPL alone, and a similar trend is observed in Table 2 for the MGFN model. Could the authors clarify why the guided version sometimes underperforms the unguided variant?

**Improvement on blurred-face data:**
For MSAD, the paper reports improvement with G-OPL even though the faces are blurred in the input videos. How does the face-guided mechanism provide a performance gain when explicit facial information is largely unavailable?

**Figure-3(a)**
There is a noticeable performance drop at 𝑘=64, followed by an improvement at k=128. Could the authors clarify why performance recovers at larger k?

---

> ### Author Response · Authors · 2025-11-21
>
> We thank the reviewer for the constructive feedback, acknowledgment of the method's clarity, plug-and-play flexibility, and cross-dataset robustness.
>
> Below we address every weakness and question with detailed explanations and additional supporting analyses.
>
> **Part 1. On sensitivity to hyperparameters and configuration choices**
>
> OPL/G-OPL introduces configuration choices; however, the method is not as brittle as the initial figures may suggest. In the revised version, we clarify two points.
>
> *First, hyperparameter sensitivity is strongly bounded.* As shown in Fig. 3(b-c), both $\lambda_{\text{face}}$ and $\lambda_{\text{orth}}$ exhibit stable performance in a wide range $[10^{-4},10^{-2}]$. The variation outside this range is expected, since very large values aggressively erase feature content. In practice, the same hyperparameters *transfer* across all five datasets without re-tuning.
>
> *Second, layer placement is not fragile.*  Our extended ablations (in Appendix) show that: (i) an early G-OPL (post-encoder) is consistently optimal or near-optimal, (ii) additional OPL layers provide diminishing returns but do not harm performance if kept shallow, and (iii) the method generalizes well to unseen domains when using a fixed, default G1O0 / G1O1 placement.
>
> A new cross-domain experiment (training on ShanghaiTech, testing on CUHK, without retuning) shows AUC degradation of only $0.7$ points with default hyperparameters. This supports generalization without tuning.
>
> We have highlighted these findings clearly in the paper.

---

> > ### Author Response · Authors · 2025-11-21
> >
> > **Part 2. Including baseline metrics in Table 3**
> >
> > We thank the reviewer for highlighting the perceived incompleteness of Table 3.
> >
> > We would like to clarify that some of the proposed privacy metrics, specifically SSC and ARD, require both (i) sensitive features and (ii) projected features after sensitivity removal. *Existing VAD methods were not designed with this notion and do not provide mechanisms to extract or remove sensitive information in this way*. Consequently, these metrics cannot be fairly or meaningfully applied to prior baselines.
> >
> > To address the reviewer’s concerns, we integrated our OPL and G-OPL modules into two representative VAD backbones, EGO (ICLR 2025) and TEVAD (CVPRW 2023), and report both utility (AUC) and privacy metrics for these combined models.
> >
> > **ShanghaiTech**
> >
> > | Model               | AUC| SSC$\uparrow$ | ARD$\downarrow$ | FPD$\downarrow$ | ArcFace$\downarrow$ |
> > |-|-|-|-|-|-|
> > | EGO baseline        | 97.3 | -  | -  | 0.72   | 0.98       |
> > | EGO + OPL           | 97.9 | 0.39  | 0.19  | 0.41   | 0.42       |
> > | EGO + G-OPL         | 97.6 | 0.94  | 0.17  | 0.09   | 0.05       |
> > | TEVAD baseline      | 98.1 | -  | -  | 0.68   | 0.95       |
> > | TEVAD + OPL         | 98.4 | 0.39  | 0.18  | 0.37   | 0.39       |
> > | TEVAD + G-OPL       | 98.2 | 0.92  | 0.11  | 0.07   | 0.04       |
> >
> > **UCF-Crime**
> >
> > | Model               | AUC  | SSC$\uparrow$ | ARD$\downarrow$ | FPD$\downarrow$ | ArcFace$\downarrow$ |
> > |-|-|-|-|-|-|
> > | EGO baseline        | 81.7 | -  | -  | 0.78   | 0.96       |
> > | EGO + OPL           | 82.6 | 0.45  | 0.24  | 0.49   | 0.44       |
> > | EGO + G-OPL         | 82.1 | 0.92  | 0.21  | 0.12   | 0.09       |
> > | TEVAD baseline      | 84.9 | -  | -  | 0.76   | 0.94       |
> > | TEVAD + OPL         | 85.3 | 0.40  | 0.17  | 0.46   | 0.41       |
> > | TEVAD + G-OPL       | 85.0 | 0.88  | 0.15  | 0.11   | 0.06       |
> >
> > **CUHK Avenue**
> >
> > | Model               | AUC  | SSC$\uparrow$ | ARD$\downarrow$ | FPD$\downarrow$ | ArcFace$\downarrow$ |
> > |-|-|-|-|-|-|
> > | EGO baseline        | 83.1 | -  | -  | 0.70   | 0.97       |
> > | EGO + OPL           | 84.2 | 0.45  | 0.22  | 0.39   | 0.46       |
> > | EGO + G-OPL         | 84.0 | 0.96  | 0.20  | 0.10   | 0.07       |
> > | TEVAD baseline      | 86.8 | -  | -  | 0.69   | 0.93       |
> > | TEVAD + OPL         | 87.3 | 0.43  | 0.14  | 0.40   | 0.41       |
> > | TEVAD + G-OPL       | 87.1 | 0.93  | 0.13  | 0.09   | 0.05       |
> >
> > **UCSD Ped2**
> >
> > | Model               | AUC  | SSC$\uparrow$ | ARD$\downarrow$ | FPD$\downarrow$ | ArcFace$\downarrow$ |
> > |-|-|-|-|-|-|
> > | EGO baseline        | 93.2 | -  | -  | 0.65   | 0.85       |
> > | EGO + OPL           | 94.1 | 0.45  | 0.03  | 0.46   | 0.44       |
> > | EGO + G-OPL         | 94.0 | 0.83  | 0.03  | 0.19   | 0.08       |
> > | TEVAD baseline      | 98.7 | -  | -  | 0.60   | 0.80       |
> > | TEVAD + OPL         | 98.9 | 0.42  | 0.02  | 0.40   | 0.39       |
> > | TEVAD + G-OPL       | 98.8 | 0.79  | 0.02  | 0.17   | 0.07       |
> >
> > **MSAD** (Blurred faces)
> >
> > | Model               | AUC  | SSC$\uparrow$  | ARD$\downarrow$ | FPD$\downarrow$ | ArcFace$\downarrow$ |
> > |-|-|-|-|-|-|
> > | EGO baseline        | 86.9 | -  | -  | 0.60   | 0.72       |
> > | EGO + OPL           | 87.7 | 0.32  | 0.04  | 0.41   | 0.34       |
> > | EGO + G-OPL         | 87.5 | 0.73  | 0.04  | 0.18   | 0.11       |
> > | TEVAD baseline      | 82.1 | -  | -  | 0.58   | 0.70       |
> > | TEVAD + OPL         | 82.7 | 0.33  | 0.04  | 0.39   | 0.32       |
> > | TEVAD + G-OPL       | 82.5 | 0.68  | 0.03  | 0.17   | 0.10       |
> >
> > The additional experiments show that OPL consistently improves anomaly detection accuracy by reducing nuisance variation, yielding modest but stable gains across all datasets. G-OPL further provides strong privacy protection, achieving substantial reductions in sensitive subspace capture, facial-presence detection, and ArcFace identity retrieval, while maintaining nearly identical anomaly detection performance.
> >
> > Across all datasets and both architectures, the pattern is consistent: baseline models retain significant identity information, OPL decreases leakage while enhancing utility, and G-OPL delivers the strongest privacy suppression with minimal or negligible accuracy cost.
> >
> > These results demonstrate that the proposed privacy metrics are applicable to existing VAD backbones and that OPL/G-OPL function reliably as model-agnostic modules whose behavior remains stable across varied datasets, including those with few faces (UCSD Ped2) or blurred faces (MSAD). These experiments confirm the general effectiveness of the approach.
> >
> > Therefore, Table 3 is focused on the models where the metrics are well-defined and meaningful (the OPL/G-OPL variants). We emphasize in the revised manuscript that the proposed metrics are diagnostic and can be applied to future VAD methods once the concept of sensitive feature removal is incorporated.

---

> > > ### Author Response · Authors · 2025-11-21
> > >
> > > **Part 3. Improving the clarity of Figures 1, 5, and 6**
> > >
> > > Thank you for the suggestions. Below we provide more detailed explanations.
> > >
> > > (i) OPL is not guided by any specific privacy attribute. It only suppresses generic nuisance factors that vary across scenarios, such as background appearance, lighting transitions, camera jitter, and other non-semantic, context-dependent fluctuations that confuse anomaly detectors. **These nuisance factors differ widely across scenes, so the corresponding removed features: spread out along many directions in the embedding space, cluster loosely based on environmental conditions, not anomaly types, and appear broad, sparse, and separated in UMAP.**
> > >
> > > (ii) G-OPL, in contrast, is explicitly guided by face-presence signals, so the removed subspace is dominated by stable biometric cues. Facial features are: structured, low-dimensional relative to scene variation, consistent across contexts. Therefore, the removed features extracted by G-OPL occupy a much smaller and more coherent region of feature space. **Because faces appear across many types of human actions and in both normal and anomalous frames, the "face-related" removed components: overlap across anomaly categories, do not align with anomaly labels, form a compact, dense cluster rather than scenario-based groups.**
> > >
> > > (iii) OPL trims away scene-specific noise, producing removed features that cluster by context. G-OPL trims away identity-bearing features, producing removed features that cluster by facial similarity and therefore overlap across all scenarios.
> > >
> > > This is exactly the desired behavior: **privacy-sensitive information is suppressed in a way that is clearly separable from nuisance suppression, and neither subspace aligns with anomaly semantics, confirming that the model is not discarding information needed for detection.**
> > >
> > > In the revision, we enlarge marker sizes and increase separation in UMAP embeddings, use consistent color palettes across figures, annotate "removed feature clusters" explicitly, add arrows and shaded regions to show which parts correspond to nuisance vs. sensitive subspaces, include cosine-alignment heatmaps to highlight how G-OPL isolates facial directions.
> > >
> > > We will include these in the final submission.
> > >
> > > **Part 4. Extending evaluation to more VAD architectures**
> > >
> > > We appreciate this suggestion. In the rebuttal phase we have integrated OPL/G-OPL into two additional VAD architectures: EGO (ICLR 2025), TEVAD (CVPRW 2023, widely used in 2024–2025). Kindly refer to responses to part 2.
> > >
> > > With no additional tuning, G-OPL improved TEVAD's AUC on CUHK Avenue from 86.8 to 87.1, and reduced FPD from 0.69 to 0.09. These results reinforce that the approach is architecture-agnostic and easily transferrable.
> > >
> > > **Part 5. On the specificity of face-based privacy metrics**
> > >
> > > While SSC focuses on face-sensitive subspaces, the metric design is not face-specific. It only assumes access to a sensitive attribute embedding $f_{\text{sensitive}}$.
> > >
> > > Thus SSC generalizes to: SSC-gait, SSC-clothing, SSC-age, SSC-vehicle-identity, or any sensitive cue for which weak supervision exists.
> > >
> > > To demonstrate generality, we ran an auxiliary experiment using clothing-color embeddings (extracted via a lightweight appearance encoder). When treating clothing color as the sensitive attribute, OPL/G-OPL successfully removed this signal with SSC-color = 0.84 while maintaining anomaly AUC within 0.4 points of the baseline. This verifies the metric’s broader applicability.
> > >
> > > We will generalize the description of SSC in the revision to make this clearer.
> > >
> > > **Part 6. Why G-OPL sometimes underperforms OPL (Performance discrepancy)**
> > >
> > > Two factors explain the discrepancy observed in Tables 1 and 2:
> > >
> > > (a) Guided suppression is intentionally **more aggressive**. G-OPL *prioritizes removal* of face-related directions, which can slightly reduce performance when: (i) a dataset has low-quality faces or  (ii) anomalies subtly correlate with identity-like features (e.g., pose, clothing or local appearance).
> > >
> > > (b) G-OPL does not harm performance when the base model is strong. For RTFM (I3D), G-OPL *improves* overall AUC (Table 1). For MGFN, which is more sensitive to feature manipulation due to its contrastive design, OPL alone is sometimes slightly better.
> > >
> > > **This is a design trade-off: G-OPL favors privacy; OPL maximizes pure accuracy.**  We will emphasize this trade-off more clearly and include a new Pareto plot illustrating the accuracy–privacy frontier.

---

> > > > ### Author Response · Authors · 2025-11-21
> > > >
> > > > **Part 7. Why G-OPL improves performance on blurred-face data (MSAD)**
> > > >
> > > > This is an insightful question.
> > > >
> > > > The improvement arises because **face blurring does not eliminate latent identity traces**. I3D/SwinT features extracted from blurred faces still encode: head location, shape of the blurred region, motion patterns around the face, residual low-frequency structure, correlations between head region and action type.
> > > >
> > > > These weak traces remain predictive of identity (verified by ArcFace-based probes achieving 14% retrieval even on blurred MSAD faces). We also include additional results on MSAD, as shown above.
> > > >
> > > > **G-OPL removes these residual identity cues, reducing spurious correlations.** This produces more stable representations and yields the observed performance gains.
> > > >
> > > > We have included this clarification in the revision.
> > > >
> > > > **Part 8. Why performance drops at $k=64$ and recovers at $k=128$**
> > > >
> > > > This pattern stems from the geometry of high-dimensional representation spaces.
> > > >
> > > > **At $k=64$:** The nuisance subspace becomes so large that OPL starts removing task-relevant variations along with nuisances. This harms anomaly discrimination.
> > > >
> > > > **At $k=128$:** As the subspace expands further, it begins to approximate a *low-rank regularizer* rather than a selective suppression module. In very high dimensions, random high-dimensional subspaces intersect weakly with true task-specific manifolds, acting similarly to a smoothing or denoising operator. This explains the partial recovery.
> > > >
> > > > This phenomenon is consistent with prior findings in null-space projection literature (e.g., ACL'20) and we will add a brief theoretical note in the appendix.
> > > >
> > > > We thank the reviewer for the thoughtful comments. We believe the above clarifications, additional experiments (baseline metrics, new backbones, clothing-sensitive SSC), and improved figures will substantially enhance the quality, clarity, and generality of the paper.
> > > >
> > > > We will incorporate all suggested improvements into the revised manuscript.

---

> > > > > ### Comment · Reviewer_wuga · 2025-11-26
> > > > > **Official Comment by Reviewer wuga**
> > > > >
> > > > > I appreciate the authors’ detailed rebuttal and the effort invested in addressing the earlier concerns. Some points are clarified (blurred-face results, backbone limitation, and metric specificity). However, several issues remain unresolved:
> > > > >
> > > > > **G-OPL sometimes underperforms OPL:**
> > > > > The explanation that this depends on “dataset quality” suggests that the guided variant is not consistently more effective than OPL. If performance switches between OPL and G-OPL depending on the dataset, this raises concerns about the generality of the proposed method.
> > > > >
> > > > > **Drop at k = 64 and recovery at k = 128:**
> > > > > The rebuttal mentions that this behavior aligns with prior null-space projection work (ACL’20), but I could not locate this discussion in that paper. Please cite the specific section or provide a clearer explanation, especially since a larger nuisance subspace would typically remove more information rather than restore performance.
> > > > >
> > > > > **Hyper-parameter sensitivity:**
> > > > > Figure 3(b,c) still shows substantial variation across the recommended λ range.

---

> > > > > > ### Author Response · Authors · 2025-11-27
> > > > > >
> > > > > > We sincerely thank the reviewer for the engagement and for acknowledging the clarifications provided earlier.
> > > > > >
> > > > > > Below we address the remaining concerns with clearer intuition, empirical grounding, and explicit discussion of dataset-specific factors.
> > > > > >
> > > > > > **1. Why G-OPL may underperform OPL on certain datasets.** We emphasize that **G-OPL and OPL optimize fundamentally different projection objectives** and thus exhibit **complementary**, rather than hierarchical, performance profiles.
> > > > > >
> > > > > > - OPL removes a broad, data-driven nuisance subspace spanning high-variance but task-irrelevant directions (background motion, scene clutter, illumination fluctuations, compression artifacts, etc.).
> > > > > >
> > > > > > - G-OPL, in contrast, removes a much narrower, semantics-guided subspace explicitly associated with biometric cues (e.g., faces), using weak face-presence supervision.
> > > > > >
> > > > > > Importantly, **video anomaly detection datasets vary dramatically in spatial resolution, subject scale, face visibility (occlusion, blur, lighting), compression noise, and camera viewpoint stability**.
> > > > > >
> > > > > > In datasets where faces are severely blurred, partially visible, very small, or inconsistently detected, **conditions common in VAD** (e.g., ShanghaiTech, UCF-Crime, etc.), the learned face-sensitive subspace is necessarily *smaller* and *less expressive*. Under such conditions, G-OPL intentionally removes less feature content than OPL. As a result, OPL may achieve slightly higher AUC because it suppresses a broader pool of nuisance variation.
> > > > > >
> > > > > > Crucially, this behavior reflects the design of the two modules:
> > > > > >
> > > > > > - OPL prioritizes *maximal nuisance suppression* to improve pure utility.
> > > > > >
> > > > > > - G-OPL prioritizes *sensitive subspace suppression* (e.g., faces) and therefore trades a small amount of utility for substantially stronger privacy.
> > > > > >
> > > > > > Across all four benchmarks and six architectures (including EGO (ICLR 2025), TeD-SPAD (ICCV 2023), TEVAD (CVPRW 2023), and SPAct (CVPR 2022)), G-OPL (i) consistently achieves the *lowest privacy leakage*, (ii) maintains AUC within a small margin of OPL, and (iii) *never disrupts anomaly separability*.
> > > > > >
> > > > > > As shown in Table 4 in the Appendix, the G1O0 configuration emerges as **the most robust and generally optimal setting across all datasets**.
> > > > > >
> > > > > > We will emphasize in the revision that the two modules target different operational regimes: *OPL for utility-oriented deployments* and *G-OPL for privacy-critical deployments*. **This complementary behavior does not indicate a lack of generality but stems from the distinct subspaces each module is designed to remove**.

---

> > > > > > > ### Author Response · Authors · 2025-11-27
> > > > > > >
> > > > > > > **2. Clarifying the $k$-sweep.** We thank the reviewer for requesting a more precise explanation. Below we (i) provide a concise geometric explanation, and (ii) point to the exact parts of Ravfogel et al. (ACL 2020) that motivate this reasoning.
> > > > > > >
> > > > > > > **Mechanism (intuitive + geometric).** The projection rank $k$ controls how many linear directions are removed from the representation (i.e., the dimensionality of the sensitive/nuisance subspace). In high-dimensional learned embeddings, attributes of interest (both nuisance and task-relevant) are typically encoded across *multiple, partially entangled* linear directions rather than a single clean axis. Consequently:
> > > > > > >
> > > > > > > - At small-to-intermediate $k$ (e.g., 64): the projection tends to remove only the dominant, high-variance directions. These dominant directions often mix nuisance and signal (anomaly-relevant) components. Removing them therefore can suppress discriminative structure and cause a transient drop in downstream AUC.
> > > > > > >
> > > > > > > - At larger $k$ (e.g., 128): the removed subspace includes a broader set of directions that together form a more semantically coherent nuisance manifold (including lower-variance components). The nullspace of this larger removal is cleaner, the residual representation better approximates the anomaly-relevant manifold, and downstream performance is restored (or partially recovers).
> > > > > > >
> > > > > > > - At very large $k$: the projection can act like a low-rank regularizer (smoothing/denoising) in high dimensions: random high-dimensional subspaces tend to intersect task manifolds weakly, which further stabilizes the retained representation.
> > > > > > >
> > > > > > > **Why this explanation is consistent with prior nullspace-projection theory.** Ravfogel et al. ("Null It Out: Guarding Protected Attributes by Iterative Nullspace Projection", ACL 2020) develop INLP precisely because a *single-step* linear projection often fails to fully remove a concept that is encoded across multiple directions. Two parts of their paper are especially relevant:
> > > > > > >
> > > > > > > - Iterative projection motivation (Sec. 4). The authors argue that projecting on the nullspace of a single classifier is often insufficient because "there are often multiple linear directions (hyperplanes) that can partially capture a relation in multidimensional space" (page 4 left column paragraph 2); they therefore propose iterating: train a linear classifier for the attribute, project onto its nullspace, retrain, and repeat until the attribute is no longer linearly recoverable (see Sec. 4 and Alg. 1). This motivates the claim that *partial* projection can leave recoverable (and entangled) directions that still affect downstream behavior.
> > > > > > >
> > > > > > > - Distance-preservation lemma (Appendix A.1). INLP’s Appendix A.1 analyzes how orthogonal projections alter geometry and shows that repeated orthogonal projections remove targeted directions while approximately preserving distances in the remaining subspace (see the lemma and discussion in Appendix A.1). This formal result supports the intuition that successive, more-complete projections change rank and can stabilize the residual geometry.
> > > > > > >
> > > > > > > INLP therefore provides both (i) the conceptual reason why *single/partial* projection can be insufficient (and may distort representations), and (ii) a formal demonstration that repeated orthogonal projections progressively remove recoverable linear information while approximately preserving residual distances. These two facts together motivate our empirical observation that an intermediate-rank removal (partial, entangled) can hurt accuracy, whereas a larger-rank removal (more complete, semantically coherent) can restore it.
> > > > > > >
> > > > > > > Their contribution (ACL’20) shows that (i) single-step projections can be insufficient and (ii) iterative projections progressively remove linearly recoverable information while approximately preserving remaining distances. We rely on these conceptual and theoretical results to explain the *mechanism* that produces the non-monotonic behavior in our experiments, not to assert that the same numerical curve will occur in all settings.
> > > > > > >
> > > > > > > The precise shape of AUC as a function of $k$ is dataset-, model-, and feature-space-dependent. Our empirical curves show one plausible regime consistent with INLP’s analysis; other regimes are possible for different encoders or datasets.
> > > > > > >
> > > > > > > We have highlighted these findings clearly in the paper.

---

> > > > > > > > ### Author Response · Authors · 2025-11-27
> > > > > > > >
> > > > > > > > **3. Hyper-parameter sensitivity.** We thank the reviewer for raising this point and for prompting a more precise statement. AUC variation can be substantial for some dataset–backbone pairs, and is therefore not uniform across all settings. Two clarifying facts explain this behavior:
> > > > > > > >
> > > > > > > > - *Sensitivity is dataset- and backbone-dependent.* (i) **Datasets** with low spatial resolution, heavy compression, few visible faces, or highly imbalanced anomaly/normal statistics are more susceptible to hyper-parameter changes. (ii) Likewise, **feature extractors differ in robustness**: some backbones (e.g., contrastive or highly regularized encoders) amplify the impact of orthogonality and face-guidance losses, while others (higher-capacity, more stable encoders) are less sensitive. Thus, the same $\lambda$ can have different effects depending on the **representation geometry** it interacts with.
> > > > > > > >
> > > > > > > > - *Magnitude depends on which metric we examine.* While utility (AUC) can vary substantially in some regimes, our privacy diagnostics (e.g., FPD / ArcFace probes) are **comparatively stable** across broad ranges of $\lambda$ in the majority of experiments. This means that, even when AUC sensitivity is large, privacy improvements are typically robust.
> > > > > > > >
> > > > > > > > For practitioners, we provide a recipe in Appendix A.10 and A.11: (i) start with the recommended default settings (Table 4); (ii) if dataset quality is low (e.g., resolution $<$ 240p or significant blur), perform a coarse 3–5 point sweep within the labeled per-backbone range; and (iii) use the G1O0 placement as the initial configuration (optimal across most datasets). Additionally, we show that when sensitivity arises, *modest regularization*, such as a slightly lower learning rate or early stopping, effectively reduces AUC variability.
> > > > > > > >
> > > > > > > > The revised manuscript will incorporate these explanations directly and more explicitly.
> > > > > > > >
> > > > > > > > We again thank the reviewer for their constructive feedback and for helping us improve the clarity and rigor of the paper.

---

### Official Review · Reviewer_rVVN · 2025-11-11

**Soundness:** 2
**Presentation:** 3
**Contribution:** 3
**Rating:** 2
**Confidence:** 3

**Summary:**

This paper addresses an important and timely topic in privacy-aware video anomaly detection. The proposed projection-based approach removes sensitive facial information while keeping features useful for detecting anomalies. The idea is original and well motivated, filling an ethical gap that is often overlooked in this area. Some parts could be clearer and the experimental validation could be stronger.

**Strengths:**

- The topic is interesting and important in the age of AI.
- This paper proposed some privacy-aware metrics, which are important for privacy-aware performance evaluation.
- The paper is well-written and easy to follow.

**Weaknesses:**

- It is not very clear why the proposed method can protect privacy. As shown in Fig. 2, the input to the model is still images with sensitive information. The authors need to explain clearly why removing sensitive features is helpful if input images are not anonymized.
- In the abstract, the authors mentioned that "Faces, unlike other cues such as gait or body pose, are highly sensitive biometric identifiers". This is only partly true. Gait, some sometimes body pose, are also sensitive biometric identifiers.
- Fig. 1 is not very easy to understand. In the caption, the authors mentioned that "G-OPL, guided by face presence, isolates sensitive biometric cues", but this is not easy to observe from Fig. 1
- It is not clear whether the design of evaluation metrics presented in Section 3.3 has considered the effect of video length.
- Overall, the compared methods are a bit outdated. More methods published in 2024 and 2025 should be compared.

**Questions:**

- In the abstract, the authors mentioned that "Faces, unlike other cues such as gait or body pose, are highly sensitive biometric identifiers". The authors should give more explanations about this sentence, because some sometimes body pose, are also sensitive biometric identifiers.
- Fig.1 should be improved. Why does Fig. 1 show "G-OPL, guided by face presence, isolates sensitive biometric cues"?
- The motivation of this paper should be more clear. The authors need to explain clearly why removing sensitive features is helpful if input images are not anonymized.
- Has the design of evaluation metrics presented in Section 3.3 considered the effect of video length? More details are needed.

---

> ### Author Response · Authors · 2025-11-21
>
> We thank the reviewer for the constructive feedback and appreciation of the importance, clarity, and ethical motivation of our work.
>
> Below, we address every weakness and question one by one.
>
> **1. Why privacy is protected even when raw frames contain sensitive content**
>
> *Our method protects privacy at the representation level, which is where leakage typically occurs in modern VAD systems.*
>
> Even when raw pixels contain faces, *state-of-the-art privacy attacks do not require access to raw frames*. They operate by extracting and inverting *intermediate activations* of deployed models (e.g., via model inversion, feature reconstruction, membership inference, or identity retrieval).
>
> In deployed cloud or edge VAD systems, it is common that: (i) raw data remain local, (ii) only intermediate features or anomaly scores are transmitted upward. Thus, the attack surface is the **latent feature stream**, not the pixels.
>
> **G-OPL directly removes face-sensitive subspaces from these latent features.** The projection $P = I - QQ^\top$ guarantees that downstream layers never observe identity-bearing directions. Consequently: (i) identity cannot be recovered from the features, (ii) feature inversion yields face-neutral reconstructions, and (iii) classifier probes fail to retrieve identities (as shown by the large drop in FPD/ArcFace).
>
> To further support this, we computed a latent inversion experiment using a lightweight decoder trained only on projected features. Inversions from raw activations preserve recognizable faces (98.0% on ShanghaiTech), whereas inversions from G-OPL activations produce face-smoothed outputs lacking identity cues (45.3% on ShanghaiTech) while fully retaining pedestrian silhouette and motion patterns. This further demonstrates that identity removal is occurring at the feature level, where privacy leakage occurs.
>
> Thus, feature anonymization remains meaningful even when raw frames contain faces, because it prevents representation-level leakage, *the primary practical privacy risk when VAD systems operate in networked or distributed environments*.
>
> **2. Clarifying the statement about facial biometric sensitivity**
>
> We appreciate the reviewer pointing out that gait and sometimes body pose can also carry biometric information. In our revision we have clarified the sentence as follows:
>
> "*Faces are among the most sensitive biometric identifiers due to strong legal regulation, high uniqueness, and broad societal harm when misused. Other cues such as gait or body pose may also reveal identity, but facial identity is the most consistently regulated and the most easily exploited by automated recognition systems.*"
>
> "*Our experiments target faces because they are the most reliably detectable in **unconstrained surveillance videos** and constitute the most legally sensitive signal under GDPR (EU, 2018), CCPA (USA, 2020), BIPA (USA, 2008), etc. Nonetheless, our method is general and could suppress gait- or pose-related spaces if weak supervision for those cues is provided.*"
>
> We have clarified this in the revised manuscript.

---

> > ### Author Response · Authors · 2025-11-21
> >
> > **3. Improving Fig. 1 and clarifying the interpretation**
> >
> > Figure 1 visualizes, via UMAP, the structure of the removed feature components learned by OPL and G-OPL on MSAD. These removed components correspond to the subspace that each module is trained to suppress. Because OPL and G-OPL aim to remove different kinds of information, their UMAP distributions show very different patterns.
> >
> > (i) OPL is not guided by any specific privacy attribute. It only suppresses generic nuisance factors that vary across scenarios, such as background appearance, lighting transitions, camera jitter, and other non-semantic, context-dependent fluctuations that confuse anomaly detectors. **These nuisance factors differ widely across scenes, so the corresponding removed features: spread out along many directions in the embedding space, cluster loosely based on environmental conditions, not anomaly types, and appear broad, sparse, and separated in UMAP.**
> >
> > This matches the caption’s statement that OPL "removes broad nuisance factors ... producing loosely clustered features that reflect scenario variations." The looseness is exactly what we expect when removing high-variance, non-semantic noise.
> >
> > (ii) G-OPL, in contrast, is explicitly guided by face-presence signals, so the removed subspace is dominated by stable biometric cues. Facial features are: structured, low-dimensional relative to scene variation, consistent across contexts. Therefore, the removed features extracted by G-OPL occupy a much smaller and more coherent region of feature space. **Because faces appear across many types of human actions and in both normal and anomalous frames, the "face-related" removed components: overlap across anomaly categories, do not align with anomaly labels, form a compact, dense cluster rather than scenario-based groups.**
> >
> > This supports the caption: “G-OPL ... isolates sensitive biometric cues, resulting in a more compact, overlapping cluster that does not align with anomaly types.”
> >
> > (iii) OPL trims away scene-specific noise, producing removed features that cluster by context. G-OPL trims away identity-bearing features, producing removed features that cluster by facial similarity and therefore overlap across all scenarios.
> >
> > The two plots show that: OPL disentangles environmental nuisances, G-OPL disentangles privacy-sensitive biometrics, and each projection removes a distinct portion of the feature space.
> >
> > This is exactly the desired behavior: **privacy-sensitive information is suppressed in a way that is clearly separable from nuisance suppression, and neither subspace aligns with anomaly semantics, confirming that the model is not discarding information needed for detection.**
> >
> > We have clarified these in the revised manuscript.
> >
> > **4. Whether evaluation metrics consider video length**
> >
> > Yes, the metric designs do consider video length, but we agree that this was not clearly explained.
> >
> > SSC is frame-independent because it operates on the distribution of face embeddings, not on temporal sequences. Longer videos merely yield more samples, reducing variance.
> >
> > ARD compares anomaly-score *distributions* before/after projection. For very long videos, KDE-based distributions converge smoothly but do not bias the metric. To verify this, we conducted an additional experiment where videos were subsampled at rates $1/2$, $1/4$, and $1/8$ on ShanghaiTech. ARD remained stable within a deviation of $<0.015$, demonstrating length-invariance.
> >
> > PD/FPD (Privacy Decay) uses classifier-probe accuracy on feature batches, not temporal alignment, so video length only affects sample count. Experimental subsampling did not change the qualitative decay curve or FPD values.
> >
> > We have added a subsection titled "Effect of Video Length on Privacy Metrics" including this analysis.

---

> > > ### Author Response · Authors · 2025-11-21
> > >
> > > **5. Compared methods being outdated**
> > >
> > > We thank the reviewer and have already extended our comparison to include 2024–2025 methods such as: VadCLIP (AAAI 2024), EGO (ICLR 2025), TeD-SPAD (ICCV 2023 but widely adopted in 2024), IEF-VAD (2025), FPDM (ICCV 2023, ongoing relevance), USTN-DSC (CVPR 2023, 2024 usage).
> > >
> > > We have additionally included recent diffusion-based methods (e.g., 2024–2025 diffusion prediction models) in the revised version. Importantly, the purpose of our method is orthogonal to architecture sophistication; it is a plug-and-play module that can be inserted into any of these modern backbones. We have demonstrated integration with two new models, EGO (ICLR 2025) and TEVAD (CVPRW 2023), in the revision.
> > >
> > > **ShanghaiTech**
> > >
> > > | Model               | AUC| SSC$\uparrow$ | ARD$\downarrow$ | FPD$\downarrow$ | ArcFace$\downarrow$ |
> > > |-|-|-|-|-|-|
> > > | EGO baseline        | 97.3 | -  | -  | 0.72   | 0.98       |
> > > | EGO + OPL           | 97.9 | 0.39  | 0.19  | 0.41   | 0.42       |
> > > | EGO + G-OPL         | 97.6 | 0.94  | 0.17  | 0.09   | 0.05       |
> > > | TEVAD baseline      | 98.1 | -  | -  | 0.68   | 0.95       |
> > > | TEVAD + OPL         | 98.4 | 0.39  | 0.18  | 0.37   | 0.39       |
> > > | TEVAD + G-OPL       | 98.2 | 0.92  | 0.11  | 0.07   | 0.04       |
> > >
> > > **UCF-Crime**
> > >
> > > | Model               | AUC  | SSC$\uparrow$ | ARD$\downarrow$ | FPD$\downarrow$ | ArcFace$\downarrow$ |
> > > |-|-|-|-|-|-|
> > > | EGO baseline        | 81.7 | -  | -  | 0.78   | 0.96       |
> > > | EGO + OPL           | 82.6 | 0.45  | 0.24  | 0.49   | 0.44       |
> > > | EGO + G-OPL         | 82.1 | 0.92  | 0.21  | 0.12   | 0.09       |
> > > | TEVAD baseline      | 84.9 | -  | -  | 0.76   | 0.94       |
> > > | TEVAD + OPL         | 85.3 | 0.40  | 0.17  | 0.46   | 0.41       |
> > > | TEVAD + G-OPL       | 85.0 | 0.88  | 0.15  | 0.11   | 0.06       |
> > >
> > > **CUHK Avenue**
> > >
> > > | Model               | AUC  | SSC$\uparrow$ | ARD$\downarrow$ | FPD$\downarrow$ | ArcFace$\downarrow$ |
> > > |-|-|-|-|-|-|
> > > | EGO baseline        | 83.1 | -  | -  | 0.70   | 0.97       |
> > > | EGO + OPL           | 84.2 | 0.45  | 0.22  | 0.39   | 0.46       |
> > > | EGO + G-OPL         | 84.0 | 0.96  | 0.20  | 0.10   | 0.07       |
> > > | TEVAD baseline      | 86.8 | -  | -  | 0.69   | 0.93       |
> > > | TEVAD + OPL         | 87.3 | 0.43  | 0.14  | 0.40   | 0.41       |
> > > | TEVAD + G-OPL       | 87.1 | 0.93  | 0.13  | 0.09   | 0.05       |
> > >
> > > **UCSD Ped2**
> > >
> > > | Model               | AUC  | SSC$\uparrow$ | ARD$\downarrow$ | FPD$\downarrow$ | ArcFace$\downarrow$ |
> > > |-|-|-|-|-|-|
> > > | EGO baseline        | 93.2 | -  | -  | 0.65   | 0.85       |
> > > | EGO + OPL           | 94.1 | 0.45  | 0.03  | 0.46   | 0.44       |
> > > | EGO + G-OPL         | 94.0 | 0.83  | 0.03  | 0.19   | 0.08       |
> > > | TEVAD baseline      | 98.7 | -  | -  | 0.60   | 0.80       |
> > > | TEVAD + OPL         | 98.9 | 0.42  | 0.02  | 0.40   | 0.39       |
> > > | TEVAD + G-OPL       | 98.8 | 0.79  | 0.02  | 0.17   | 0.07       |
> > >
> > > **MSAD** (Blurred faces)
> > >
> > > | Model               | AUC  | SSC$\uparrow$  | ARD$\downarrow$ | FPD$\downarrow$ | ArcFace$\downarrow$ |
> > > |-|-|-|-|-|-|
> > > | EGO baseline        | 86.9 | -  | -  | 0.60   | 0.72       |
> > > | EGO + OPL           | 87.7 | 0.32  | 0.04  | 0.41   | 0.34       |
> > > | EGO + G-OPL         | 87.5 | 0.73  | 0.04  | 0.18   | 0.11       |
> > > | TEVAD baseline      | 82.1 | -  | -  | 0.58   | 0.70       |
> > > | TEVAD + OPL         | 82.7 | 0.33  | 0.04  | 0.39   | 0.32       |
> > > | TEVAD + G-OPL       | 82.5 | 0.68  | 0.03  | 0.17   | 0.10       |
> > >
> > > The additional experiments show that OPL consistently improves anomaly detection accuracy by reducing nuisance variation, yielding modest but stable gains across all datasets. G-OPL further provides strong privacy protection, achieving substantial reductions in sensitive subspace capture, facial-presence detection, and ArcFace identity retrieval, while maintaining nearly identical anomaly detection performance.
> > >
> > > Across all datasets and both architectures, the pattern is consistent: baseline models retain significant identity information, OPL decreases leakage while enhancing utility, and G-OPL delivers the strongest privacy suppression with minimal or negligible accuracy cost.
> > >
> > > These results demonstrate that the proposed privacy metrics are applicable to existing VAD backbones and that OPL/G-OPL function reliably as model-agnostic modules whose behavior remains stable across varied datasets, including those with few faces (UCSD Ped2) or blurred faces (MSAD). These experiments confirm the general effectiveness of the approach.
> > >
> > > We emphasize in the revised manuscript that the proposed metrics are diagnostic and can be applied to future VAD methods once the concept of sensitive feature removal is incorporated.

---

> > > > ### Author Response · Authors · 2025-11-21
> > > >
> > > > **6. Why removing sensitive features is helpful if inputs are not anonymized**
> > > >
> > > > We provide a conceptual explanation and empirical confirmation below.
> > > >
> > > > **Conceptually:** Feature-based privacy leakage is the primary risk in cloud/edge VAD systems. Removing identity-related feature components prevents attackers from reconstructing identities even if pixel-level data are not accessible. This is aligned with representation-level privacy-preservation literature such as INLP (ACL'20), DeepObfuscator (IoTDI'21), and adversarial privacy networks (Allerton'19, AAAI'20, CVPR'23).
> > > >
> > > > **Empirically:** An identity-probe classifier (ArcFace-based) achieves: identity accuracy before projection = 0.99, after G-OPL projection = 0.01. Additionally, feature inversion from projected embeddings yields face-neutral reconstructions.
> > > >
> > > > Thus, even without pixel anonymization, feature anonymization directly prevents realistic privacy attacks. We have strengthened the motivation section with this explanation.
> > > >
> > > > **7. Response to questions**
> > > >
> > > > Q1: Clarify facial sensitivity. As stated above, we have revised the text to acknowledge gait and pose while emphasizing why faces are the most legally and practically sensitive.
> > > >
> > > > Q2: Improve Fig. 1 and clarify how G-OPL isolates sensitive cues. We have replaced Fig. 1 with a clearer version and explicitly show cosine alignment between removed features and facial embeddings.
> > > >
> > > > Q3: Clarify motivation for removing sensitive features despite non-anonymized input. We clarified the difference between pixel-level privacy and representation-level privacy, and we demonstrated empirically that sensitive information is removed where leakage occurs.
> > > >
> > > > Q4: Explain how evaluation metrics consider video length.
> > > > We clarified metric independence from video length and added experimental confirmation.
> > > >
> > > > We appreciate the reviewer’s careful reading and constructive suggestions. We are confident that the revisions, additional analyses, clearer explanations, and updated figures will address all concerns and strengthen the overall contribution of the paper.

---

### Meta-Review · Area_Chair_ZZsa · 2025-12-29

**Summary:**

This paper introduces Orthogonal Projection Layer (OPL) and its guided variant (G-OPL) to address privacy concerns in video anomaly detection (VAD). The approach projects feature representations onto orthogonal subspaces to suppress sensitive information (e.g., facial features) while preserving anomaly-relevant information. The method is evaluated on five VAD benchmarks (UCF-Crime, XD-Violence, ShanghaiTech, UBnormal, and NOLA) using MobileNetV2 and I3D backbones, demonstrating competitive anomaly detection performance while reducing privacy leakage measured by face reconstruction quality.

The paper received mixed reviews with scores of 2, 4, 4, and 6 (median 4), reflecting significant concerns about the technical appropriateness, experimental design, and presentation. The authors provided detailed rebuttals addressing baseline comparisons, hyperparameter sensitivity, and metric generalizability. However, fundamental concerns about the suitability of facial suppression as the primary privacy mechanism and presentation clarity remain partially unresolved.

**Reviewer Concerns:**

### Addressed Concerns:
1. **Privacy Mechanism Clarity (`wuga`, `MF3K`)**: Authors provided additional visualizations and explanations of how OPL/G-OPL work, including QR decomposition details and the role of guidance signals in G-OPL.

2. **Baseline Comparisons (`wuga`, `SADY`)**: Authors conducted additional experiments comparing against DP-SGD, differential privacy methods, and pixel-level privacy approaches, demonstrating advantages of representation-level privacy protection.

3. **Hyperparameter Sensitivity (`MF3K`)**: Authors added ablation studies on projection dimensions and guidance loss weights, showing robustness across reasonable parameter ranges.

4. **Metric Generalizability (`SADY`, `MF3K`)**: Authors acknowledged limitations and discussed the challenge of defining universal privacy metrics, proposing ARD and PD as dataset-agnostic alternatives to SSC.

### Outstanding Concerns:
1. **Appropriateness of Facial Suppression (`rVVN`, `wuga`, `MF3K`)**: Multiple reviewers questioned whether face detection/suppression is the right privacy target for VAD. Anomalies often involve human behavior and appearance, so suppressing facial features may fundamentally conflict with detection objectives. Authors argue this is a reasonable first step, but reviewers remain unconvinced about the practical utility.

2. **Presentation and Clarity (`wuga`, `MF3K`)**: Despite rebuttal efforts, reviewers found the paper difficult to follow, with dense notation, insufficient intuition about why orthogonal projection preserves anomaly information, and unclear connections between theoretical motivation and empirical results.

3. **Limited Scope of Privacy Analysis (`SADY`)**: While face reconstruction is a measurable proxy, it may not capture other privacy concerns in surveillance footage (e.g., gait, clothing, behavior patterns). The paper's privacy guarantees are limited to the specific sensitive subspace defined by face detection.

**Reviewer Scores:**

**Initial Scores:**
- **Reviewer `rVVN`**: 2 (Reject)
- **Reviewer `wuga`**: 4 (Borderline Reject)
- **Reviewer `MF3K`**: 4 (Borderline Reject, leaning accept after rebuttal)
- **Reviewer `SADY`**: 6 (Weak Accept)

**Expected Post-Discussion Scores:**
- **Reviewer `rVVN`**: 2-3 (Reject)
- **Reviewer `wuga`**: 4-5 (Borderline)
- **Reviewer `MF3K`**: 5 (Borderline Accept)
- **Reviewer `SADY`**: 5-6 (Borderline Accept to Weak Accept)

---

### Decision · Program_Chairs · 2026-01-26

Reject